## OPEN

# Proton-gated anion transport governs macropinosome shrinkage

Mariia Zeziulia [1,2,3], Sandy Blin[1,2], Franziska W. Schmitt[1,2,4], Martin Lehmann[1] and Thomas J. Jentsch [1,2,5] ✉

Intracellular organelles change their size during trafficking and maturation. This requires the transport of ions and water across their membranes. Macropinocytosis, a ubiquitous form of endocytosis of particular importance for immune and cancer cells, generates large vacuoles that can be followed optically. Shrinkage of macrophage macropinosomes depends on TPC-mediated $Na^+$ efflux and $Cl^-$ exit through unknown channels. Relieving osmotic pressure facilitates vesicle budding, positioning osmotic shrinkage upstream of vesicular sorting and trafficking. Here we identify the missing macrophage $Cl^-$ channel as the proton-activated $Cl^-$ channel ASOR/TMEM206. ASOR activation requires $Na^+$-mediated depolarization and luminal acidification by redundant transporters including $H^+$-ATPases and CLC $2Cl^-/H^+$ exchangers. As corroborated by mathematical modelling, feedback loops requiring the steep voltage and pH dependencies of ASOR and CLCs render vacuole resolution resilient towards transporter copy numbers. *TMEM206* disruption increased albumin-dependent survival of cancer cells. Our work suggests a function for the voltage and pH dependence of ASOR and CLCs, provides a comprehensive model for ion-transport-dependent vacuole maturation and reveals biological roles of ASOR.

With the exception of tubular structures, reduction of vesicle size cannot occur solely by the release of smaller vesicles because vesicle volume depends on the third power, and surface area on the square, of the radius. Although small amounts of luminal fluid might be taken up by tubular endolysosomes in 'kiss-and-run' processes[1], decrease of vesicle volume requires transmembrane water flux that is driven by osmotic gradients. The shrinkage ('resolution') of the large vacuoles generated by macropinocytosis[2–4] can be followed in macrophages in pulse-chase experiments. Macropinosome (MP) resolution requires luminal $Na^+$ and $Cl^-$ (ref. [5]). Whereas $Na^+$ flows through TPC1 and TPC2 cation channels[5], the channel(s) mediating the required parallel $Cl^-$ conductance remained unknown[5,6]. In this Article, using macrophages from mice with disrupted $Cl^-$ transporter genes, we set out to identify the underlying channel, elucidate its interaction with other vesicular ion transporters in vesicle maturation and study its biological roles.

## Results

In bone-marrow-derived macrophages (BMDMs), addition of macrophage colony-stimulating factor (M-CSF) rapidly triggers the formation of MPs, which can be labelled by engulfed dextran-coupled fluorescent dyes[7]. Their initial luminal ion composition closely mirrors that of the extracellular medium. We followed the volume of 70 kDa tetramethylrhodamine (TMR)–dextran containing MPs from 5 to 15 min after M-CSF addition in live cell imaging (Fig. 1a,b). MPs shrank to roughly 20% of initial volume during this period (Fig. 1c,d and Supplementary Video 1). Concomitantly, vesicular fluorescence intensity, reflecting luminal TMR–dextran concentration, increased by ~50% (Fig. 1c,d). As described[5], substitution of luminal $Na^+$ (by N-methyl-D-glucamine) or $Cl^-$ (by gluconate) strongly impaired both MP resolution and the increase in fluorescence (Fig. 1c,d).

**Identification of MP $Cl^-$ conductance.** Several candidates were proposed to mediate the $Cl^-$ conductance in parallel to TPC $Na^+$ channels[6]. These include CLC $2Cl^-/H^+$ exchangers[8], CLIC1[9] and VRAC/LRRC8 anion channels[10] (Fig. 1e). Despite their name, CLICs (*CL*Intracellular *C*hannels) are unlikely to form physiological $Cl^-$ channels[9]. Volume-regulated anion channels (VRACs), hexamers containing LRRC8A (refs. [11,12]) and at least one other LRRC8 isoform[11], reside in the plasma membrane, but contradictory reports[13,14] suggest that VRACs may also regulate lysosomal volume. Because MPs are initially formed by the plasma membrane, we also considered the plasma membrane $Cl^-$ channel ClC-2 and the ubiquitously expressed acid-activated anion channel ASOR[15,16] (also known as PAORAC[15] or PAC[17]), which we[18] and others[17] have recently identified as being formed by TMEM206 proteins. ClC-2 and ClC-6 were almost absent, ClC-3 and ClC-4 barely detectable, and ClC-5, ClC-7, LRRC8A and TMEM206 robustly expressed in BMDMs (Fig. 1f–m and Extended Data Fig. 1). We investigated the subcellular localization of the most highly expressed candidates, ClC-7 and TMEM206, in BMDMs under resting conditions (Fig. 2a,c,d and Extended Data Fig. 2) and after M-CSF exposure (Fig. 3). As in other cells[19,20], ClC-7 co-localized with lysosomal LAMP1 and partially with late endosomal rab7, but not with early endosomal EEA1 (Fig. 2d and Extended Data Fig. 2c). Unlike ClC-7, ASOR/TMEM206 was detected at the plasma membrane (Fig. 2a) where it mediated typical acid-activated, outwardly rectifying $Cl^-$ currents (Fig. 2b). However, most of TMEM206 was intracellular where it co-localized with EEA1 and rab5, but only partially with LAMP1 and rab7 (Fig. 2c and Extended Data Fig. 2a). Confirming a recent report[21], ASOR was found on endosomes also in human embryonic kidney (HEK) cells (Extended Data Fig. 3).

M-CSF rapidly induced the formation of MPs that were early on positive for EEA1 and rab5, but not rab7 (Fig. 3a,d at 4 min), which they acquired later (Fig. 3b,e at 7 min). TMEM206, but not ClC-7,

[1]Leibniz-Forschungsinstitut für Molekulare Pharmakologie (FMP), Berlin, Germany. [2]Max-Delbrück-Centrum für Molekulare Medizin (MDC), Berlin, Germany. [3]Graduate Program of the Freie Universität Berlin, Berlin, Germany. [4]Graduate Program of the Humboldt Universität Berlin, Berlin, Germany. [5]NeuroCure Cluster of Excellence, Charité Universitätsmedizin, Berlin, Germany. ✉e-mail: Jentsch@fmp-berlin.de

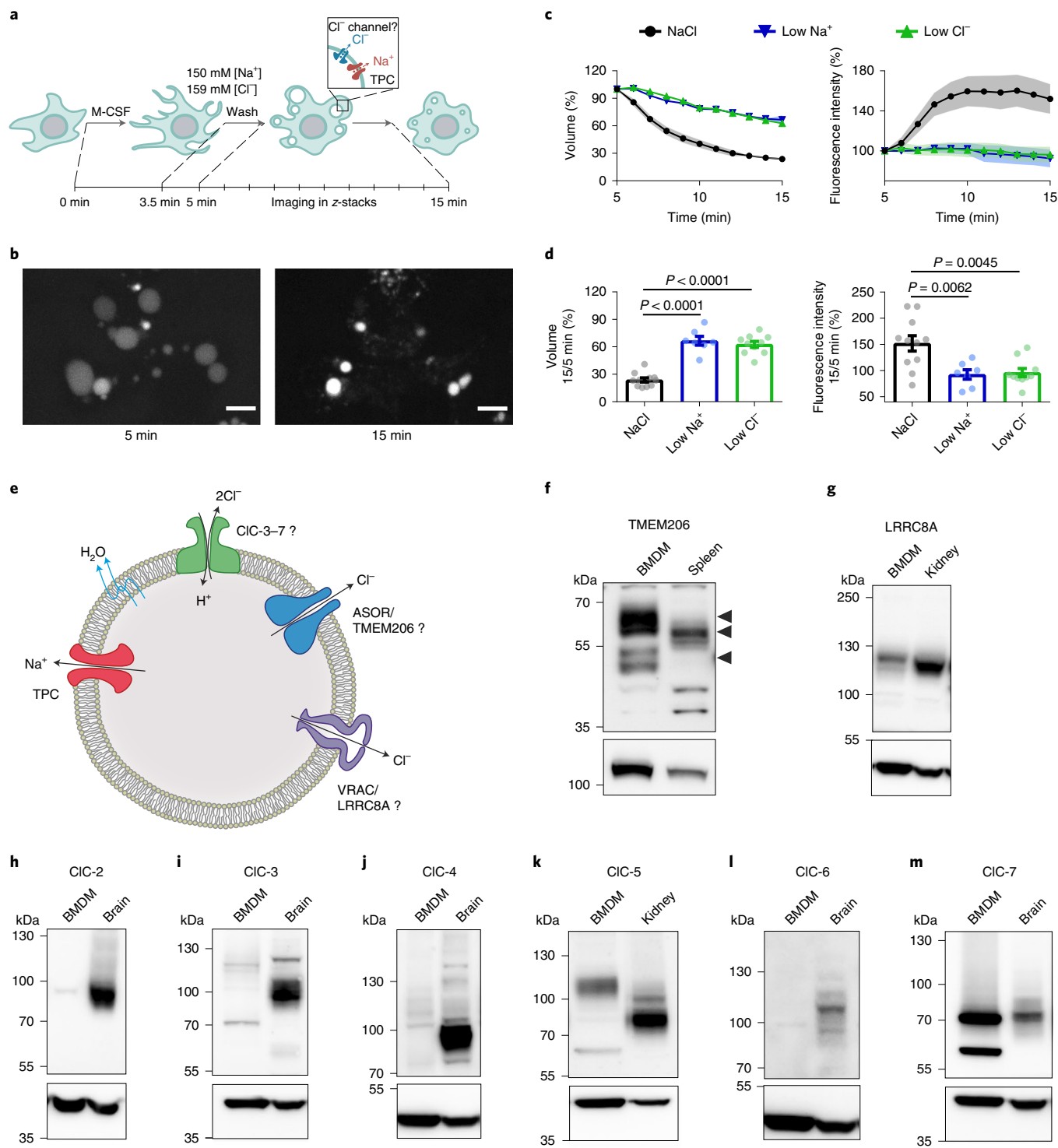

**Fig. 1 | Ions and candidate ion transporters in MP shrinkage. a**, Scheme of experimental approach. **b**, Fluorescence images of MPs formed in presence of 70 kDa TMR–dextran, 5 and 15 min after M-CSF addition. Scale bars, 5 μm. **c**, Removal of luminal Na⁺ or Cl⁻ impairs MP resolution, measured by vesicle volume (left) or TMR-dextran fluorescence (right). $n = 11$, $N = 743$ (WT); $n = 7$, $N = 533$ (low Na⁺, near 0 mM); $n = 10$, $N = 869$ (low Cl⁻, 9 mM). $n$ is number of animals, $N$ is number of MPs. Plot of mean ± standard error of the mean (s.e.m.) (shown as bands), averaging means from individual mice. **d**, Mean MP volume 15 min after M-CSF normalized to volume at 5 min as function of luminal ion concentrations. Data points, mean values from individual mice. Error bars, s.e.m. One-way ANOVA with Tukey's multiple comparison shown with regard to NaCl. **e**, Ion transporter candidates. **f**–**m**, Western blot expression analysis of TMEM206 (specific bands indicated by arrows) (**f**), LRRC8A (**g**), ClC-2 (**h**), ClC-3 (**i**), ClC-4 (**j**), ClC-5 (**k**), ClC-6 (**l**) and ClC-7 (**m**) in mouse BMDMs, compared with organs highly expressing the respective proteins. α1 Na/K-ATPase (**f**) and actin (**g**–**m**) were used as loading control. $n = 3$ for each western blot. Source numerical data and unprocessed blots are available in source data.

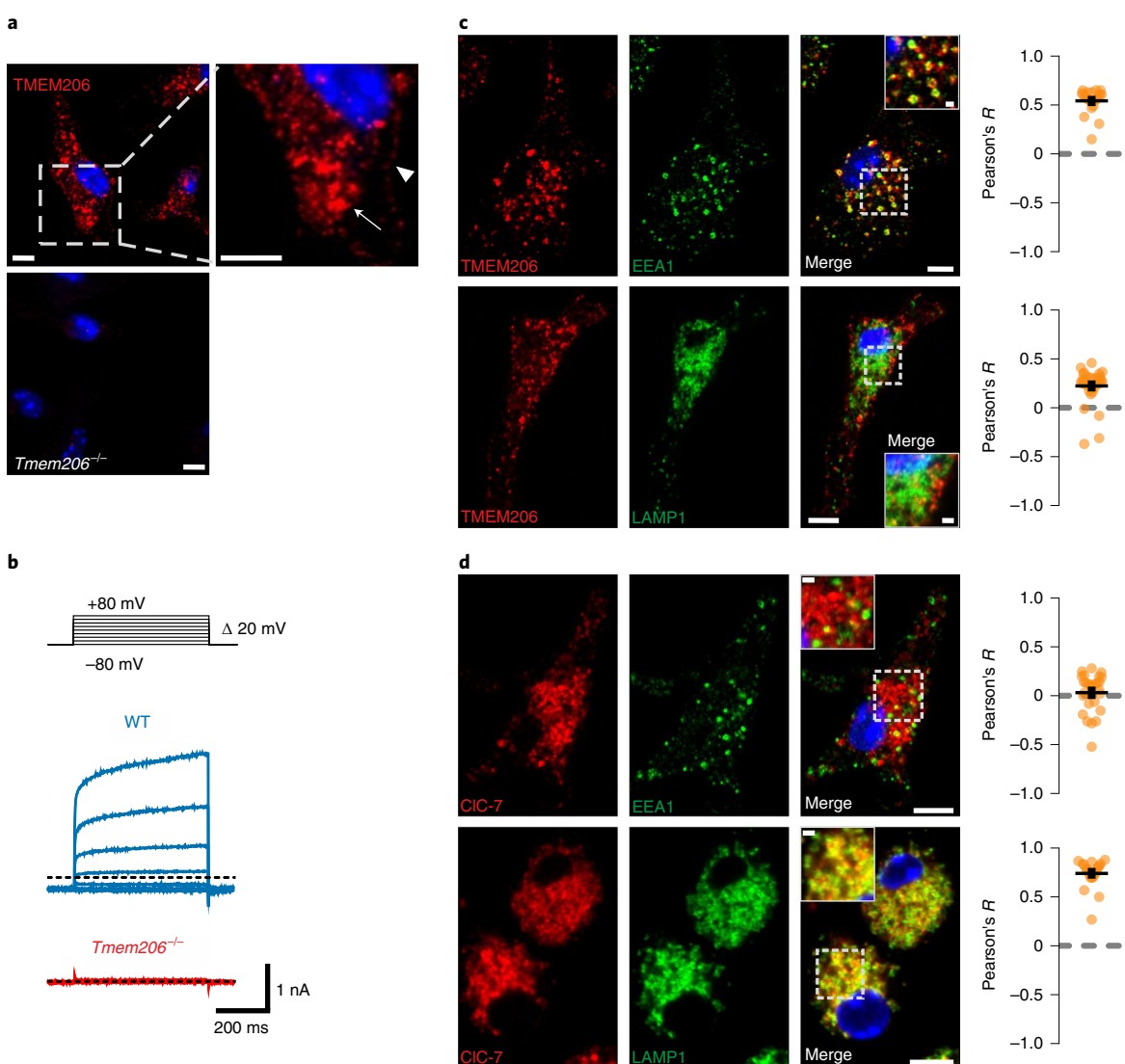

**Fig. 2 | Endogenous expression of TMEM206 and ClC-7 in mouse primary macrophages. a**, Immunofluorescence analysis for TMEM206 in BMDMs. The channel resides on intracellular vesicles (arrow) but also at the plasma membrane (arrowhead). $Tmem206^{-/-}$ cells prove antibody specificity. Scale bars, 5 μm. **b**, Representative traces of acid-activated (pH$_o$ 4.8) anion currents in WT, but not $Tmem206^{-/-}$ BMDMs, obtained with whole-cell patch-clamp recordings. Voltage step protocol shown on top. $N = 8$ cells per genotype. **c**, TMEM206 co-localizes with early endosomal EEA1, but not with lysosome marker LAMP1. Pearson's $R$ calculated from 21 cells (EEA1) and 33 cells (LAMP1) from two independent experiments. **d**, ClC-7 is present in lysosomes (LAMP1) and does not co-localize with early endosome marker EEA1. Pearson's $R$ calculated from 27 cells (EEA1) and 23 cells (LAMP1) from two independent experiments. Mean ± s.e.m. Scale bars, 5 μm and 1 μm for enlargements. Source numerical data are available in source data.

was detected on MPs during the first ~10 min (Fig. 3a–e). Later (Fig. 3f), MPs acquired ClC-7, probably by fusion with lysosomes. ASOR trafficking was followed by live cell imaging of BMDMs transfected with fluorescently tagged TMEM206 (Fig. 3c) that localized similar to the native protein (Extended Data Fig. 2a,b). MPs formed by applying M-CSF together with TMR–dextran showed red luminal fluorescence surrounded by green ASOR-GFP (green fluorescent protein) (Fig. 3c and Supplementary Video 2). They shrank over time and concentrated TMR–dextran in their lumina. Hence, ASOR/TMEM206 emerged as an excellent candidate for the Cl$^-$ channel involved in MP shrinkage.

We collected BMDMs from $Tmem206^{-/-}$ mouse lines (Extended Data Fig. 4a–d), which lacked obvious pathology, and from Cx3cr1-Cre$^{ERT2}$; $Clcn7^{lox/lox}$ mice in which $Clcn7$ was disrupted in the monocyte lineage[22] to avoid the osteopetrosis and early death of $Clcn7^{-/-}$ mice[20]. Whereas $Clcn7$ disruption had no

effect (Extended Data Fig. 5d), ASOR deletion impaired shrinkage to the same degree as luminal Cl$^-$ replacement (Fig. 4a–c and Supplementary Videos 1 and 3). In human HT-1080 cancer cells[23], an 80% reduction of TMEM206 protein levels by short interfering RNA (siRNA) (Extended Data Fig. 4e) sufficed to slow the resolution of MPs that were generated in response to epidermal growth factor (EGF) (Fig. 4d–f). Off-target effects were excluded by using three $Tmem206^{-/-}$ mouse lines generated with different single guide RNAs (sgRNAs) (Extended Data Fig. 4d) and by rescuing MP resolution of $Tmem206^{-/-}$ BMDMs by transfecting TMEM206 (Fig. 4g). Overexpression of TMEM206 failed to significantly accelerate MP resolution beyond wild type (WT) levels in both $Tmem206^{-/-}$ and WT BMDMs. Rescue of resolution depended on ASOR's ion conductance because the transport-deficient K319C mutant[18] had no significant effect (Fig. 4g). Rescue of MP resolution did not require an N-terminal tyrosine motif that enhances endosomal targeting[21]

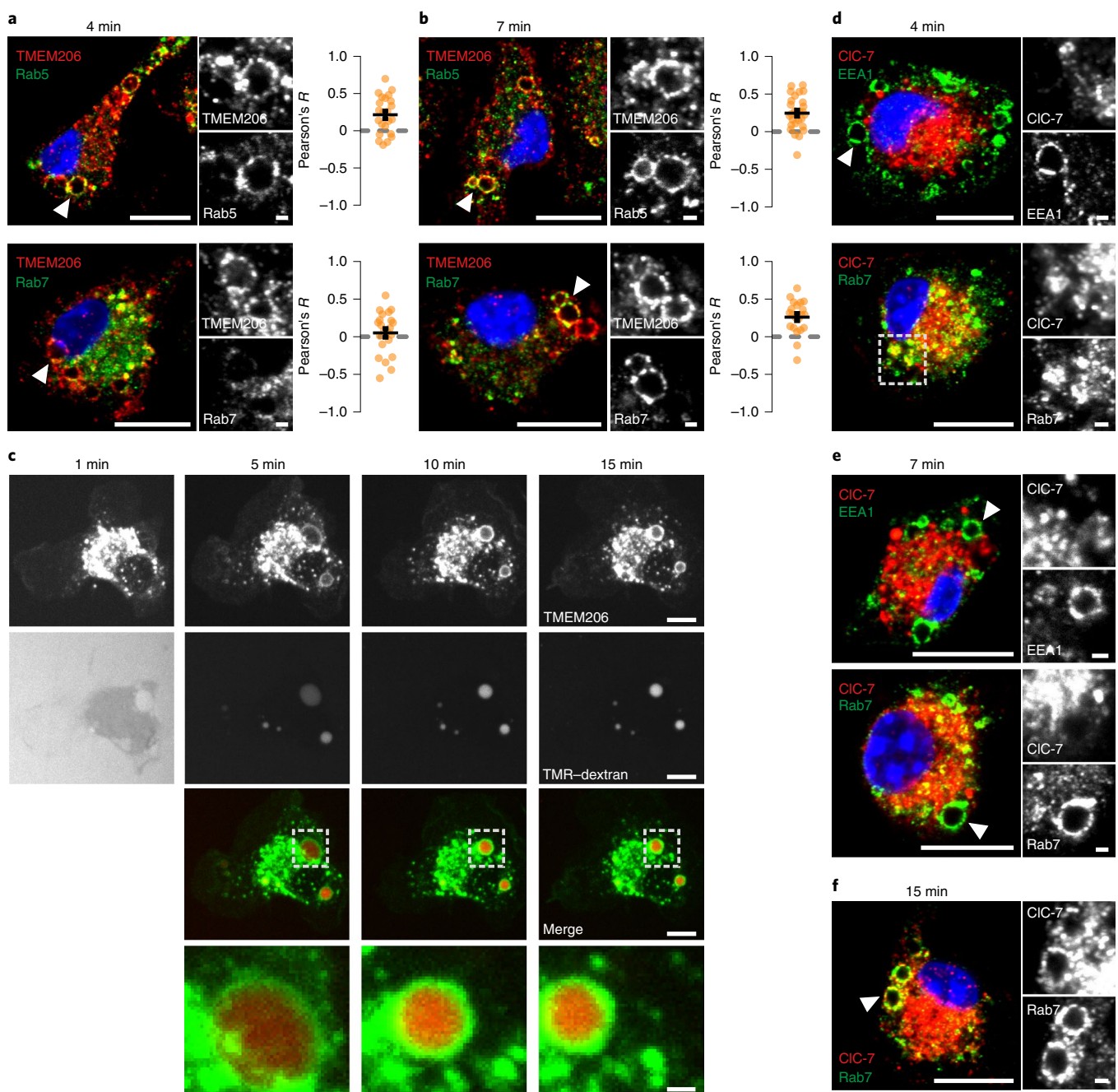

**Fig. 3 | Expression of TMEM206 and ClC-7 on BMDM MPs. a,b**, TMEM206 on MPs 4 min (**a**) and 7 min (**b**) after M-CSF co-localize with rab5 (**a** and **b**) and rab7 (**b**) respectively. Pearson's $R$ calculated from 23 cells (Rab5, 4 min), 27 cells (Rab5, 7 min), 22 cells (Rab7, 4 min) and 21 cells (Rab7, 7 min) from at least two independent experiments. Scale bars, 10 μm and 1 μm for enlargements. **c**, Frames from live cell imaging of TMEM206-GFP transfected BMDMs at various timepoints after addition of M-CSF together with TMR–dextran (Supplementary Video 2). Scale bars, 5 μm and 1 μm for enlargements (bottom). **d–f**, ClC-7 is absent from MPs at 4 min (**d**) or 7 min (**e**) after M-CSF induction, but is on rab7-positive MPs after 15 min (**f**). Areas chosen for magnification indicated by white arrowheads on the left. Rab7 was absent from MPs at 4 min (**d**). Scale bars, 10 μm and 1 μm for enlargements. Source numerical data are available in source data.

(Fig. 4h). Disruption of *Tmem206* or ion replacements did not affect acute, M-CSF-triggered formation of MPs of BMDMs (Extended Data Fig. 6a). This is expected from ASOR's role in luminal salt loss and ensuing osmotic shrinkage, which is only effective after the lumen has lost its connection to the extracellular medium, and from the voltage and pH dependence of ASOR that shuts it down at the plasma membrane. Likewise, ablation of ASOR/TMEM206, which is also found on phagosomes (Extended Data Fig. 6b), did not

impair the uptake of fluorescent beads by phagocytosis (Extended Data Fig. 6c).

We examined the localization of another candidate[6], heteromeric LRRC8/VRAC (ref. [11]) (Fig. 1e,g). Both LRRC8A and LRRC8D subunits localized to the plasma membrane and variably to cytoplasmic puncta in BMDMs from mice expressing epitope-tagged subunits[24,25] (Extended Data Figs. 7a,b and 8a,b) and in transfected HeLa cells (Extended Data Fig. 8d). Contrasting

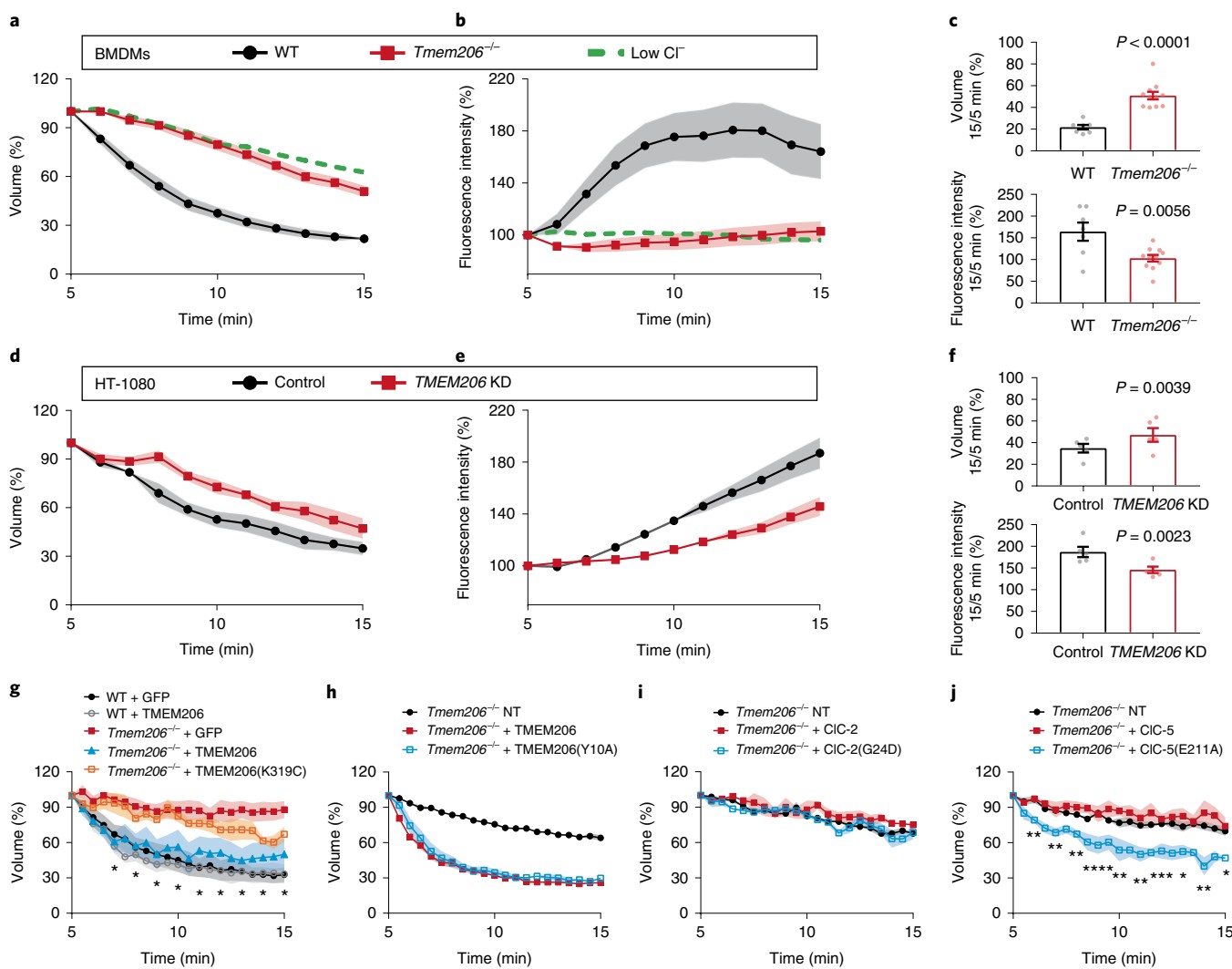

**Fig. 4 | MP resolution impaired by *Tmem206* ablation and rescued by Cl⁻ channels. a–c**, *Tmem206* disruption in BMDMs impairs MP resolution, $n = 7$, $N = 419$ (WT); $n = 11$, $N = 922$ (*Tmem206⁻/⁻*). Impact of Cl⁻ substitution on WT resolution (green dashed line, data from Fig. 1c) shown for comparison. Data included in WT control partially overlap with NaCl control in Fig. 1c. Two-tailed Mann–Whitney test (volume), unpaired two-tailed *t*-test (intensity). **d–f**, KD of *TMEM206* in HT-1080 cancer cells impairs MP resolution, $n = 5$, $N = 37$ (control, transfected with non-targeting siRNA), $N = 38$ (*Tmem206⁻/⁻*). One-sample two-tailed *t*-test comparing ratio (KD and corresponding control) with hypothetical value of 1. **g**, TMEM206, but not dead-pore mutant TMEM206(K319C) (ref. [18]), rescues MP resolution of *Tmem206⁻/⁻* BMDMs. $n = 5$, $N = 37$ (WT + GFP); $n = 5$, $N = 10$ (WT + TMEM206); $n = 4$ (*Tmem206⁻/⁻* + GFP $N = 11$ or +TMEM206 $N = 9$); $n = 5$, $N = 16$ (*Tmem206⁻/⁻* + TMEM206(K319C)); Kruskal–Wallis comparison with Dunn's post-hoc test with regard to *Tmem206⁻/⁻* + GFP done on cells for every minute, significance shown between *Tmem206⁻/⁻* + GFP ($N^* = 8$) and *Tmem206⁻/⁻* + TMEM206 ($N^* = 8$); *Tmem206⁻/⁻* + TMEM206(K319C) ($N^* = 10$) is not significantly different from *Tmem206⁻/⁻* + GFP. **h**, Overexpression of WT and Y10A mutant TMEM206 in *Tmem206⁻/⁻* macrophages similarly enhances MP shrinkage, $n = 4$, $N = 98$ (NT), $N = 31$ (*Tmem206⁻/⁻* + TMEM206), $N = 28$ (*Tmem206⁻/⁻* + TMEM206(Y10A)). **i**, Neither ClC-2 nor 'open' ClC-2(G24D) (ref. [24]) rescues resolution in *Tmem206⁻/⁻* BMDMs. $n = 4$, $N = 43$ (NT); $n = 4$ (*Tmem206⁻/⁻* + ClC-2 $N = 13$ or +ClC-2(G24D) $N = 12$). **j**, ClC-5(E211A) uncoupled mutant[27,28], but not WT ClC-5, rescues *Tmem206⁻/⁻* MP resolution. $n = 3$, $N = 8$ *Tmem206⁻/⁻* + ClC-5, $n = 4$, $N = 9$ +ClC-5(E211A), $n = 4$, $N = 40$ NT); one-way ANOVA with Dunnett's post-hoc test (with regard to NT) done on cells for every minute. Significance shown between *Tmem206⁻/⁻* (NT) ($N^* = 20$) and *Tmem206⁻/⁻* + ClC-5(E211A) ($N^* = 8$). *$P \leq 0.05$, **$P \leq 0.01$, ***$P \leq 0.001$ and ****$P \leq 0.0001$. All plots present mean ± s.e.m. (shown as bands). $N^*$ is number of cells. Source numerical data are available in source data.

with a recent report[14], these puncta did not co-localize with lysosomes. Whereas LRRC8D was not found on MPs (Extended Data Fig. 8c), LRRC8A was detected on MPs in less than 10% of the cells 4 min, but not 7 min, after M-CSF stimulation (Extended Data Fig. 7c). Disrupting the essential LRRC8A subunit[11,12] had no significant effect on MP resolution (Extended Data Fig. 5e). Small effects, however, might have been missed as *Lrrc8a* expression was reduced by only ~60% in Cx3cr1-Cre^ERT2; *Lrrc8a*^lox/lox macrophages (Extended Data Fig. 5f).

We asked whether other Cl⁻ conductors might replace ASOR in MP resolution. Overexpression of ClC-2, whose disruption did not affect MP resolution (Extended Data Fig. 5a), failed to rescue resolution (Fig. 4i). This failure might be due to its voltage dependence[8], which is opposite to that of ASOR[15] and predicts low currents in lumen-negative MPs. However, transfection of the aldosteronism-causing ClC-2(G24D) mutant[24], which yields large linear Cl⁻ currents, had no effect either (Fig. 4i). ClC-2, a plasma membrane channel, may have been excluded from MPs. We therefore

used the E211A ClC-5[unc] uncoupled, channel-like mutant of the early endosomal anion–proton exchanger ClC-5 (ref. [25]), which mediates a pure, voltage-independent Cl− conductance[26–28]. Unlike WT ClC-5, ClC-5[unc] rescued resolution of *Tmem206−/−* MPs (Fig. 4j). Hence, ASOR can be replaced by another appropriately targeted Cl− conductance, but not by a 2Cl−/H+ exchanger.

**Role of luminal acidification.** ASOR/TMEM206 needs both cytoplasmic positive voltages and extracellular (or luminal) acidity[15,18] for activity. In the absence of other conductances, macropinosomal Na+ channels together with an inside-out tenfold Na+ gradient would drive the luminal potential (referred to cytoplasm) to −60 mV, a value amply sufficient for ASOR activity[15,29]. Agreeing with previous work[30], 10 min after M-CSF stimulation, luminal pH reached ~5.5 (Fig. 5a). This value is close to pH$_{1/2}$ of ASOR (~5.3) (refs. [15–18]). If ASOR retains its pH dependence in MPs, luminal alkalinization should inhibit MP shrinkage. Indeed, extracellular NH$_4$Cl, which alkalinizes cellular compartments, abolished vesicle resolution (Fig. 5b,c).

V-type H+-ATPases acidify diverse intracellular organelles[31]. In macrophages, which express proton pumps at the plasma membrane[32], they might reach MPs directly from the outer membrane and/or by vesicular fusion. The lysosomal a3 proton pump subunit[31], against which good antibodies are available[33], reached MPs when they acquired ClC-7 and rab7 (Extended Data Fig. 9). Proton pump inhibition with bafilomycin A partially alkalinized MPs already at the beginning of the measurement (Fig. 5d) and partially inhibited MP resolution (Fig. 5f,g).

Both luminal Cl− removal and disruption of ASOR/TMEM206 led to a marked luminal acidification (Fig. 5a). Both procedures should increase the driving force for electrogenic H+ uptake by making the lumen more negative: in WT, low luminal Cl− causes ASOR-mediated influx of negatively charged Cl−, whereas ASOR knockout (KO) prevents the efflux of Cl− driven by lumen-to-cytosol Cl− gradients. The predicted negative shift in luminal potential might enhance H+-ATPase activity. However, since ASOR KO induced acidification also in the presence of bafilomycin (Fig. 5d), there must be additional electrogenic acidification mechanisms.

Attractive candidates are CLC 2Cl−/H+ exchangers that couple vesicular H+ uptake to Cl− efflux. The lumen-negative voltage predicted for *Tmem206−/−* MPs would not only increase the electrochemical potential for H+ entry and Cl− efflux, but also activate CLC exchangers that open at cytoplasmic potentials > +20 mV (ref. [8]). To test for CLC anion/proton exchange, we imposed opposite Cl− gradients on *Tmem206−/−* MPs and measured luminal pH. Assuming that without ASOR most, if not all, Cl− crosses the membrane through CLC 2Cl−/H+ exchangers, we predicted that lumen-to-cytosol and cytosol-to-lumen Cl− gradients will accumulate or deplete, respectively, luminal H+. Indeed, luminal pH was more acidic with high than with low luminal Cl− concentration, and this effect was independent of proton pump activity (Fig. 5e). Whereas luminal Cl− removal alkalinized *Tmem206−/−* MPs, it acidified WT MPs (Fig. 5a). This surprising contrast likely results from a large difference in MP potential, generated by the ASOR Cl− conductance, between high and low luminal Cl− conditions (Fig. 5i and Supplementary Note Fig. 3a,b). The resulting lumen-negative voltage in the presence of ASOR provides a favourable driving force for electrogenic H+ influx with low luminal Cl−.

Having demonstrated macropinosomal chloride/proton exchange activity (Fig. 5e), we sought the underlying CLC exchanger. We had already excluded late endosomal/lysosomal ClC-7 (Figs. 2 and 3 and Extended Data Fig. 5d). The predominantly neuronal ClC-6 (ref. [34]) was almost absent from BMDMs (Fig. 1l). ClC-3 and ClC-4 were barely detectable in BMDMs (Fig. 1i,j), reside in late endosomes[35,36] and did not affect MP resolution (Extended Data Fig. 5). By contrast, ClC-5 is robustly expressed in macrophages (Fig. 1k).

where it co-localized with early endosomal markers (Fig. 6b) as in other cells[25,37]. Moreover, the rescue of *Tmem206−/−* MP resolution by ClC-5[unc] (Fig. 4j) suggested that ClC-5 reaches MPs at least when overexpressed. Upon M-CSF stimulation, native ClC-5 was indeed detected on large MPs, where it co-localized with ASOR/TMEM206 (Fig. 5j,k), rab5 and later rab7 (Fig. 6c,d). ClC-5 might facilitate MP resolution directly by mediating Cl− efflux and indirectly by activating ASOR through Cl−-gradient-driven acidification. However, *Clcn5* disruption failed to significantly decrease MP resolution even when H+-ATPases were inhibited (Fig. 5f,g and Extended Data Fig. 5c) and no effect of *Clcn5* disruption on luminal pH was observed (Fig. 5h and Extended Data Fig. 10c), suggesting further redundancies in acidification mechanisms. Theoretically, even an H+ conductance may acidify vacuoles by up to one pH unit when they are depolarized by TPCs and the tenfold lumen-to-cytosol Na+ gradient. However, a sizeable parallel Cl− conductance (embodied by ASOR) would render the lumen more positive and reduce the acidification achievable by an H+ conductance.

**Model for MP resolution.** Our results suggest a model for MP resolution (Fig. 7a,b) that incorporates as main players previously implicated TPC channels[5], newly identified ASOR/TMEM206, ClC-5 and H+-ATPases, in addition to other H+ transporters such as an H+ conductance and a high water permeability. Both ClC-5 and ASOR are strongly outwardly rectifying and thus virtually inactive at physiological plasma membrane voltages. ASOR shows higher plasma membrane expression than ClC-5 (ref. [8]) and might be directly incorporated into MPs upon their formation even when lacking an N-terminal endosomal targeting motif[21]. Both ClC-5 and ASOR probably reach MPs also by fusion with endosomes. MP resolution is triggered by Na+ efflux through newly inserted endosomal TPCs[5] that changes the voltage from lumen positive to lumen negative. This depolarization directly activates ClC-5 and potentially other CLCs. It is also permissive for ASOR activity, which, however, additionally needs luminal acidification. The lumen-negative voltage also increases the electrochemical driving force for CLC-mediated Cl− efflux and H+ uptake. Together with TPC-mediated Na+ efflux, CLC-mediated efflux of Cl− probably contributes to MP shrinkage, aided by the fact that counter-transported H+ is osmotically inactive as it binds to buffers. A direct contribution of ClC-5 to resolution, however, is much smaller than that of ASOR, as evident from our failure to observe an effect of *Clcn5* disruption on MP resolution (Extended Data Fig. 5c). A more important effect of ClC-5 may be a contribution to luminal acidification, which in turn activates ASOR. Na+ efflux through TPCs, and electrically coupled Cl− efflux through ASOR (with a minor contribution of CLCs), causes luminal loss of Na+ and Cl− and osmotic shrinkage. This shrinkage, in turn, prevents a rapid decline of luminal Na+ and Cl− concentrations, maintaining appropriate gradients for further shrinkage, which gradually decrease with increasing luminal concentrations of non-transported osmoticants.

The feasibility of this transport scheme was tested in a reductionist mathematical model (Supplementary Note and Supplementary Note Figs. 1–4). It considers the voltage and pH dependencies of ASOR and CLCs (Supplementary Note Fig. 2), assumes infinite water permeability and neglects insertion or removal of transporters over time. The model semi-quantitatively predicts MP shrinkage, suggests a (rather minor) role of CLC exchangers during early phases when the less acidic luminal pH favours transport through CLCs rather than ASOR (Supplementary Note Fig. 3a). Later CLCs will operate close to electrochemical equilibrium and are partially inhibited by luminal acidification. The model explains the opposite effects of luminal Cl− removal on luminal pH in the presence (Fig. 5i and Supplementary Note Figs. 1b,d and 3a) and absence of ASOR (Fig. 5i and Supplementary Note Figs. 1c and 3b–d). This difference mainly results from changes in membrane potential.

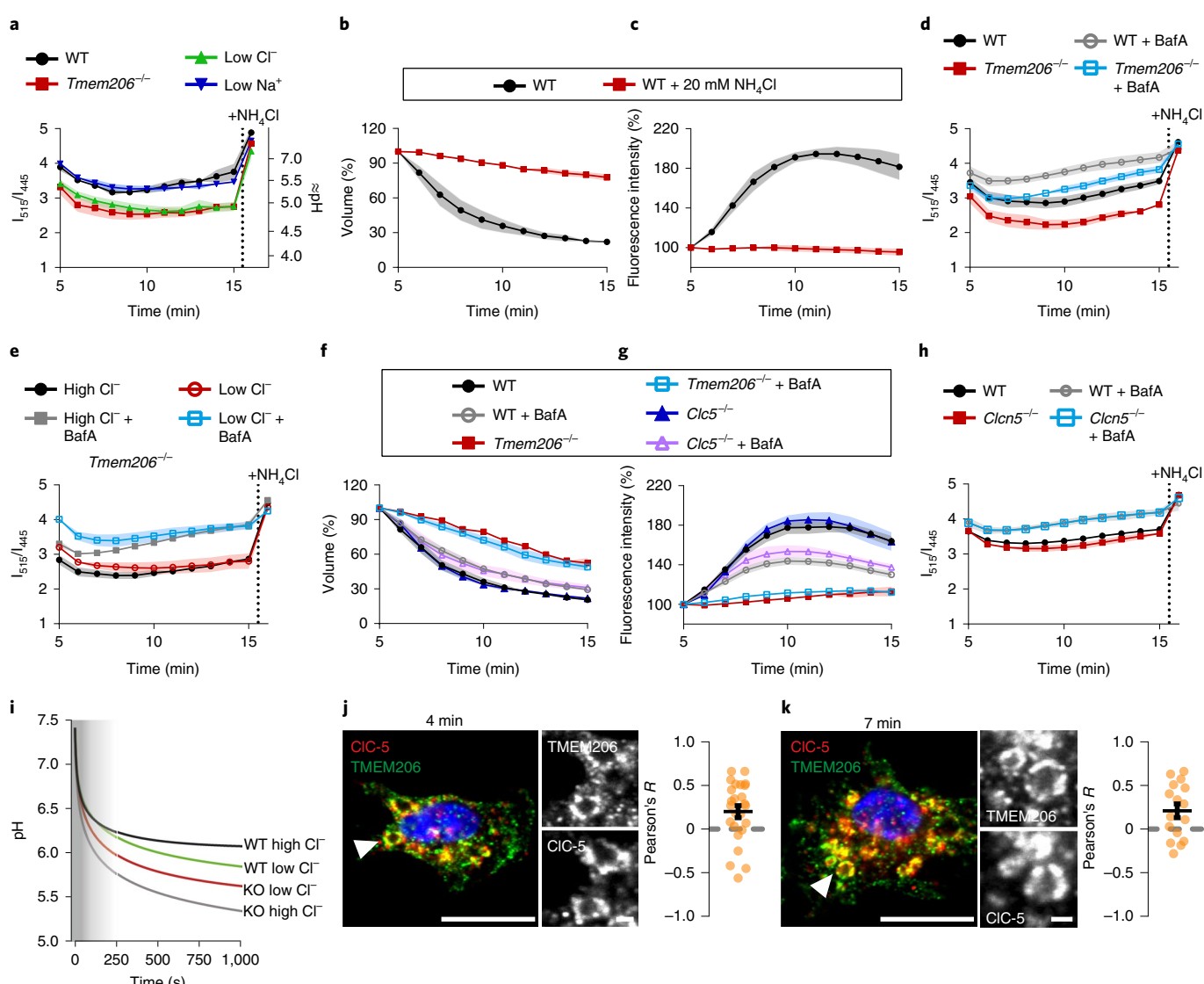

**Fig. 5 | Roles of V-type ATPase and CLC 2Cl⁻/H⁺ exchangers in luminal pH and resolution of MP. a**, Disruption of *Tmem206* or low luminal Cl⁻ acidify MP lumina. $n = 3$, $N = 81$ (WT); $n = 7$, $N = 142$ (*Tmem206⁻/⁻*); $n = 4$, $N = 150$ (low Cl⁻); $n = 5$, $N = 198$ (low Na⁺). **b,c**, Addition of 20 mM NH₄Cl abolishes MP shrinkage, $n = 4$ ($N = 419$ WT, $N = 749$ WT + NH₄Cl). **d**, Bafilomycin (BafA) alkalinizes MPs in both WT and *Tmem206⁻/⁻* BMDMs. $n = 4$ (WT alone $N = 175$ or +BafA $N = 188$); $n = 3$, $N = 117$ (*Tmem206⁻/⁻*); $n = 4$, $N = 194$ (*Tmem206⁻/⁻* + BafA). **e**, pH changes upon lowering luminal Cl from 159 mM to 9 mM in *Tmem206⁻/⁻* BMDMs indicate Cl⁻/H⁺ exchange ($n = 3$; high Cl⁻ alone $N = 287$, +BafA $N = 429$, low Cl⁻ alone $N = 298$, +BafA $N = 369$). **f,g**, BafA decreases MP shrinkage in WT and *Clcn5⁻/⁻*, but not *Tmem206⁻/⁻* BMDMs $n = 7$, $N = 456$ (WT); $n = 8$, $N = 468$ (WT + BafA); $n = 3$ (*Tmem206⁻/⁻* alone $N = 359$ or +BafA $N = 278$); $n = 5$ (*Clcn5⁻/⁻* alone $N = 421$ or + BafA $N = 434$). **h**, BafA alkalinizes WT and *Clcn5⁻/⁻* MPs. $n = 4$ (WT alone $N = 480$ or +BafA $N = 384$); $n = 6$ (*Clcn5⁻/⁻* alone $N = 433$ or +BafA $N = 480$). **i**, Reductionist mathematical model (Supplementary Note) semi-quantitatively explains results shown in **a** and **e**. Overlay from pH panels of Supplementary Note Fig. 3a,b. Shaded area, early timepoints that could not be determined experimentally. **j,k**, TMEM206 and ClC-5 are co-expressed on same MPs, at both 4 and 7 min after M-CSF. Small panels, magnified MPs indicated by arrowheads. Pearson's *R* calculated from 25 cells (4 min), 16 cells (7 min) from three (4 min) and two (7 min) independent experiments. All plots present mean ± s.e.m. (shown as bands) averaging from individual mice. Scale bars, 10 μm and 1 μm for the magnification panels. In **a**, **d**, **e** and **h**, 20 mM NH₄Cl was added to show dye responsiveness to alkalinization (bleaching control). Source numerical data are available in source data.

The model also shows that CLCs, without ASOR but together with TPCs, can in principle support vesicle shrinkage with or without additional acidifying transporters (Supplementary Note Fig. 3d), albeit less efficiently than ASOR (Supplementary Note Fig. 3a). It also predicts that resolution is largely independent of the particular combination of acidifying transporters (CLCs, H⁺-ATPase and H⁺ conductance) (Supplementary Note Fig. 3a, e-j). Even a model vesicle containing only an H⁺ conductance together with TPC and ASOR showed marked shrinkage, but at a slightly lower rate owing to reduced luminal acidification (Supplementary Note Fig. 3j). A hint for such a conductance comes from only partially impaired MP shrinkage in bafilomycin-treated *Clcn5⁻/⁻* BMDMs (Fig. 5f,g) and the acidification upon luminal Cl⁻ removal (Fig. 5a), which cannot be caused by anion/proton exchangers (Supplementary Note Fig. 3h). However, it can be explained by V-type-ATPases or an H⁺ conductance (Supplementary Note Fig. 3i,j) that respond with increased H⁺ transport to more negative luminal potentials. The H⁺ conductance might be embodied by the Hv1 H⁺-channel[38,39], which, reminiscent of ASOR and ClC-5, is strongly outwardly rectifying.

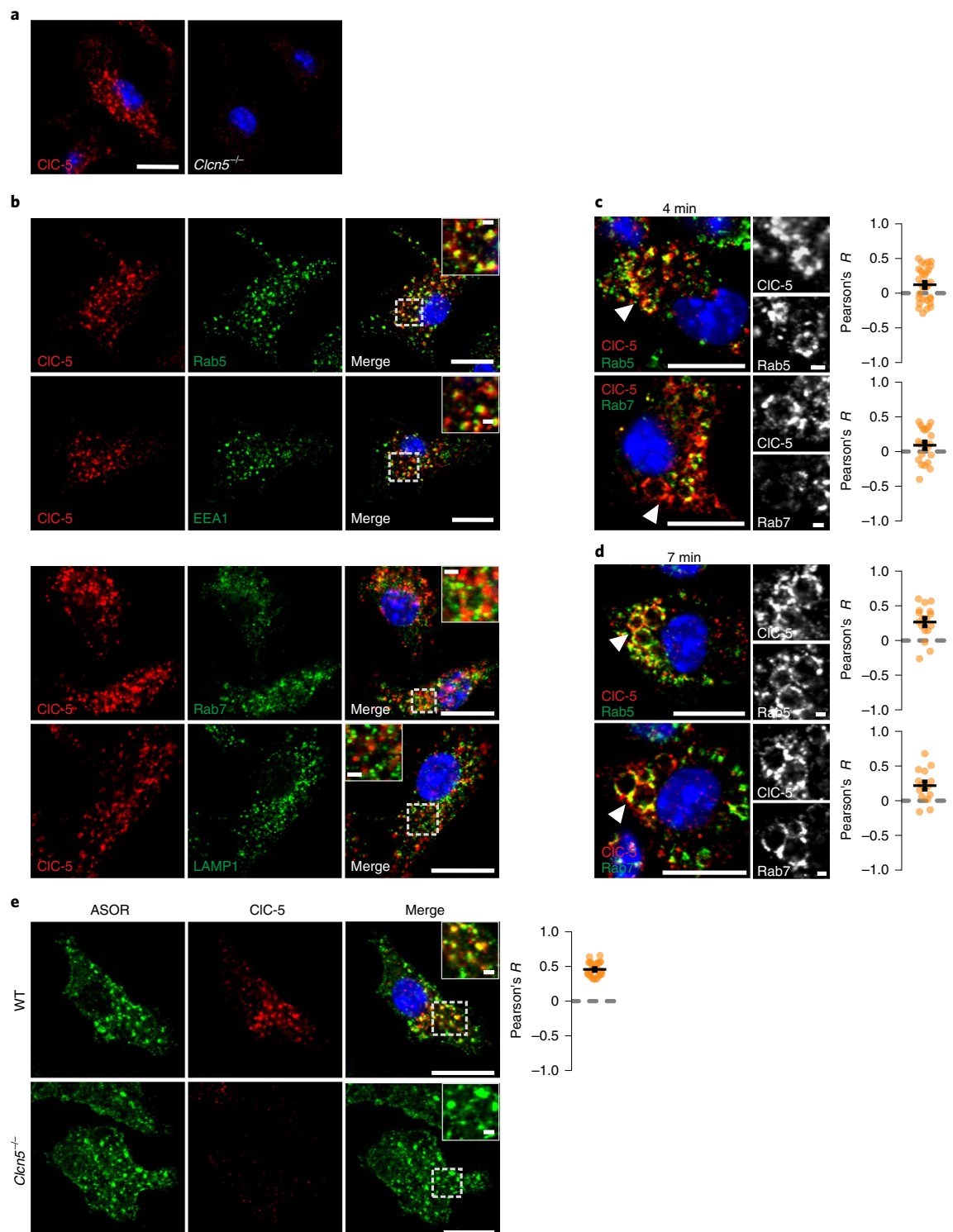

**Fig. 6 | ClC-5 co-localizes with TMEM206 in macrophages endosomes and MPs. a**, ClC-5 immunostaining in WT and *Clcn5*−/− (control) BMDMs. **b**, ClC-5 was co-localized with endolysosomal markers rab5 and EEA1, partially with late endosomal marker rab7 but not with lysosomal LAMP1. **c,d**, ClC-5 is present on MPs labelled by rab5 and rab7 stainings, at 4 min (**c**) and 7 min (**d**) after M-CSF addition. Pearson's *R* calculated from 31 cells (Rab5, 4 min), 17 cells (Rab5, 7 min), 21 cells (Rab7, 4 min) and 15 cells (Rab7, 7 min). **e**, TMEM206 (green) and ClC-5 (red) are present on the same vesicles in BMDMs. Absence of ClC-5 antibody labelling in *Clcn5*−/− BMDMs confirms the specificity of staining. Pearson's *R* calculated from 29 cells from two (Rab5 and ClC-5/TMEM206) and one (Rab7) independent experiments. Mean ± s.e.m. Scale bars, 10 μm and 1 μm for the magnification panels. Source numerical data are available in source data.

The relative insensitivity of volume changes to acidification mechanisms may result from feedback loops that control ASOR and CLC activity. ASOR Cl− currents oppose the depolarizing, TPC-mediated Na+ currents, resulting in a hyperpolarization that directly inhibits ASOR in a negative feedback. This hyperpolarization also reduces the electrochemical driving force for H+ entry and inhibits the activity of CLC exchangers, which show a voltage dependence similar to ASOR. Both effects result in

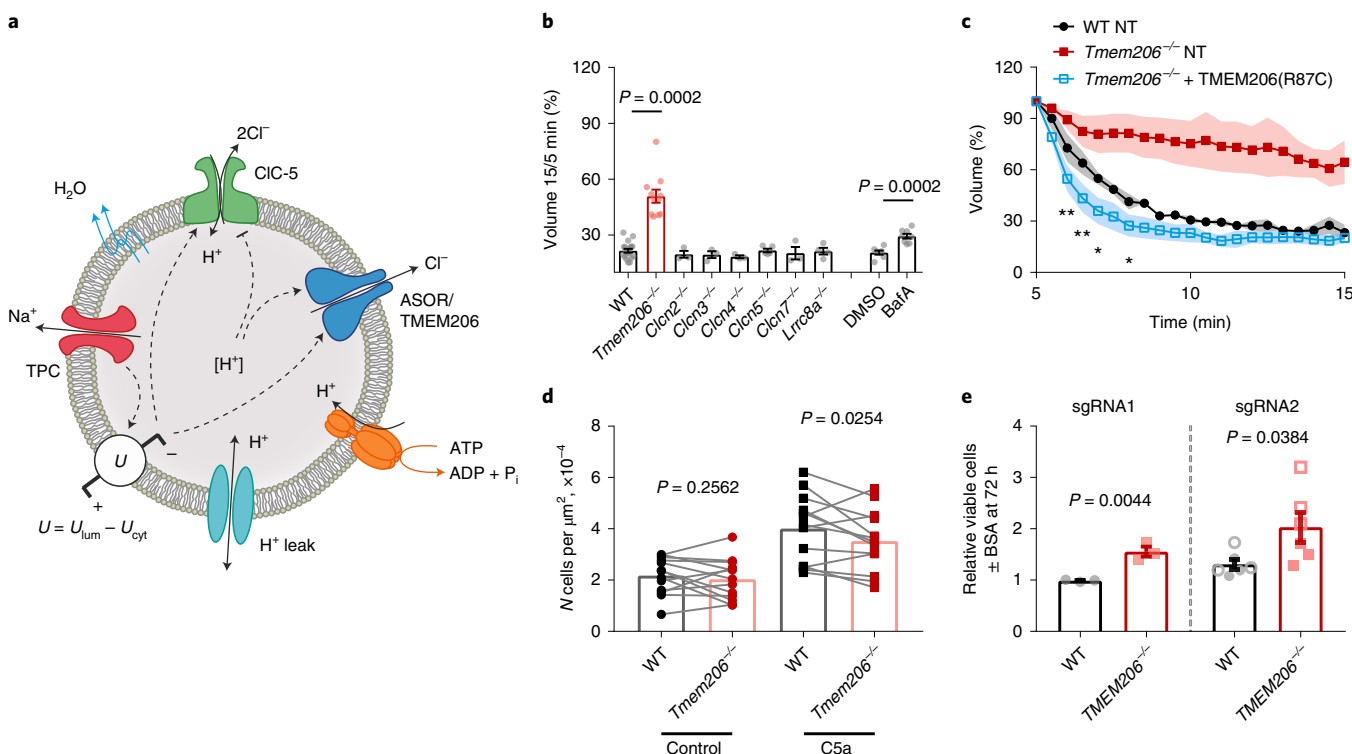

**Fig. 7 | Physiological role of ASOR. a**, Model for roles of TPC, ClC-5, ASOR and V-ATPase in MP shrinkage. The voltage $U$ is defined as difference between the luminal and cytoplasmic electrical potentials ($U_{lum}$ and $U_{cyt}$ respectively). **b**, Effects of anion transporter disruption on MP shrinkage. $n = 19$ (WT); $n = 11$ ($Tmem206^{-/-}$); $n = 3$ ($Clcn2^{-/-}$, $Clcn3^{-/-}$ and $Clcn4^{-/-}$); $n = 5$ ($Clcn5^{-/-}$); $n = 3$ ($Clcn7^{-/-}$); $n = 4$ ($Lrrc8a^{-/-}$), $n = 7$ (WT + DMSO); $n = 8$ (WT + BafA). Kruskal–Wallis (Dunn's post hoc) test for all KO versus WT (includes WT controls to all corresponding KO in the same dataset) with Kruskal–Wallis test (Dunn's normalization for multiple comparison). Unpaired two-tailed $t$-test for DMSO versus BafA. **c**, Transfection of TMEM206(R87C) in $Tmem206^{-/-}$ BMDMs accelerated MP shrinkage compared with NT cells. One-way ANOVA with Dunnett's post-hoc test comparing with WT NT. Significance shown for 14 ($Tmem206^{-/-}$ + TMEM206(R87C)) and 9 (WT NT) cells from $n = 3$ mice each ($N = 22$ WT NT, $N = 26$ $Tmem206^{-/-}$ NT, $N = 17$ $Tmem206^{-/-}$ + TMEM206(R87C) MPs). *$P \leq 0.05$, **$P \leq 0.01$. **d**, $Tmem206^{-/-}$ macrophages migrate slower than WT in a scratch assay in presence of C5a. Data obtained 12 h after scratch. $n = 13$, one sample two-tailed $t$-test comparing ratio (WT to corresponding KO) with hypothetical value of 1. Lines connect values obtained from WT and KO BMDMs that were prepared, handled and imaged in parallel. **e**, *TMEM206* disruption enhances the positive effect of 3% BSA on viability of MIA PaCa-2 cancer cells in amino-acid-depleted medium measured after 72 h. $n = 3$, cells generated with two sgRNAs; two independent cell lines (open and filled symbols) for sgRNA2, unpaired two-tailed $t$-test. For all plots, mean ± s.e.m. is shown. Source numerical data are available in source data.

luminal alkalinization that additionally inhibits the strongly pH-dependent ASOR (Supplementary Note Fig. 4). The combination of both voltage- and pH-mediated negative feedback loops predicts that ASOR currents, and hence MP resolution, should be resilient to ASOR expression levels beyond a certain threshold (Supplementary Note Fig. 4).

Agreeing with this prediction, ASOR overexpression in WT or $Tmem206^{-/-}$ macrophages failed to significantly enhance MP resolution beyond WT rates (Fig. 4g). The model predicts that an ASOR mutant with alkaline-shifted pH dependence leads to faster MP resolution by loosening the 'pH brake' on ASOR activity (Supplementary Note Fig. 4a). We resorted to the R87C mutant that yields notable currents already at pH 7.4 and displays WT-like rectification[18]. Transfecting this mutant into $Tmem206^{-/-}$ cells (to avoid mutant/WT heteromers of the trimeric channel[40,41]) indeed accelerated vesicle resolution compared with WT (Fig. 7c). This is not only consistent with our model, but also excludes that the failure of WT ASOR overexpression to accelerate resolution is due to a rate-limiting magnitude of Na+ (TPC) currents.

Importantly, if both ASOR and CLCs lacked their steep voltage and pH dependencies, they might dominate over the TPC-mediated Na+ conductance and clamp luminal pH to near-neutral or even alkaline values (Supplementary Note Fig. 1a,b), which may interfere

with many processes that depend on acidic luminal pH. Collectively, our work suggests an intricate coupling between TPCs, ASOR and CLCs that involves several feedback loops that are enabled by the strong modulation of ASOR and CLCs by voltage and pH, conserved properties whose 'purpose' has remained enigmatic.

Osmotic shrinkage of MPs facilitates budding of vesicles through the formation of tubular extensions that bind BAR-domain proteins[5,42]. These vesicles are needed to recycle plasma membrane proteins back to the cell surface. Recycling is especially important for macropinocytosis because it internalizes large chunks of plasma membrane. Hence, impaired MP shrinkage may lower the abundance of surface receptors[5]. Hypothesizing that this includes the receptor for the C5a complement, which stimulates macrophage migration[43], we compared migration of WT and $Tmem260^{-/-}$ BMDMs in the absence and presence of C5a. Consistent with our hypothesis, migration of $Tmem206^{-/-}$ BMDMs was mildly reduced compared with WT in the presence, but not in the absence, of C5a (Fig. 7d).

Macropinocytosis of extracellular proteins is used by many cancer cells to supply amino acids under nutrient starvation[4,44,45]. This process has often been studied in MIA PaCa-2 human pancreatic cancer cells[44], but operates also in other tumours displaying *RAS* mutations[4,45] that stimulate constitutive macropinocytosis[4].

Using clustered regularly interspaced short palindromic repeats (CRISPR)–Cas9 genomic editing, we generated several independent *TMEM206*[−/−] and control edited *TMEM206*[+/+] MIA PaCa-2 lines and assayed their survival in amino-acid-depleted medium with or without 3% bovine serum albumin (BSA) (Fig. 7e). In all cases, addition of BSA increased the number of *TMEM206*[−/−] cells relative to *TMEM206*[+/+] control cells. These results are consistent with unchanged rate of MP formation in *TMEM206*[−/−] cells (Extended Data Fig. 6a) in conjunction with impaired recycling of internalized albumin and increased MP acidification (Fig. 5a,i). The slow resolution of KO MPs may give them more time to fuse with degradative lysosomes while still containing substantial amounts of albumin and being over-acidified, leading to an increased cellular supply of amino acids. Decreased ASOR activity might hence contribute to the growth of certain cancers. Indeed, The Cancer Genome Atlas database (https://www.proteinatlas.org/) suggests that low *TMEM206* expression weakly correlates with decreased survival of patients with pancreatic cancer that displays a high prevalence of *KRAS* mutations[46].

## Discussion

Previous studies focused on plasma-membrane-resident ASOR and its detrimental role in acid-induced cell death and stroke[17,18,47,48]. It now emerges that its main physiological function is in the endocytic pathway. In addition to MPs, ASOR/TMEM206 is found on endosomes[21]. Similar to our results for MPs, endosomes were more acidic in *Tmem206*[−/−] HEK cells[21]. Transferrin endocytosis was increased in *Tmem206*[−/−] cells owing to increased transferrin receptor recycling[21], arguing against the hypothesis that ASOR-mediated endosome shrinkage increases endocytic trafficking. An important difference between MPs and endosomes is their initial luminal ion composition. Endosomal Cl[−] concentration is initially low (~15 mM) in several cell types and progressively increases over time in parallel to acidification[49,50]. This increase depends on ClC-5 (ref. [49]) and ClC-3 (ref. [50]) for early and late endosomes, respectively. The low initial Cl[−] concentration was attributed to negative surface charges that can decrease luminal anion concentrations in small endosomes, but not in vastly larger MPs. Unlike MPs, endosomes may accumulate Cl[−] into their lumen through both CLCs[49,50] and ASOR in a process driven by proton pumps. Model calculations[51] and analysis of mice expressing mutant CLCs mutated into uncoupled Cl[−] conductors[26,51] show that both Cl[−] channels and 2Cl[−]/H[+] exchangers support proton-pump-driven acidification and luminal Cl[−] accumulation, with exchangers being more efficient than channels. The described effect[21] of ASOR KO on endosomal pH and Cl[−] may reflect the higher efficiency of CLCs versus Cl[−] channels in active acidification[51] rather than luminal Cl[−] loss. Once endosomes have acquired sufficiently high luminal Cl[−] concentration, they might use ASOR and TPCs for shrinkage and vesicle budding as found here with MPs.

In conclusion, acid-activated ASOR/TMEM206 Cl[−] channels, together with luminal acidification and previously identified TPCs[5], are crucial for MP shrinkage (Fig. 7a). MPs express additional, minor Cl[−] exit pathways such as ClC-5 (Fig. 5j,k and Supplementary Note Fig. 3d) because they shrink, albeit at a much slower rate, also in the absence of ASOR. ASOR operates in vesicle-intrinsic feedback loops that render MP pH and resolution remarkably resilient towards variations in transporter copy numbers during membrane remodelling by vesicle fusion and budding, processes that are difficult to balance with local variations of vesicle densities. Our work provides a comprehensive framework for understanding an important aspect of macropinocytosis, an uptake mechanism that is not only crucial for immune cells, but also for nutrient uptake in cancer cells[44,45]. Ion concentration- and voltage-based feedback loops as those described here may play roles in other vesicular compartments.

## Online content

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

## Methods

All animal experiments, including the generation of new mouse lines, were approved by Berlin authorities (LAGeSo). Animals were housed under standard conditions (12 h/12 h dark–light cycle with food and water ad libitum) in the MDC animal facility according to institutional guidelines. In all experiments, 8- to 22-week-old (8–16 weeks old for BMDM preparation) C57Bl/6 mice (*Mus musculus*) of both sexes were used.

**Generation of *Tmem206* KO mice.** *Tmem206*[−/−] mice were generated using CRISPR–Cas9 genome-editing technique. Different combinations of guide RNAs (g1, g2, g5 or g6) were injected by pronuclear microinjection in fertilized C57BL/6 J mouse oocytes that were implanted into foster mothers by the MDC Transgenic Core Facility. g1 (GGTGGTTGAGAACTCGGAGC) and g5 (CAAGGTCCAGGTGTCGTGCC), both targeting exon 2 that codes for the TMEM206 N terminus, were injected together, which led to a frame-shifting 62 bp deletion, as determined by Sanger sequencing of PCR bands. Similarly, either g1 or g2 (GCTCAGCGAGGAGTTGGAGC; also targeting exon 2) were injected together with g6 (CCGTGATGGTCTGGTAGACC; targeting exon 3), which in both cases led to slightly different deletions of ~10 kbp. These injections led to the generation of three different TMEM206 KO mouse lines. Absence of protein expression was validated by western blot using custom-made anti-TMEM206 antibodies.

**Generation of TMEM206 antibodies.** Polyclonal antibodies against TMEM206 were raised in rabbits against the peptide VKTKEEDGREAVEFRQET corresponding to a sequence in the extracellular loop of mouse TMEM206 (T6 LB-3), and to IKIRKRYLKRRGQATNHIS corresponding to a sequence in the C-terminus (T6 CT-1) (Pineda Antibody Service, Berlin). Both peptides were coupled to KLH via N-terminally added cysteines. Sera were affinity purified against the respective peptides and proved specific by western blot using *TMEM206*[−/−] HEK cells[18] as control.

**Cell culture.** HeLa (DSMZ, ACC 57), HT-1080 (DSMZ, ACC 315), MIA PaCa-2 (DSMZ, ACC 733) and MIA PaCa-2 TMEM206 KO (generated by CRISR–Cas9), HEK293 (DSMZ, ACC 305) and HEK 293 *TMEM206*[−/−] (generated by CRISPR–Cas9)[18] were maintained in DMEM (PAN Biotech, P04-03550) supplemented with 10% FBS (PAN Biotech, P40-37500) and 1% penicillin–streptomycin (PAN Biotech, P06-07100) at 37 °C and 5% CO₂. Cells were regularly tested for mycoplasma contamination with a negative outcome. Among the cancer cell lines, MIA PaCa-2 cells were used for growth assays under nutrient restriction (Fig. 7e), for which they are a well-established model system[44], whereas HT-1080 were used to examine MP resolution because these cells, but not MIA PaCa-2 cells[52], express sufficiently high levels of EGF receptors to allow acute stimulation of macropinocytosis.

**BMDM isolation, culture and transfection.** Murine BMDMs were isolated from tibiae and femora of 8- to 16-week-old C57Bl/6 mice of both sexes[53]. Cells were flushed from bones with PBS (PAN Biotech, P04-36500) and erythrocytes lysed by adding red blood cell lysis buffer (Abcam, ab204733). Remaining cells were resuspended in complete DMEM. After 4 h, non-attached cells present in the supernatant were collected and resuspended in complete DMEM with 20 ng ml⁻¹ recombinant murine M-CSF (Pepro Tech, 315-02). Cells were cultured in humid environment at 37 °C and 5% CO₂. On day 3, the amount of medium was doubled by adding complete DMEM supplemented with 20 ng ml⁻¹ M-CSF. Cells were used for experiments on days 6–14 after isolation. To induce the deletion of the *Clcn7* and *Lrrc8a* genes in BMDMs in Cx3cr1-Cre^ERT2; *Clcn7*^lox/lox and Cx3cr1-Cre^ERT2;*Lrrc8a*^lox/lox mice, respectively, 2 μM of *trans*-4-hydroxytamoxifen (Sigma-Aldrich, H7904) was added to the medium on the day of preparation and on day 3 together with M-CSF. Cx3cr1-Cre^ERT2 mouse line was crossed with *Clcn7*^lox/lox mouse line and with ROSA26^floxSTOP−YFP labelling Cre-expressing cells with YFP to identify the cells with deleted *Clcn7* in live cell imaging.

For transfections, cells were lifted after 6–7 days in culture with Versene (Thermo Fisher, 15040066) and transfected using P2 Primary Cell 4D-Nucleofector kit (Lonza, V4XP-2024) according to manufacturer's protocol. Macrophages were imaged 24 h after nucleofection. Transfected channels always carried a GFP tag (C-terminal for TMEM206, N-terminal for ClC-5) except for ClC-2 (for which we used internal ribosome entry site (IRES)-GFP) to distinguish transfected from non-transfected (NT) cells.

**Electrophysiology.** Whole-cell patch-clamp recordings of WT and *Tmem206*[−/−] BMDMs were performed at 20–22 °C using an EPC-10 USB patch-clamp amplifier and PatchMaster software (HEKA Elektronik). Patch pipettes had a resistance of 1–3 MΩ. Currents were sampled at 1 kHz and low-pass filtered at 10 kHz. The pipette solution contained (in mM): 150 CsCl, 10 EGTA, 10 HEPES and 5 MgCl₂ pH 7.2 with CsOH (320 mOsm kg⁻¹). The pH 7.4 bath solution contained (in mM): 145 NaCl, 5 KCl, 1 MgCl₂, 2 CaCl₂, 10 glucose and 10 HEPES pH 7.4 with NaOH (320 mOsm kg⁻¹, adjusted with mannitol). Solution at pH 4.8 was buffered with 5 mM Na₃ citrate instead of HEPES. From a holding potential of −30 mV, the voltage was clamped in 1-s-long 20 mV steps from −80 mV to +80 mV, preceded and followed by a 0.5 s step to −80 mV every 4 s.

**Immunocytochemistry.** WT and *TMEM206*[−/−] HEK293 cells were seeded on glass coverslips and fixed after 1 day with 4% paraformaldehyde (PFA). After antigen retrieval (citrate buffer, pH 6 at 80 °C for 18 min), cells were permeabilized with 0.1% Triton X-100 (Roth, 3051.2) for 30 min. Cells were then incubated overnight at 4 °C with custom-made primary TMEM206 antibodies targeting the C-terminus (1/500) and against endosome markers: mouse anti-Rab5 (BD Biosciences, 610725; 1/200), mouse anti-EEA1 (Abcam, ab70521; 1/250) and mouse anti-Rab7 (Santa Cruz, sc-376362; 1/100). Subsequently, cells were incubated with secondary antibodies coupled to AlexaFluor555 or AlexaFluor488. Coverslips were mounted on slides and images were acquired with a confocal microscope with a 63× NA 1.4 oil-immersion lens (LSM880, Zeiss).

Similar protocols were applied to stain BMDMs with anti-TMEM206 (as above) and the following antibodies: rat anti-LAMP1 (BD Pharmingen, 553792; 1/500), sheep anti-EEA1 (R&D Systems, AF8047; 1/20) and mouse anti-Rab5A (Cell Signaling, 46449; 1/400). We used several custom-made antibodies from our laboratory: rabbit anti- ClC-7 (1/100, ref. [19]; after antigen retrieval (citrate buffer, pH 6 at 95 °C for 10 min)), anti-ClC-5 (rabbit, 1/250, ref. [25] or guinea pig (1/100, ref. [26]), and against the a3 subunit of the H⁺-ATPase (guinea pig, 1/100, ref. [33]). For ClC-3, LRRC8A and LRRC8D, we used different knock-in mice previously generated in our group: Venus-tagged ClC-3 (ref. [36]), 3× HA-tagged LRRC8A (ref. [54]) and TdTomato-tagged LRRC8D (ref. [55]). Proteins were detected using chicken anti-GFP (Aveslab, GFP-1020; 1/500), rabbit anti-HA (Cell Signaling, 3724; 1/500) and rabbit anti-RFP (Rockland, 600-401-379; 1/500) antibodies, respectively. For immunohistochemistry of newly generated MPs, BMDMs were incubated with M-CSF (100 ng ml⁻¹) for 4 min and then fixed immediately (timepoint 4 min), or fixed 3 min after replacing the M-CSF containing medium by complete medium (without M-CSF) (timepoint 7 min). Key stainings were done with BMDMs derived from two to three animals. On average, 15 cells per condition were analysed.

To study VRAC localization in HeLa cells, these were transfected with untagged LRRC8A and TdTomato-tagged LRRC8D (1:1) in pcDNA3.1 backbone vector using Lipofectamine 2000 (Invitrogen, 11668-019). One day after transfection, cells were fixed with PFA 4% and LRRC8A/LRRC8D heteromers were detected using rabbit anti-RFP antibody (Rockland, 600-401-379; 1/500).

**Western blot.** To ascertain the specificity of our TMEM206 antibodies in western blots, WT and *Tmem206*[−/−] HEK293 cells were collected by scraping in buffer containing (in mM): 140 NaCl, 20 Tris–HCl pH 7.4, 5 EDTA, 4 Pefabloc (Roth, A154.3) and cOmplete proteinase inhibitors (Merck, 11836145001). After homogenization, cell debris was removed by centrifugation at 1,000g for 10 min. Membrane pellets obtained from the supernatant by centrifuging at 269,000g for 30 min were resuspended in lysis buffer containing (in mM): 140 NaCl, 50 Tris–HCl pH 6.8, 5 EDTA, 1% SDS, 4 Pefabloc (Roth, A154.3) and cOmplete proteinase inhibitors (Merck, 11836145001). Protein concentration was determined by bicinchoninic acid assay. Samples were denatured at 55 °C in Laemmli buffer for 10 min.

To determine tissue distribution of TMEM206, organs were collected from mice (killed by cervical dislocation) and directly snap-frozen. BMDMs were collected as described above. Membrane fractions were prepared as described for HEK cells. To detect other transport proteins, tissues were lysed and incubated on ice for 15 min in RIPA buffer, containing (in mM): 50 Tris–HCl pH 8, 150 NaCl, 1% NP40, 0.5% sodium desoxycholate, 0.1 % SDS, 4 Pefabloc and cOmplete proteinase inhibitor cocktail. Samples were then sonicated and centrifuged at 20,000g for 10 min at 4 °C to remove non-lysed cells. After measuring the protein concentration by bicinchoninic acid assay, samples were denatured at 55 °C in Laemmli buffer for 10 min.

Samples (30 μg protein) were separated by SDS–PAGE and blotted. Detection used our anti-TMEM206 antibody (1/1,000, against extracellular loop), rabbit anti-ClC-2 (ref. [56]) (1/1,000), rabbit anti-ClC-3 (ref. [57]) (1/500), rabbit anti-ClC-4 (ref. [57]) (1/200), rabbit anti-ClC-5 (ref. [26]) (1/200), rabbit anti-ClC-6 (ref. [34]) (1/500), rabbit anti-ClC-7 (ref. [20]) (1/100) and rabbit anti-LRRC8A (ref. [11]) (1/1,000). Na/K ATPase α1 subunit and actin (loading controls) were detected with a mouse anti-Na/K ATPase α1 subunit antibody (Millipore, 05–369; 1/10,000), and mouse or rabbit anti-actin antibody (Sigma, A2228 and A2066; 1/1,000 for both), respectively.

**Macropinocytosis live imaging.** Twenty-four hours before imaging, 1.5×10⁵ BMDMs were seeded in M-CSF-free complete DMEM on the glass part of uncoated glass-bottom 35 mm dishes (MatTek, P35G-1.5-10-C). First, medium was replaced by a live cell imaging buffer containing (in mM): 150 NaCl, 1 MgCl₂, 1 CaCl₂, 5 KCl, 5 glucose and 20 HEPES pH 7.2. The osmolality (determined by a freezing point osmometer (Gonotec, Osmomat 3000)) was 320 mOsm kg⁻¹. For ion substitutions, NaCl was replaced with 150 mM N-methyl-D-glucamine-chloride (Na⁺ replacement) or Na-gluconate (Cl⁻ replacement). Cells were stimulated in imaging buffer containing 100 ng ml⁻¹ recombinant M-CSF and one of the fluorophore-conjugated dextrans: 70 kDa TMR–dextran (D1818) (0.5 μg μl⁻¹) or Oregon Green–dextran (D7173) (0.3 μg μl⁻¹), both from Thermo Fisher Scientific. After 3.5 min, cells were washed and resolution of the vesicles was monitored every minute from minute 5 to minute 15 in imaging buffer (every

30 s for overexpression experiments). In pH measurements with 70 kDa Oregon Green 488–dextran, 20 mM NH$_4$Cl was added after 15 min to alkalinize the MP as a control for Oregon Green bleaching. Where stated, V-ATPase was inhibited with 100 nM bafilomycin A1 (Alfa Aestar, J61835) (10 min pre-incubation before M-CSF stimulation, and throughout the rest of the experiment). Where stated, MPs were alkalinized with 20 mM NH$_4$Cl added together with M-CSF. After washing, 20 mM NH$_4$Cl remained in the imaging buffer.

Imaging was performed with a Nikon spinning disc microscope (Yokogawa spinning disk CSU-X1) operated with NIS software in live cell imaging buffer at 37 °C (Okolab incubator) every minute in z-stacks with 0.6 µm steps on 60× oil objective (Plan-Apo, NA 1.40, Nikon) with an additional 2× magnification on an EMCCD Camera (AU-888, Andor). Depending on fluorophore, the following lasers were used: excitation 488 nm (emission 525/50 nm) for GFP, excitation 515 nm (emission 540/30 nm) for YFP, excitation 561 nm (emission 600/50 nm) for TMR-coupled dextran with 200 ms exposure time. For pH measurements with Oregon Green 488, two excitation wavelengths were used: 445 nm and 515 nm with emission at 540/30 nm. Final pixel size was 0.11 µm.

For MP shrinkage in HT-1080 cells, these were transfected with ON-TARGETplus human siRNA SMART pool against TMEM206 (55248) (Dharmacon, L-021036-02-0005) or non-targeting control pool (Dharmacon, D-001810-10-05) with Lipofectamine 3000 transfection reagent (Invitrogen, L3000015) according to the manufacturer protocol. Forty-eight hours post-transfection, cells were seeded in complete DMEM on uncoated glass-bottom 35 mm dishes (MatTek) dishes and imaged 24 h later. Sixteen hours before imaging, cells were starved in FBS-free DMEM. Human EGF (PeproTech, AF-100-15; 200 ng ml$^{-1}$) was used to stimulate micropinocytosis. Imaging and tracking of MPs was done as for BMDMs. After each experiment, knockdown (KD) efficiency was verified by western blot.

**Oregon Green calibration.** BMDMs were seeded as for resolution or pH experiments. Cells were stimulated with imaging buffer containing 100 ng ml$^{-1}$ recombinant M-CSF and 0.3 µg µl$^{-1}$ 70 kDa Oregon Green–dextran for 3 min. Macrophages were then washed and the solution was replaced by the pH calibration solution containing 10 µM monensin (Sigma-Aldrich, M5273) and 10 µM nigericin (Sigma-Aldrich, N7143). Imaging was done after 5 min incubation with ionophores in z-stacks as described above. Calibration solution contained (in mM): 5 NaCl, 115 KCl, 1.2 MgSO$_4$, 10 glucose and 25 of buffer (for solutions with pH 4–5 citric acid/sodium citrate buffer, pH 5.5–6.5 MES and pH 7.2–8.0 HEPES).

**Data analysis of macropinocytosis experiments.** We began by reducing every three-dimensional stack of images to a two-dimensional maximum projection along the z axis (using ImageJ's native maximum projection function). Further analysis used Python (version 3.8) scripts. Each image sequence was analysed starting from the last timepoint, tracking MPs with high signal through subsequent z-projected images back to the first time step where signal-to-noise ratios and detection efficiency were much lower.

For the detection of MPs, intensity histograms of each image were truncated at the upper 95th and bottom 15th percentiles to distinguish MPs from background and remove outlier pixels. The histogram-truncated image was then segmented using an inverse_gaussian_gradient filter (scikit-image function skimage. segmentation.inverse_gaussian_gradient, parameters alpha = 100, sigma = 5 pixels), followed by thresholding the image below the 30th percentile of the intensity distribution (of the inverse filtered images). In other words, all pixel values below the 30th percentile of the distribution were set to 1 or 'True', and all values above the 30th percentile were set to 0 or 'False'. We then detected peripheral cell contours upon this image mask using the findContours and drawContours functions in the OpenCV package (version 4.2.0). Contour masks were then generated using the detected contours, with image regions outside the masks being set to 0. Initial candidates for MPs (blobs) were detected by filtering the image with a sequence of difference of Gaussian (DoG) filters (scikit-image package, DoG function skimage.feature.blob_dog) with minimum and maximum standard deviations of the Gaussian kernels used set to 1 and 15 pixels correspondingly and absolute lower bound for scale space maxima set to 0.01. To further select only true MPs among this initial set of blobs, we cropped the image into 50 × 50 pixel boxes centred around each blob and then found edges within each blob using the canny detector (scikit-image skimage.feature.canny, sigma = 4 pixels lower and upper thresholds 0 and 100 pixels corresponding). We further used the edge-enhanced images to detect five circles using the Hough Circle Transform function of scikit-image package (function: scimage.transform.hough_circle, hough_circle_peaks, parameters: radii 3–35 pixels). These final circles were considered 'true' MPs only if at least two of five detected circles within each cropped image lies less than or equal to five pixels (which covers the circular area of 0.95 µm$^2$) distance from the centre of the image; in case of multiple circles falling under the five-pixel-within-centre criterion, the final radius of MPs was calculated as a 95th percentile of among these detected radii. Tracking the same MP from time step to time step was based on correlations in the displacements of the centroids: for each subsequent image, whichever of the detected MP was the closest to the centre of a MP in a previous (reference) image, was considered to be the same. The MP, however, was only considered the same if the distance between centres of MPs

from consecutive timeframes did not exceed 30 pixels (that is, 3.3 µm). Volume was calculated from the radius by assuming a perfectly spherical MP (that is, $V = (4/3)\pi r^3$).

We measured the intensity of the MPs on the original image as the average pixel intensities within an inner circle whose radius was half that of the radius of the MP (where the 'full radius' was determined using the Hough circle detector described above).

For ratiometric pH measurements, MPs were detected as described above at the $\lambda_{exc} = 515$ nm channel. Intensities were measured on both images at the same coordinates. Background values were calculated as 2nd percentile across the whole z-projected image individually for each time step image and further subtracted from MP intensity values. As a final step, for each MP, intensities at the pH-sensitive wavelength ($\lambda = 515$ nm) were normalized to the corresponding pH-insensitive wavelength ($\lambda = 445$ nm).

When using heterologous expression, transfected cells were selected and cropped from the image manually on the basis of GFP expression.

**Scratch assay.** BMDMs were seeded in M-CSF-free complete DMEM on eight-well glass bottom µ-slide (Ibidi, 80827) at 85% confluency minimally 24 h before the experiment. Cells were placed either in complete DMEM containing 150 ng ml$^{-1}$ recombinant murine C5a (PeproTech, 315-40) or in a C5a-free DMEM. The cell monolayer was scratched with a 10-µl pipette tip and immediately placed on the microscope for imaging. Phase contrast imaging of the scratch and surrounding cells was performed on a Nikon Ti Eclipse microscope operated with NIS software in complete DMEM ± C5a at 37 °C, 5% CO$_2$ (Okolab incubator) every hour for 12 h on 20× air objective (Plan Apo NA 0.75, Nikon) with back-illuminated sCMOS camera (Prime95b, Photometrics) using Nikon Volume Contrast function. Final pixel size was 0.55 µm.

Data were analysed using custom code using Python (version 3.8). The first image in the time series ($t = 0$) was segmented using an inverse_Gaussian_gradient filter (scikit-image function skimage.segmentation.inverse_gaussian_gradient, parameters: alpha = 100, sigma = 5 pixels), followed by thresholding the image below the 20th percentile of the intensity distribution (of the inverse filtered images). Then images were thresholded using skimage.filters.threshold_local function of scikit-image (parameters: block size 41, offset 10). Further, we performed morphological opening of the image (erosion followed by a dilation) using skimage.morphology.opening function with disk footprint with 40 px radius following by two morphological reconstructions of the image (first dilation, then erosion, skimage.morphology.reconstruction function) to separate scratch area from surrounding cells. We detected the borders of the scratch upon this image mask using the findContours and drawContours functions in the OpenCV package (version 4.2.0) and selecting the longest detected contour. Using the same mask for the original scratch area, we detected and counted the number of cells at every following time step image by filtering the image with a sequence of difference of Gaussian (DoG) filters (scikit-image package, DoG function skimage.feature.blob_dog) with minimum and maximum standard deviations of the Gaussian kernels used set to 5 and 30 pixels correspondingly and absolute lower bound for scale space maxima set to 0.02. Finally, we subtracted the number of cells detected on the scratch area at $t = 0$ from number of cells detected at every following timeframe and normalize this number to the original area of scratch (calculated in number of pixels and further converted to µm$^2$).

**MIA PaCa-2 _TMEM206_ KO generation.** Clonal MIA PaCa-2 KO cell lines were generated using CRISPR-Cas9 with two different sgRNAs: sgRNA1 (GGACCGAGAAGACGTTCTTC) and sgRNA2 (CAGCTGTAAGCACCATTACG) as described previously for HEK cells[18]. KO was verified by Sanger sequencing and western blot. For growth assays, three KO clones (SB-1G11 Δ8 nucleotides, FS-1A2 Δ1 and +1 nucleotides, and FS-1A7 + 1 nucleotide) and three clones that went through the transfection process but remained WT.

**MIA PaCa-2 growth assay.** Twenty-four hours before experiments, cells were seeded on 96-well plate in quadruplicates at 30% density in complete DMEM. The next day the medium was replaced with one of following: (1) amino-acid-free DMEM (Genaxxon, C4150.0500) supplemented with 5% essential amino acids (Merck, M5550), 10% dialysed FBS (Pan Biotech, P30-2101), 1% penicillin–streptomycin and additional 1 mM CaCl$_2$ (final Ca$^{2+}$ concentration 2 mM) or (2) amino-acid-free DMEM supplemented with 5% essential amino acids, 10% dialysed FBS, 1% penicillin–streptomycin, 3% BSA (Pan Biotech, P06-10200) and additional 3 mM CaCl$_2$ (final Ca$^{2+}$ concentration 4 mM). Ca$^{2+}$ concentration was adjusted in BSA-containing solutions to obtain similar free Ca$^{2+}$ concentrations (BSA is known to markedly bind Ca$^{2+}$ (ref. [58])) because constitutive macropinocytosis is stimulated by extracellular free Ca$^{2+}$ through Ca$^{2+}$-sensing receptors[59]. After 72 h incubation at 37 °C and 5% CO$_2$, we evaluated cell survival in both conditions with XTT assay (Cell Signaling, 9095) measuring absorbance at 450 nm with an absorbance plate reader (ASYS Hitech) as readout. Data are presented (Fig. 7e) as absorbance ratios between BSA-exposed and unexposed cells.

**Phagocytosis of beads.** WT and _Tmem206$^{-/-}$_ BMDMs were seeded on coverslips for assessing phagocytosis in situ. The day after, yellow-green fluorescent beads

(Fluoresbrite carboxylated microspheres, Polyscience, 17147-5 and 18141-2) of 3 or 6 μm diameter were diluted in complete DMEM and applied onto cells for 30 min at 37 °C. When using cytochalasin D (Sigma, C8273), cells were first pre-treated for 1 h with cytochalasin D at 10 μM and beads were then diluted in complete DMEM with 10 μM cytochalasin D. After incubation, cells were washed twice with warm complete medium and fixed with 4% PFA. Cells were finally permeabilized to stain nuclei with DAPI (Sigma-Aldrich, MBD0015). About ten images per condition were randomly acquired with a confocal microscope with a 63× NA 1.4 oil-immersion lens (LSM880, Zeiss), and beads present inside the cells were counted manually using Fiji software.

**Phagocytosis of *E. coli*.** In order to determine whether TMEM206 is present in phagosomes, WT BMDM were incubated with green fluorescent *Escherichia coli* (*E. coli* (K-12 strain) BioParticles, Alexa Fluor 488 conjugate, Thermo Fisher, E13231) diluted in complete DMEM to obtain a macrophages/*E. coli* ratio of 1/10. After 15 min of incubation, cells were washed twice with complete DMEM and incubated for additional 5 min with complete DMEM at 37 °C. Cells were then fixed with 4% PFA and stained for TMEM206 as described above.

**Statistics and reproducibility.** Statistical tests were performed using GraphPad Prism (v.7.03). Statistical test was chosen on the basis of data distribution normality, which was checked with Shapiro–Wilk test. Either unpaired two-tailed parametric *t*-test or two-tailed non-parametric Mann–Whitney test was performed for comparing two groups of data. For multiple comparisons, either one-way analysis of variance (ANOVA) (parametric) or Kruskal–Wallis (non-parametric) tests were used.

On average, 30 MPs were detected and successfully tracked through all timeframes per one round of imaging on the MatTek dish. One to five MatTek dishes were imaged per condition per different BMDM preparation. MP measurements (volume, fluorescent intensity or intensity ratio) were averaged first per dish and then per BMDM preparation, which formed one data point on the plot. For overexpression experiments, MP measurements (volume and fluorescent intensity) were averaged per cell and then per transfection round (within the same animal). Statistical comparison was done on averaged values per cropped image with transfected cell. Plots represent means ± s.e.m. *n* is animal number, and *N* is number of MPs.

No statistical methods were used to predetermine sample sizes. All western blot or immunofluorescence experiments were repeated at least two times independently with similar results being obtained. All MP shrinkage experiments were repeated at least three times independently with similar results.

**Co-localization analysis.** For co-localization analyses separate cells or single MPs were cropped in rectangles minimizing the surrounding background area. Pearson's correlation coefficient above threshold was calculated using Fiji (bisection threshold regression). Pearson's *R* was plotted for separate cells (each dot corresponds to one cell) or first averaged across MPs present within the same one cell and then plotted (each dot corresponds to an average of MPs per cell). Plots represent means ± s.e.m. *n* is number of analysed cells.

**Reporting summary.** Further information on research design is available in the Nature Research Reporting Summary linked to this article.

## Data availability
Source data are provided with this paper. All other data supporting the findings of this study are available from the corresponding author on reasonable request.

## Code availability
The code is freely avaliable on Github repository: mathematical model (https://github.com/mzeziulia/MP_volume_modelling); MP detection and analysis (https://github.com/mzeziulia/MP_detection_analysis); scratch assay analysis (https://github.com/mzeziulia/Scratch_assay).

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

## Acknowledgements
We thank F. Ullrich for measuring the currents shown in Fig. 2b, M. M. Polovitskaya for electrophysiological control experiments and help with schemes, A. Günther and C. Backhaus for technical assistance and C. Heins for consulting on Python programming. This work was supported by the European Research Council (ERC) Advanced Grant VOLSIGNAL (#740537), the Deutsche Forschungsgemeinschaft (FOR 2652 (Je164/14-2), SFB1365 (B02) and Exc257 'NeuroCure') and the Prix Louis-Jeantet de Médecine to T.J.J., and by a European Union's Horizon 2020 individual fellowship under the Marie Sklodowska-Curie grant agreement #795753 to S.B.

## Author contributions
M.Z. designed, performed and analysed MP live cell imaging and physiological assays and implemented numerical simulations of the mathematical model in Python. S.B. generated and analysed *Tmem206⁻/⁻* mice and KO cell lines and designed, performed and analysed immunohistochemistry and western blot experiments. F.W.S. generated KO cell lines and performed and quantified immunohistochemistry. M.L. implemented MP live cell imaging. T.J.J. designed and evaluated experiments, developed the mathematical model and wrote the manuscript. M.Z., S.B., F.W.S. and T.J.J. edited the manuscript.

## Funding
V. (FMP)

## Competing interests
The authors declare no competing interests.

## Additional information
**Extended data** is available for this paper at https://doi.org/10.1038/s41556-022-00912-0.

**Correspondence and requests for materials** should be addressed to Thomas J. Jentsch.

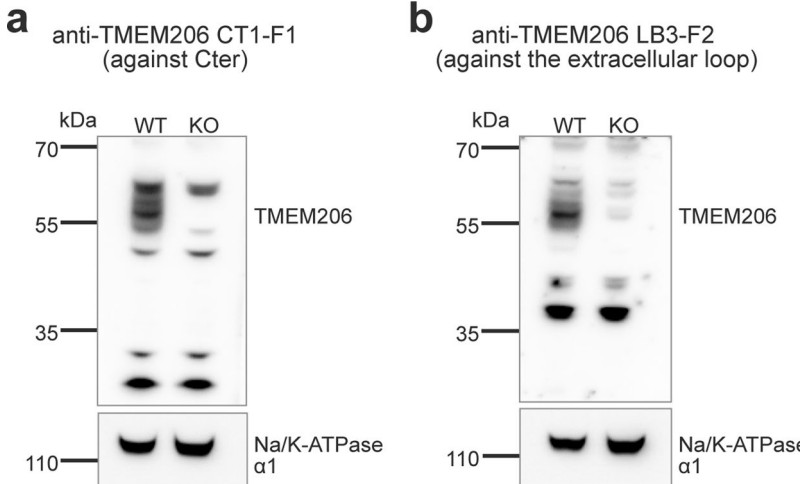

**Extended Data Fig. 1 | Specificity of custom-made antibodies against TMEM206.** Western Blot of membrane preparations of WT and *Tmem206* KO HEK cells[18] to check for specificity of two custom-made rabbit anti-TMEM206 antibodies, one directed against the C-terminus of mouse TMEM206 (against the peptide sequence IKIRKRYLKRRGQATNHIS) **(a)** and another directed against the extracellular loop (against the peptide sequence VKTKEEDGREAVEFRQET) **(b)**. Both antibodies had been affinity-purified with the cognate peptide. While both antibodies recognized several unspecific bands, bands at the correct size were missing in KO samples. CT1-F1 antibody was used for the immunodetection of TMEM206 in BMDMs and HEK cells after antigen retrieval and proved to be specific as evident from loss of signal in *Tmem206* KO cells (Fig. 2a, Extended Data Fig. 3a). Na/K-ATPase α1 subunit was used as loading control. 30 μg of membrane preparation were loaded per lane for each sample. Source unprocessed blots are available in source data.

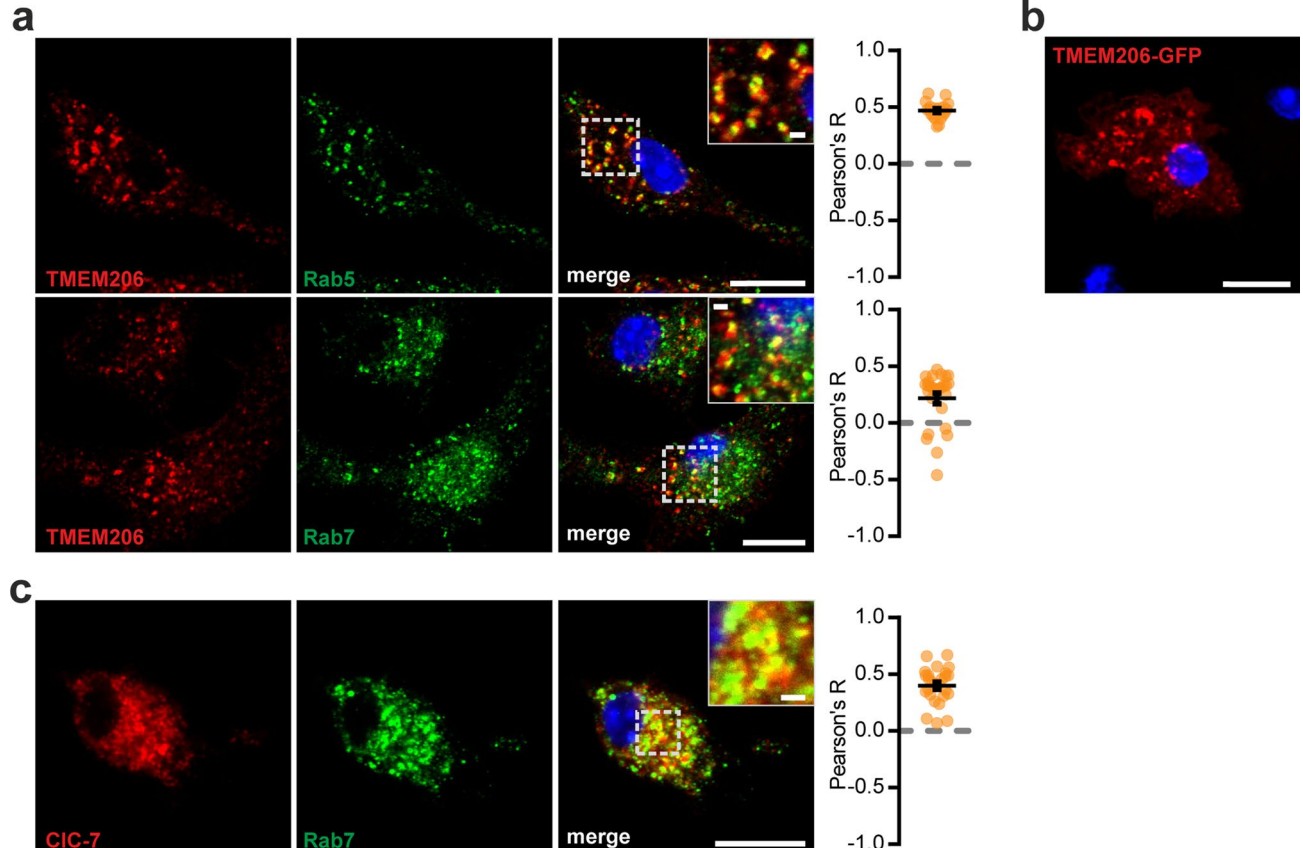

**Extended Data Fig. 2 | Expression patterns of TMEM206 and ClC-7 in primary bone marrow-derived macrophages. (a)** Endogenous TMEM206 co-localizes with early endosomal rab5 and partially with late endosomal rab7. Pearson's R calculated from 19 cells (Rab5), 27 cells (Rab7) from 2 independent experiments. **(b)** Expression of human TMEM206 C-terminally tagged with GFP in *Tmem206−/−* primary macrophages gives similar expression pattern as endogenous TMEM206 protein. **(c)** ClC-7 co-localizes with late endosome marker rab7. Pearson's R calculated from 26 cells from 2 independent experiments. Mean ± s.e.m. Scale bars: 10 μm, 1 μm for enlargements. Source numerical data are available in source data.

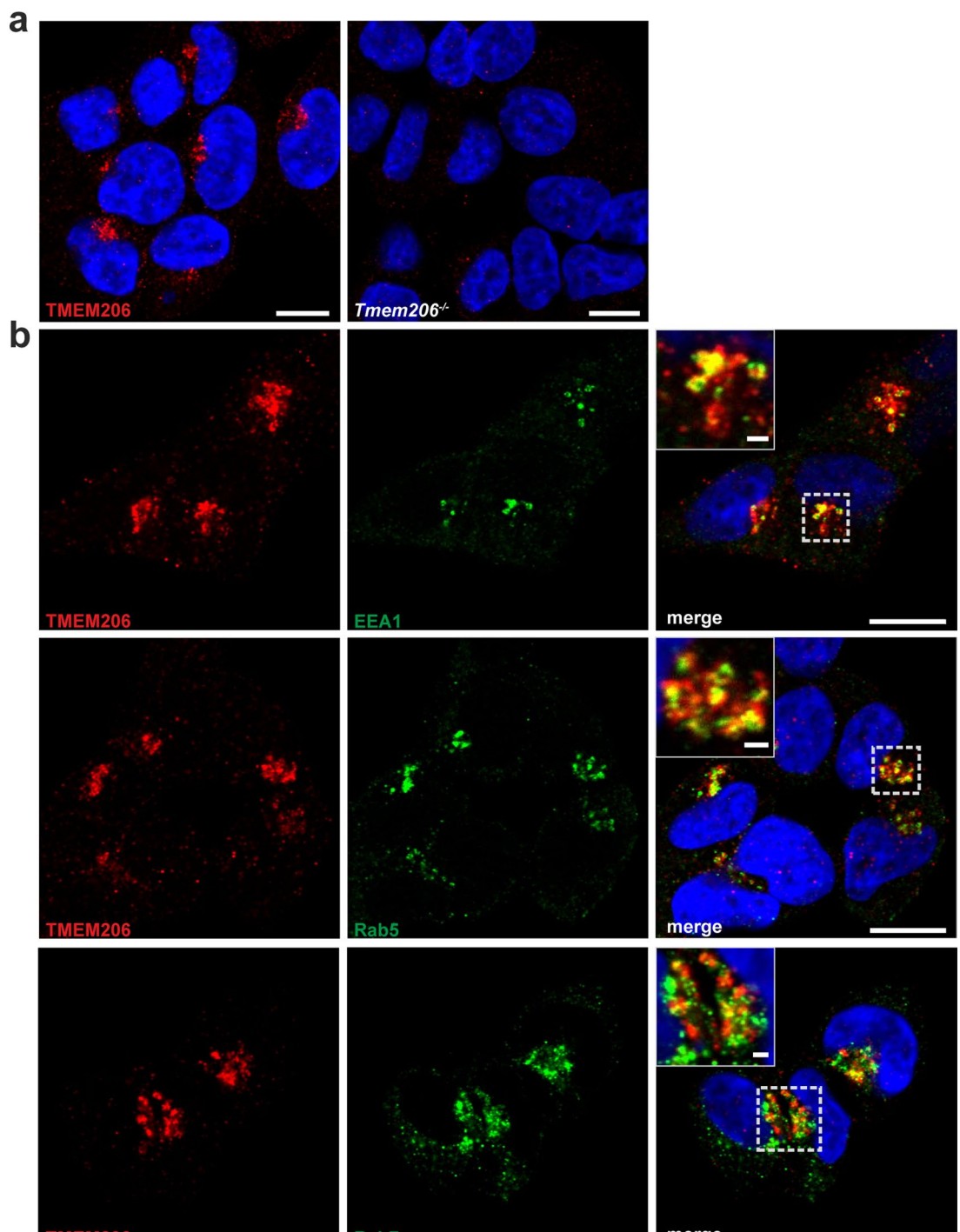

**Extended Data Fig. 3 | TMEM206 is present in early endosomes in HEK cells. (a)** Immunofluorescence staining of endogenous TMEM206 in HEK WT cells. This staining is abolished in TMEM206 KO HEK cells. **(b)** Endogenous TMEM206 co-localizes with EEA1 and rab5, early endosomes markers, and partially with rab7, a marker of late endosomes. Scale bars: 10 μm, 1 μm for enlargements.

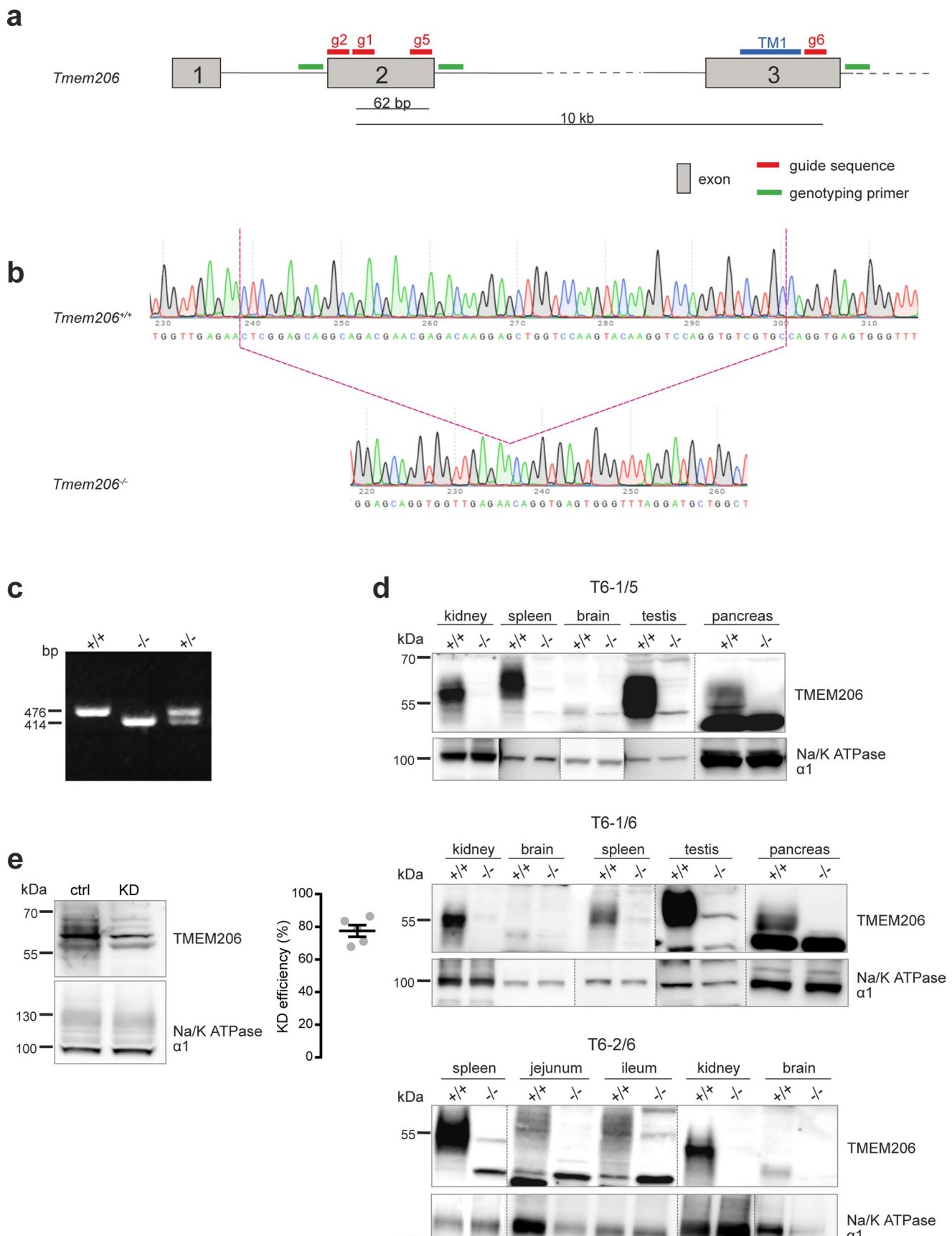

**Extended Data Fig. 4 | See next page for caption.**

**Extended Data Fig. 4 | Generation of *Tmem206*⁻/⁻ mouse lines and *TMEM206* KD in HT-1080 cells. (a)** Strategy for disrupting *Tmem206* in mice using CRISPR-Cas9 genome editing technique. gRNAs (in red) g1 and g5, both targeting exon 2, coding for the N terminus of TMEM206, were injected together and led to a 62 bp deletion. Similarly, g1 or g2, also targeting exon 2, were injected together with g6 targeting exon 3 after the sequence coding for the first transmembrane domain, leading to a deletion of ≈ 10 kbp. These injections led to the generation of three different *Tmem206* KO mouse lines called T6-1/5 (injection of g1 and g5), T6-1/6 (injection of g1 and g6) and T6-2/6 (injection of g2 and g6). **(b)** Sequencing of one of the founders of the T6-1/5 line. **(c)** Genotyping PCR of WT homozygous (+/+), heterozygous (+/-) and KO homozygous (-/-) mice from the T6-1/5 line using the genotyping primers (a, in green) flanking the exon 2. **(d)** Western blot analysis of TMEM206 expression in different tissues from WT and KO mice confirmed that TMEM206 is deleted in all 3 different KO lines. Key experiments were done with BMDMs from these three different lines to exclude possible off-target effects of sgRNAs. 30 μg protein of membrane preparation were loaded per lane for each sample. TMEM206 proteins were detected using the custom-made antibodies targeted against the extracellular loop. Na/K-ATPase α1 subunit was used as loading control. For the WB done with animals from the T6-1/5 line, TMEM206 signals from different organs were obtained from different membranes but at the same exposure. This also holds true for the T6-1/6 line, except for the pancreas samples, which were obtained after shorter exposure time compared to the other organs. For the T6-2/6 line, all the signals come from the same membrane and exposure time. Na/K ATPase signals were obtained from different membranes and exposure time was different among the different organs (as organs express different amounts of Na/K ATPase) for the three lines. Lanes coming from different membrane or exposure time are delineated by dotted grey lines. Importantly, WT and KO samples (from the corresponding organs and mouse lines) always had the same exposure time to confirm the lack of TMEM206 expression in the KO animals. **(e)** TMEM206 knock-down (KD) efficiency in HT-1080 cells compared to cells transfected with non-targeting siRNA (ctrl), quantified in Fiji, 30 μg of protein per lane, n=5 (independent transfections). Na/K ATPase is a loading control. TMEM206 protein was detected using custom-made antibody against the extracellular loop. Mean ± s.e.m. Source numerical data and unprocessed blots are available in source data.

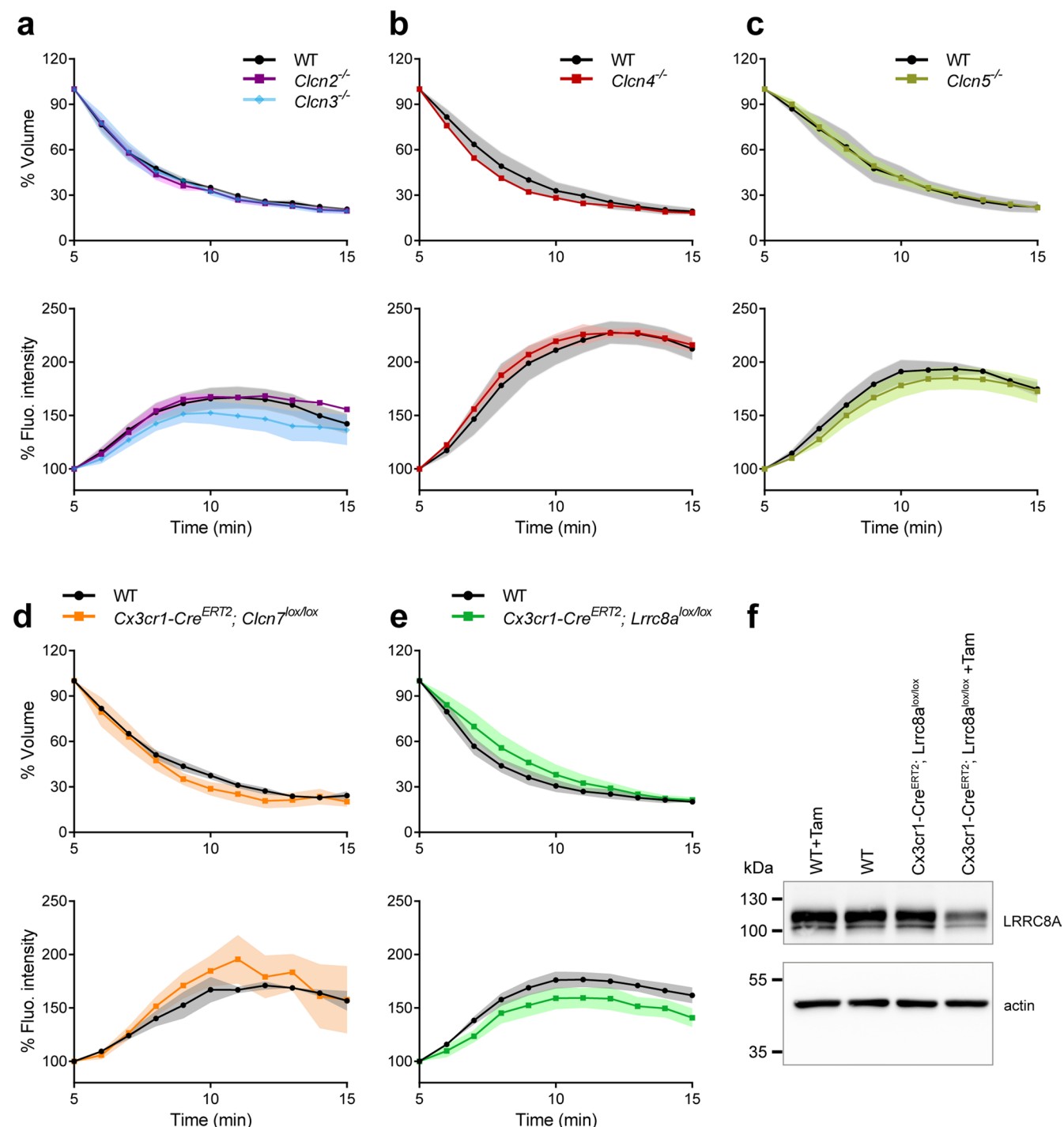

**Extended Data Fig. 5 | Disruption of *Clcn2*, *Clcn3*, *Clcn4*, *Clcn5*, *Clcn7* and *Lrrc8a* did not affect macropinosome resolution.** (a-e) For each genotype volume decrease (upper panels) and fluorescence intensity increase (lower panels) are not significantly different from WT (*n*=3 mice for *Clcn2*[−/−] (*N*=294) and *Clcn3*[−/−] (*N*=266), *n*=4, *N*=125 for WT); *n*=3 for *Clcn4*[−/−] (*N*=270) and WT (*N*=235); *n*=5, *N*=822 for *Clcn5*[−/−] and *n*=3, *N*=263 for WT; *n*=3 for *Clcn7*[−/−] (*N*=19) and WT (*N*=74); *n*=4, *N*=311 for *Lrrc8a*[−/−] and *n*=3, *N*=159 for WT. *Clcn7*[−/−] cells were selected by YFP expression. Plot of mean ± s.e.m. (shown as bands), averaging means from individual mice. **(f)** Western blot showing decreased expression of LRRC8A in Cx3cr1-Cre[ERT2]; *Lrrc8a*[lox/lox] BMDMs after tamoxifen (Tam) induction (top panel). Actin (bottom panel) was used as loading control. 10 μg of total protein were loaded per sample. Source numerical data and unprocessed blots are available in source data.

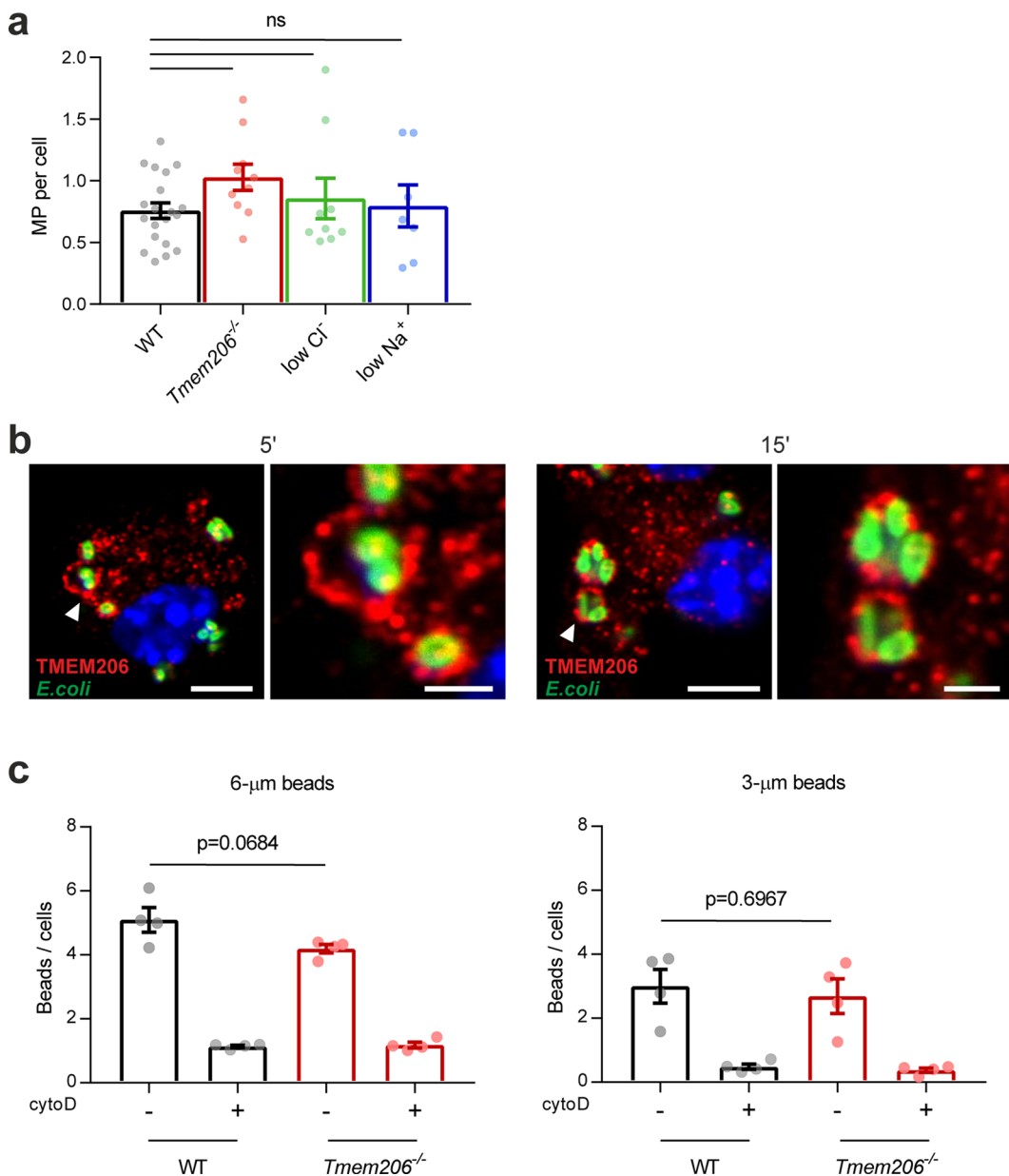

**Extended Data Fig. 6 | Initial formation of macropinosomes and phagosomes is not affected by *Tmem206* knock-out. (a)** No effect of ion replacement or Tmem206 disruption on initial formation of macropinosomes. Each dot represents the mean number of detected macropinosomes per cell from BMDM preparation from one individual mouse, with at least 100 BMDMs evaluated per mouse. n=20 (WT); n=10 (*Tmem206⁻/⁻*); n=9 (low Cl⁻, 9 mM); n=7 (low Na⁺, nominally 0 mM). No significant difference between any conditions (Kruskal-Wallis test with Dunn's multiple comparison). p = 0.3264 (WT – *Tmem206⁻/⁻*), p > 0.9999 (WT – low Cl⁻), p > 0.9999 (WT – low Na⁺). **(b)** BMDMs were incubated with killed fluorescent *E. coli* (green) and fixed after 5 and 15 min of incubation. TMEM206 was detected by immunostaining (red). Vesicles containing one or more bacteria were decorated with TMEM206-positive dots. Scale bars: 5 μm, 2 μm for enlargements. **(c)** Number of beads of 3- or 6-μm diameter phagocytosed by WT and *Tmem206⁻/⁻* macrophages, incubated with or without cytochalasin D (10 μM), an actin inhibitor. Two sizes (3- and 6-μm diameter) of beads were tested since disruption of Trpml1[60] or BK channels[61] has been reported to differentially affect only larger beads[61,62]. Similar to our observation for the formation of macropinosomes, disruption of *Tmem206* lacked an effect on the uptake of either size class of beads. n=4 mice per condition. Unpaired two-tailed t test. Mean ± s.e.m. Source numerical data are available in source data.

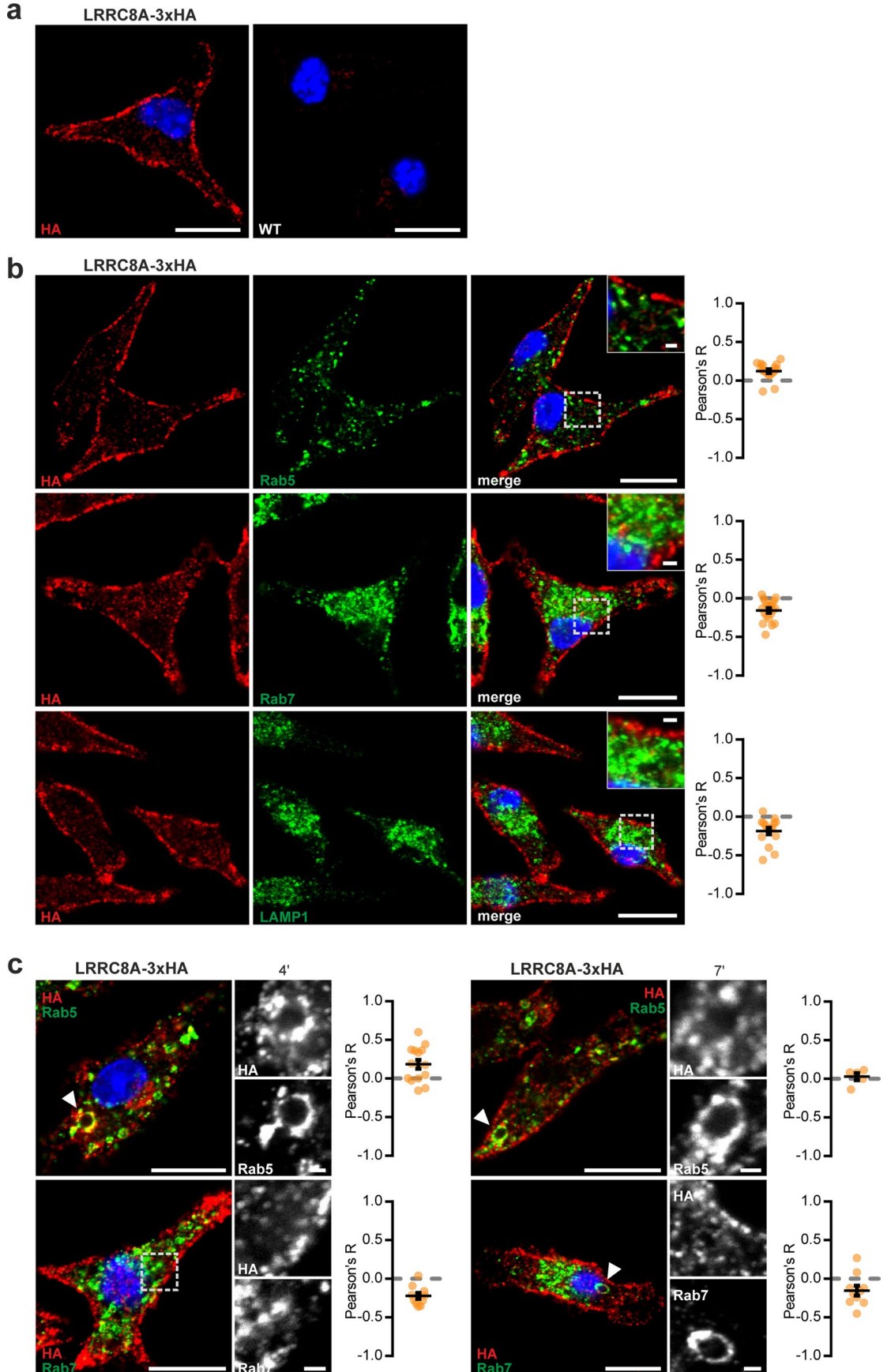

**Extended Data Fig. 7 | See next page for caption.**

**Extended Data Fig. 7 | Localization of LRRC8A in BMDMs. (a)** *Lrrc8a*[3xHA/3xHA] mice[54] that express a LRRC8 protein C-terminally tagged with 3 copies of the HA-epitope from the native genomic locus were used to determine the subcellular localization of the essential VRAC subunit LRRC8A. Anti-HA antibodies detected the protein in the plasma membrane of *Lrrc8a*[3xHA/3xHA] BMDMs but not in WT control macrophages. **(b)** LRRC8A is present at the plasma membrane of *Lrrc8a*[3xHA/3xHA] BMDMs. There is no significant co-localization with endolysosomal markers rab5 (early endosomes), rab7 (late endosomes) or LAMP1 (lysosomes). Pearson's R calculated from 18 cells (Rab5), 20 cells (Rab7), 16 cells (LAMP1). **(c)** 4 min after M-CSF addition, LRRC8A can be detected in early macropinosomes (co-stained by rab5) in ≈5-10% of cells, but not in rab7-positive mature macropinosomes. All small panels represent magnified macropinosomes or regions of interest indicated by white arrowheads. Pearson's R calculated from 15 cells (Rab5, 4min), 10 cells (Rab7, 4min), 5 cells (Rab5, 7min), 10 cells (Rab7, 7min). Mean ± s.e.m. Scale bars: 10 μm and 1 μm for the magnification panels. Source numerical data are available in source data.

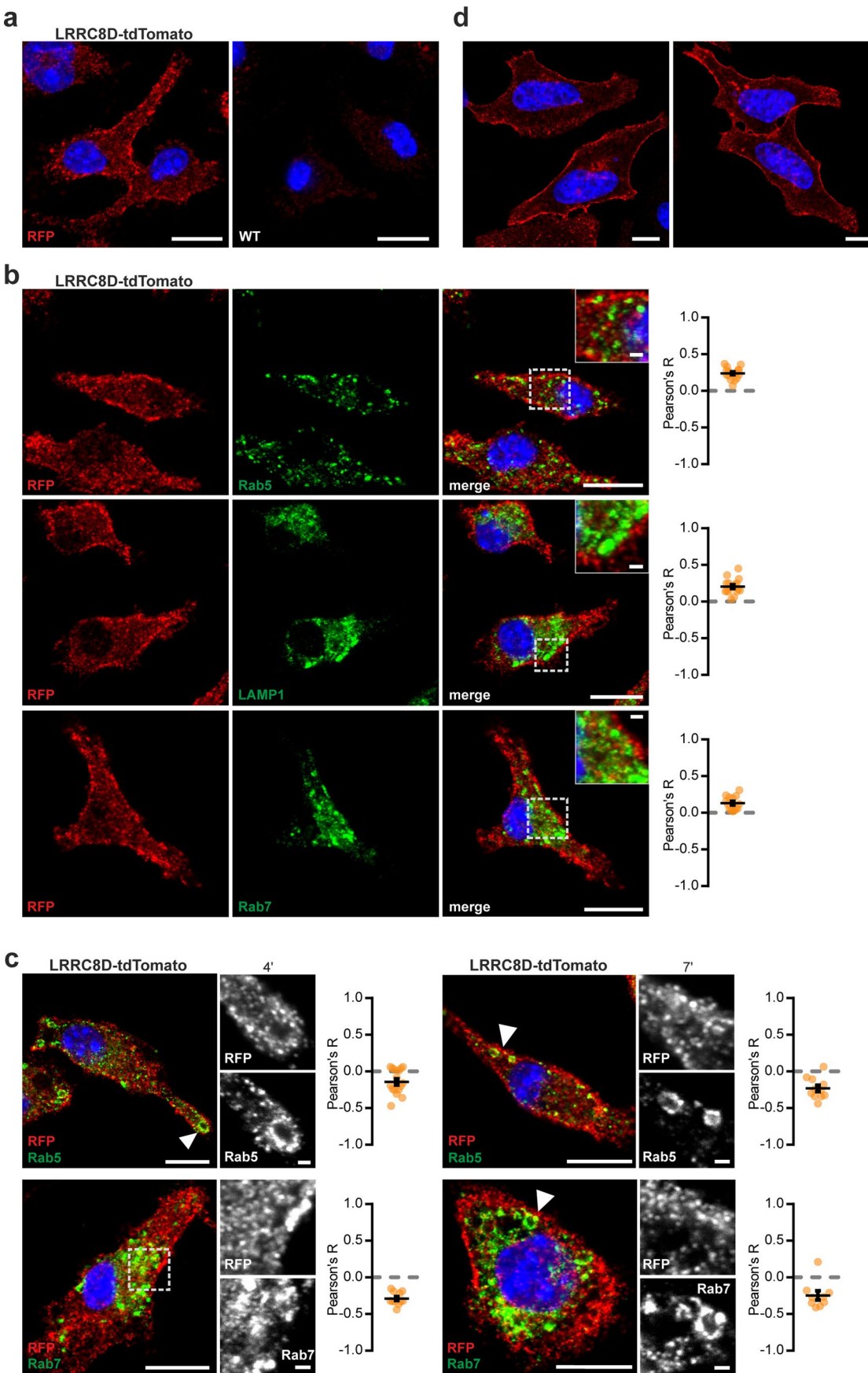

**Extended Data Fig. 8 | See next page for caption.**

**Extended Data Fig. 8 | Localization of LRRC8D in BMDMs. (a)** Localization of LRRC8D used BMDMs from *Lrrc8d*^tdTomato/tdTomato^ *knock-in* mice[55] that were stained with an anti-RFP antibody. No fluorescence signal was detected in WT macrophages (negative control). **(b)** No significant co-localization of LRRC8D in BMDMs with early endosomal rab5, late endosomal rab7 or LAMP1 (lysosomes) was observed. Pearson's R calculated from 17 cells (Rab5), 17 cells (LAMP1), 14 cells (Rab7). **(c)** LRRC8D could not be detected in macropinosomes (detected by rab5 and rab7 stainings), neither at 4 nor at 7 min after M-CSF stimulation. All small panels represent magnified macropinosomes or regions of interest indicated in left panels by white arrowheads. Pearson's R calculated from 16 cells (Rab5, 4min), 11 cells (Rab7, 4min), 11 cells (Rab5, 7min), 9 cells (Rab7, 7min). **(d)** Localization of LRRC8A/LRRC8D heteromers in transfected cells. Untagged-LRRC8A and tdTomato-tagged LRRC8D were co-transfected (ratio 1:1) in HeLa WT cells to determine the subcellular localization of LRRC8A/LRRC8D heteromers. Note that LRRC8D needs LRRC8A to leave the endoplasmic reticulum and for its transport to the plasma membrane[11]. In contrast to work by others[14], we detected LRRC8D almost exclusively at the plasma membrane but not in lysosomes (consistent with our results on BMDMs, Extended Data Figs. 7 and 8). 4 different cells expressing both proteins are shown. Mean ± s.e.m. Scale bars: 10 μm and 1 μm for enlargements. Source numerical data are available in source data.

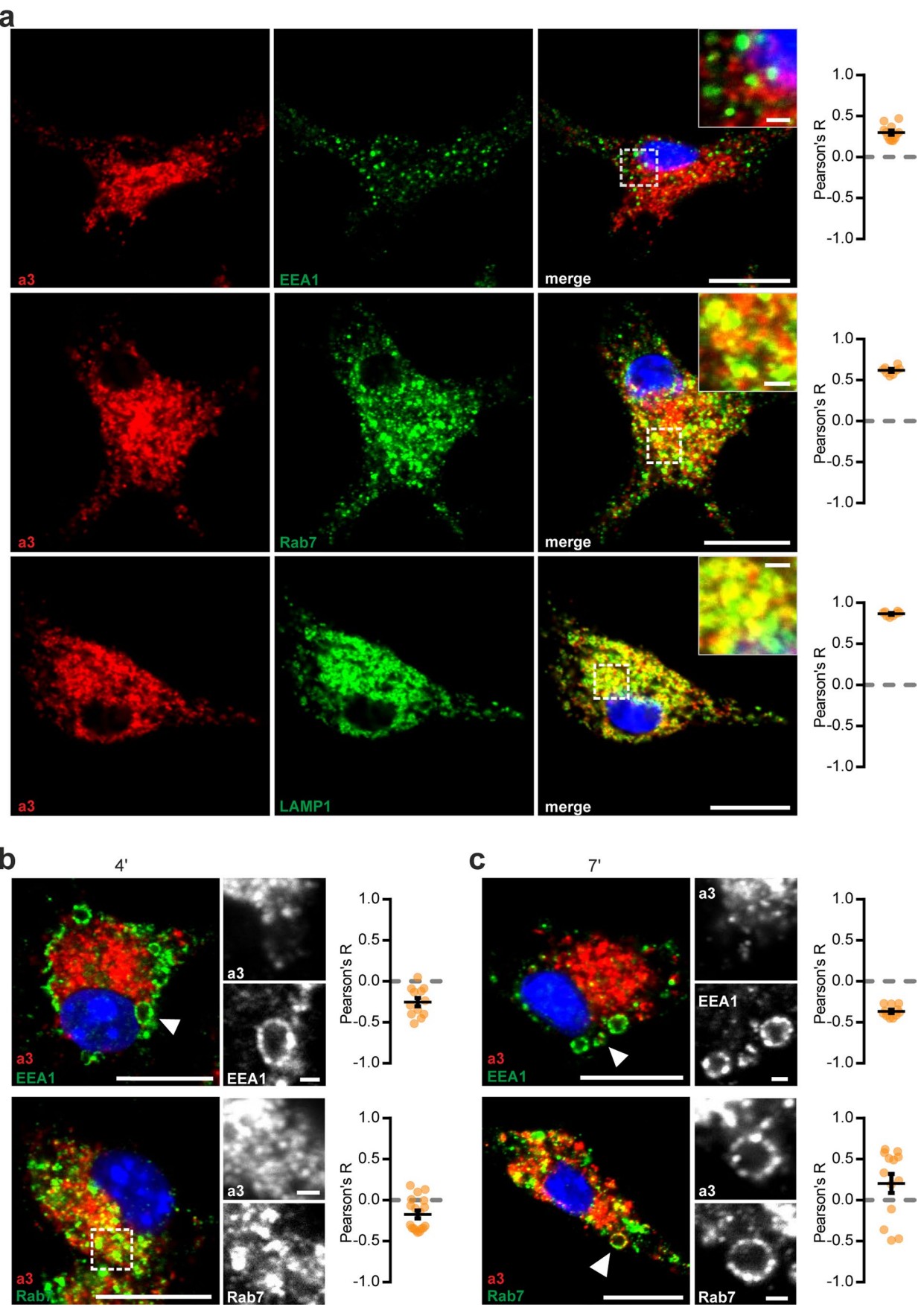

**Extended Data Fig. 9 | See next page for caption.**

**Extended Data Fig. 9 | Localization of the a3 V-ATPase subunit in BMDMs. (a)** The V-ATPase a3 subunit shows significant co-localization with lysosomal/late endosomal LAMP1 and late endosomal rab7, but not with early endosomal rab5 in unstimulated BMDMs. Pearson's R calculated from 12 cells (EEA1), 8 cells (Rab7), 8 cells (LAMP1). (b, c) Upon exposure to M-CSF, the a3 subunit was detected in rab7-positive macropinosomes after 7 min **(c)**, but not after 4 min **(b)**. It was not found in rab5-positive MPs. Pearson's R calculated from 12 cells (EEA1, 4min), 18 cells (Rab7, 4min), 11 cells (EEA1, 7min), 13 cells (Rab7, 7min). All small panels represent enlarged images of macropinosomes or regions of interest indicated by white arrowheads. Mean ± s.e.m. Scale bars: 10 μm and 1 μm for enlargements. Source numerical data are available in source data.

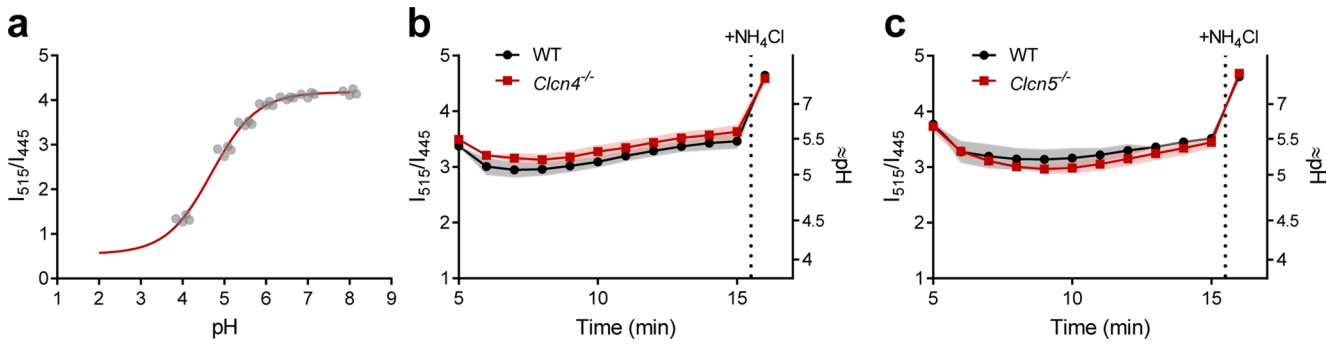

**Extended Data Fig. 10 | Luminal pH of macropinosomes. (a)** Calibration of 70 kDa Oregon Green dye with Boltzmann sigmoidal fit (red line), *n=1* (BMDMs prepared from one animal), ≈ 150 cells per replicate (each dot is a new field of view, 4 fields in total that belong to 2 different imaging dishes). pH$_{½}$ =4.7 calculated from calibration curve is similar to provided by manufacturer pK$_a$=4.7. **(b-c)** Neither disruption of *Clcn4* **(b)** nor of *Clcn5* **(c)** significantly changed macropinosomal pH when measured from 5 min after M-CSF application onwards. *n=3* for *Clcn4⁻/⁻* (*N=337*) and corresponding WT (*N=118*); *n=5, N=389* for *Clcn5⁻/⁻* and *n=2, N=141* for corresponding WT. Plot of mean ± s.e.m. (shown as bands), averaging means from individual mice. Source numerical data are available in source data.

# Reporting Summary

## Statistics

For all statistical analyses, confirm that the following items are present in the figure legend, table legend, main text, or Methods section.

| n/a | Confirmed | |
|---|---|---|
| ☐ | ☒ | The exact sample size (*n*) for each experimental group/condition, given as a discrete number and unit of measurement |
| ☐ | ☒ | A statement on whether measurements were taken from distinct samples or whether the same sample was measured repeatedly |
| ☐ | ☒ | The statistical test(s) used AND whether they are one- or two-sided<br>*Only common tests should be described solely by name; describe more complex techniques in the Methods section.* |
| ☒ | ☐ | A description of all covariates tested |
| ☐ | ☒ | A description of any assumptions or corrections, such as tests of normality and adjustment for multiple comparisons |
| ☐ | ☒ | A full description of the statistical parameters including central tendency (e.g. means) or other basic estimates (e.g. regression coefficient) AND variation (e.g. standard deviation) or associated estimates of uncertainty (e.g. confidence intervals) |
| ☐ | ☒ | For null hypothesis testing, the test statistic (e.g. *F*, *t*, *r*) with confidence intervals, effect sizes, degrees of freedom and *P* value noted<br>*Give P values as exact values whenever suitable.* |
| ☒ | ☐ | For Bayesian analysis, information on the choice of priors and Markov chain Monte Carlo settings |
| ☒ | ☐ | For hierarchical and complex designs, identification of the appropriate level for tests and full reporting of outcomes |
| ☐ | ☒ | Estimates of effect sizes (e.g. Cohen's *d*, Pearson's *r*), indicating how they were calculated |

*Our web collection on statistics for biologists contains articles on many of the points above.*

## Software and code

Policy information about availability of computer code

| Data collection | Live imaging of the macropinosomes was performed using Nikon spinning disc microscope (Yokogawa spinning disk 695 CSU-X1) operated with NIS software (version 5.02.01 Build 1270). Scratch assay was performed using Nikon Ti Eclipse microscope operated with NIS software (version 5.21.03 Build 1481). For immunocytochemistry pictures, the confocal microscope LSM880 (Zeiss, Zen Blue software 2.3) was used. Electrophysiological recordings were performed using an EPC-10 USB patch-clamp amplifier and PatchMaster software (HEKA Elektronik, version 2x90.3). For growth assay absorbance was measured with absorbance plate reader (ASYS Hitech). |
|---|---|
| Data analysis | All microscopy images were analysed using Fiji 2.0/2.1 versions. Custom data analysis code was written using Python 3.8 using OpenCV (version 4.2.0) and scikit-image (version 0.16.2) packages. Plotting and statistical analyses were performed with GraphPad Prism 7.03. |

For manuscripts utilizing custom algorithms or software that are central to the research but not yet described in published literature, software must be made available to editors and reviewers. We strongly encourage code deposition in a community repository (e.g. GitHub). See the Nature Portfolio guidelines for submitting code & software for further information.

## Data

Policy information about availability of data

All manuscripts must include a data availability statement. This statement should provide the following information, where applicable:
- Accession codes, unique identifiers, or web links for publicly available datasets
- A description of any restrictions on data availability
- For clinical datasets or third party data, please ensure that the statement adheres to our policy

Data is freely available from authors upon request. Code is freely avaliable on Github repository: mathematical model (https://github.com/mzeziulia/

MP_volume_modelling), macropinosome detection and analysis (https://github.com/mzeziulia/MP_detection_analysis), scratch assay analysis (https://github.com/mzeziulia/Scratch_assay)

# Field-specific reporting

Please select the one below that is the best fit for your research. If you are not sure, read the appropriate sections before making your selection.

☒ Life sciences ☐ Behavioural & social sciences ☐ Ecological, evolutionary & environmental sciences

For a reference copy of the document with all sections, see nature.com/documents/nr-reporting-summary-flat.pdf

# Life sciences study design

All studies must disclose on these points even when the disclosure is negative.

| | |
|---|---|
| Sample size | Sample sizes were not predetermined with statistical means but were based on standard numbers in the field. Each experiment based on live cell microscopy was performed on at least 3 animals resulting in 10-800 vesicles analysed per condition. For immunocytochemistry, on average 15 cells per condition were analysed. Western Blot analyses were performed on 2-3 different animals. For growth assay 3 different knock-out cell lines generated with 2 different sgRNAs were used, experiments were repeated 3 times, each measurement in 4 technical replicates. For scratch assay 13 pairs of animals were used. |
| Data exclusions | No data were excluded. |
| Replication | All experiments were repeated independently on minimum 2 different animals with the same outcome, all obtained quantitative data is shown in figures. Key experiments based on live cell imaging were performed on 3 different TMEM206 knock out mouse lines. Experiments using cell lines were repeated independently on cells with different passage number with the same outcome. Growth assay was performed 3 times on 3 different knock-out cell lines generated with 2 different sgRNAs with the same outcome. |
| Randomization | No animal randomization was done because mice are assigned to their group based on genotype. Microscopic fields were randomly chosen when possible. |
| Blinding | Genotypes of animals for scratch assay and genotypes of MIA PaCa-2 cells for growth assay were blinded. No blinding was applied for macropinosome shrinkage assay since it was logistically impossible but microscopic fields were randomly chosen.<br>Key experiments based on live cell imaging were analyzed automatically by a custom code, minimizing potential analysis biais. |

# Reporting for specific materials, systems and methods

We require information from authors about some types of materials, experimental systems and methods used in many studies. Here, indicate whether each material, system or method listed is relevant to your study. If you are not sure if a list item applies to your research, read the appropriate section before selecting a response.

## Materials & experimental systems

| n/a | Involved in the study |
|---|---|
| ☐ | ☒ Antibodies |
| ☐ | ☒ Eukaryotic cell lines |
| ☒ | ☐ Palaeontology and archaeology |
| ☐ | ☒ Animals and other organisms |
| ☒ | ☐ Human research participants |
| ☒ | ☐ Clinical data |
| ☒ | ☐ Dual use research of concern |

## Methods

| n/a | Involved in the study |
|---|---|
| ☒ | ☐ ChIP-seq |
| ☒ | ☐ Flow cytometry |
| ☒ | ☐ MRI-based neuroimaging |

## Antibodies

| | |
|---|---|
| Antibodies used | The following antibodies were used :<br>anti-LAMP1 (rat, BD Pharmingen, 553792)<br>anti-EEA1 (mouse, Abcam, ab70521)<br>anti-EEA1 (sheep, R&D Systems, AF8047)<br>anti-Rab5A (mouse, Cell Signaling, 46449)<br>anti-Rab5 (mouse, BD Bioscience, BD610725)<br>anti-Rab7 (mouse, SCBT, sc-376362)<br>anti-GFP (chicken, Aves lab, GFP-1020)<br>anti-HA (rabbit, Cell Signaling, 3724)<br>anti-RFP (rabbit, Rockland, 600-401-379)<br>anti- α1 Na/K ATPase clone C464.6 (mouse, Millipore, 05−369)<br>anti-actin (mouse, Sigma, A2228)<br>anti-actin (rabbit, Sigma, A2066) |

anti-ClC-2 (rabbit, Jentsch lab)
anti-ClC-3 (rabbit, Jentsch lab)
anti-ClC-4 (rabbit, Jentsch lab)
anti-ClC-5 (rabbit or guinea pig, Jentsch lab)
anti-ClC-6 (rabbit, Jentsch lab)
anti-ClC-7 (rabbit, Jentsch lab)
anti-a3 subunit of H+ ATPase (guinea pig, Jentsch lab)
anti-TMEM206 against the Cterminus (rabbit, Jentsch lab)
anti-TMEM206 against the extracellular loop (rabbit, Jentsch lab)
anti-LRRC8A (rabbit, Jentsch lab)

| Validation | All commercially available antibodies were validated by manufacturers and used in published papers (see manufacturers' websites for immunohistochemistry on western blot examples). |
|---|---|

anti-LAMP1 antibody (rat, BD Pharmingen, 553792) was validated in https://www.citeab.com/antibodies/2410022-553792-bd-pharmingen-purified-rat-anti-mouse-cd107a
anti-EEA1 (mouse, Abcam, ab70521) was validated in https://www.abcam.com/eea1-antibody-1g11-early-endosome-marker-ab70521.html
anti-EEA1 (sheep, R&D Systems, AF8047) was validated in https://www.rndsystems.com/products/human-mouse-rat-eea1-antibody_af8047
anti-Rab5A (mouse, Cell Signaling, 46449) was validated in https://www.cellsignal.com/products/primary-antibodies/rab5a-e6n8s-mouse-mab/46449
anti-Rab5 (mouse, BD Bioscience, BD610725) was validated in https://www.bdbiosciences.com/en-de/products/reagents/microscopy-imaging-reagents/immunofluorescence-reagents/purified-mouse-anti-rab5.610725
anti-Rab7 (mouse, SCBT, sc-376362) was validated in https://www.scbt.com/p/rab-7-antibody-b-3?productCanUrl=rab-7-antibody-b-3&_requestid=10564832
anti-GFP (chicken, Aves lab, GFP-1020) was validated in https://www.aveslabs.com/products/anti-green-fluorescent-protein-antibody-gfp
anti-HA (rabbit, Cell Signaling, 3724) was validated in https://www.cellsignal.de/products/primary-antibodies/ha-tag-c29f4-rabbit-mab/3724
anti-RFP (rabbit, Rockland, 600-401-379) was validated in https://www.rockland.com/categories/primary-antibodies/rfp-antibody-pre-adsorbed-600-401-379/
anti-$\alpha$1 Na/K ATPase clone C464.6 (mouse, Millipore, 05–369) was validated in https://www.merckmillipore.com/DE/de/product/Anti-Na-K-ATPase-1-Antibody-clone-C464.6,MM_NF-05-369?ReferrerURL=https%3A%2F%2Fwww.google.com%2F
anti-actin (mouse, Sigma, A2228) was validated in https://www.sigmaaldrich.com/DE/en/product/sigma/a2228
anti-actin (rabbit, Sigma, A2066) was validated in https://www.sigmaaldrich.com/DE/en/product/sigma/a2066

Newly generated custom-made TMEM206 antibodies were validated with Western Blot analysis and immunocytochemistry using HEK TMEM206 KO cells and tissues/cells from different Tmem206 knock-out mouse lines. All other custom-made antibodies (ClC-2, ClC-3, ClC-4, ClC-5, ClC-6, ClC-7, LRRC8A and a3 subunit of V-ATPase) were previously produced in our lab and were validated according to the field's highest standards by Western Blot analysis and immunohistochemistry on tissues from respective knock-out mouse lines (see methods and reference sections in the manuscript).

# Eukaryotic cell lines

Policy information about cell lines

| Cell line source(s) | Wild-type HEK293, HeLa, HT-1080 and MIA PaCa-2 are from the Deutsche Sammlung von Mikroorganismen und Zellkulturen, Germany. |
|---|---|
| Authentication | Parental cells of all above mentioned cells lines were purchased from the Deutsche Sammlung von Mikroorganismen und Zellkulturen, Germany. |
| Mycoplasma contamination | In all experiments cells were mycoplasma negative. |
| Commonly misidentified lines (See ICLAC register) | No commonly misidentified lines were used. |

# Animals and other organisms

Policy information about studies involving animals; ARRIVE guidelines recommended for reporting animal research

| Laboratory animals | C57BL/6 Mice (Mus musculus) aged of 8-22 weeks (8-16 weeks for BMDMs preparation) and from both sexes were used in this study. The following lines were used : B6;Lrrc8atm2c(EUCOMM)Hmgu-em2(HA)Tjj<br>B6;Lrrc8dem1(tdtomato; loxP)Tjj<br>B6;129/Sv-Clcn2tm1Tjj<br>B6;129/Sv-Clcn3tm1Tjj<br>B6;129/Sv-Clcn4tm1Tjj<br>B6;129/Sv-Clcn5tm1Tjj<br>B6;129/Sv-Clcn7tm3.1Tjj x Cx3cr1CreER x ROSA26floxSTOP-YFP<br>B6;Lrrc8atm2a(EUCOMM)Hmgu x Cx3cr1CreER<br>and the 3 newly generated TMEM206 knock-out lines (as described in the methods section). |
|---|---|
| Wild animals | No wild animals were used. |

| Field-collected samples | No samples were collected from the field. |
| Ethics oversight | All animal experiments and the generation of new mouse lines were approved by Berlin authorities (LAGeSo, G0005/19 licence). |

Note that full information on the approval of the study protocol must also be provided in the manuscript.

