## [Peer Review File · Nature Cell Biology]

Peer Review Information

Journal: Nature Cell Biology

Manuscript Title: Proton-gated anion transport governs endocytic vacuole shrinkage

Corresponding author name(s): Thomas Jentsch

Reviewer Comments & Decisions:

Decision Letter, initial version:

Dear Professor Jentsch,

Thank you for submitting your manuscript, "Proton-gated anion transport governs endocytic vacuole shrinkage", to Nature Cell Biology. It has now been seen by 3 referees, who are experts in channel/ion regulation/pH (referee 1); ion regulation in trafficking (referee 2); and macropinocytosis (referee 3). As you will see from their comments (attached below), they find this work of potential interest, but have raised substantial concerns, which in our view would need to be addressed with considerable revisions before we can consider publication in Nature Cell Biology.

We discussed the referee reports in detail within the editorial team, including the chief editor, and found the reviewers' points valid. We have identified key referee points that should be addressed with priority to strengthen the main new results and ensure technical soundness, as opposed to requests that are overruled as being beyond the scope of the current study. To guide the scope of the revisions, I have listed these points below. We are committed to providing a fair and constructive peer-review process, so please feel free to contact me if you would like to discuss any of the referee comments further. Our typical revision period is six months; however, please let us know if you anticipate delays or issues addressing the reviews, we are happy to further discuss the reviews as needed.

In our view, for resubmission to the journal, it would be essential to:

- A) Please address Rev#2 pt #2: Extending the cell biological analysis of ASOR/TMEM206 function by further dissecting the contribution of ASOR/TMEM206 to macropinosome function in macrophages and macropinocytosis in cells is essential to strengthen the core new cell biological findings.
- B) Please address Rev#2 pt #3 and address the role of TMEM206/ASOR in macropinosome regulation in vivo.
- C) Please address Rev#1 pt #1 by testing whether cancer cells (e.g., cancer cell lines) may also rely

on ASOR/TMEM20 to strengthen the relevance of the results for cell biologists. If the data indicate that cancer cells may use other channels, claims about the universal importance of the channel should be toned down.

D) Please address Rev#3 major point #1 by adding quantitations to demonstrate the robustness of the imaging data.

E) Please address Rev#3 major #2 by providing more discussion and explanation of how the results and those of Osei-Owusu et al. Cell Reports can be understood, which is important for clarity in the field; please also tackle Rev#1's requests for clarification throughout. Other points from Rev#3 should be addressed in the text as well. All other referee concerns pertaining to providing controls, methodological details, clarifications and textual changes should be addressed.

F) Finally please pay close attention to our guidelines on statistical and methodological reporting (listed below) as failure to do so may delay the reconsideration of the revised manuscript. In particular please provide:

We would be happy to consider a revised manuscript that would satisfactorily address these points, unless a similar paper is published elsewhere, or is accepted for publication in Nature Cell Biology in the meantime.

- ensure that it conforms to our format instructions and publication policies (see below and <https://www.nature.com/nature/for-authors>).

- provide a point-by-point rebuttal to the full referee reports verbatim, as provided at the end of this letter.

- provide the completed Reporting Summary (found here <https://www.nature.com/documents/nr-reporting-summary.pdf>). This is essential for reconsideration of the manuscript will be available to editors and referees in the event of peer review. For more information see <http://www.nature.com/authors/policies/availability.html> or contact me.

2When submitting the revised version of your manuscript, please pay close attention to our [href="https://www.nature.com/nature-research/editorial-policies/image-integrity">Digital Image Integrity Guidelines](https://www.nature.com/nature-research/editorial-policies/image-integrity). and to the following points below:

Nature Cell Biology strongly supports public availability of data. Please place the data used in your paper into a public data repository, or alternatively, present the data as Supplementary Information. If data can only be shared on request, please explain why in your Data Availability Statement, and also in the correspondence with your editor. Please note that for some data types, deposition in a public repository is mandatory - more information on our data deposition policies and available repositories appears below.

[REDACTED]

We hope that you will find our referees' comments and editorial guidance helpful. Please do not hesitate to contact me if there is anything you would like to discuss. Thank you again for considering NCB for your work.

Best wishes,

Melina

Melina Casadio, PhD
Senior Editor, Nature Cell Biology
ORCID ID: <https://orcid.org/0000-0003-2389-2243>

Reviewers' Comments:

Reviewer #1:

Remarks to the Author:

In this manuscript the authors identify the ASOR/PAC/TMEM206 proton activated Cl⁻ channel as the previously unknown Cl⁻ conductance essential for macropinosome (MP) resolution. The role of other key transporters and channels, namely CIC-5 and TPC channels, in volume decrease is also clearly delineated, and a mathematical model is constructed that qualitatively matches many of the results including those that are counter-intuitive. My enthusiasm for this work is slightly decreased because the Qiu group at Hopkins recently reported that ASOR localized to the endosomal membrane and plays a role in Cl⁻ conductance (Osei-Owusu et al. Cell Reports 2021). However, they did not explore the role of volume regulation, and their Cell Reports paper, while quite comprehensive, does not involve any mathematical modeling, which significantly strengthens the current manuscript. For these last two reasons, I still find a lot of merit in this current study. In particular, the authors used several functional mutants to better understand all of these regulatory aspects including over expression, E211A CIC-5 uncoupling mutants, ASOR mutants (K319C transport deficient and R87C mutant that exhibits current at neutral pH), and chemical manipulation of ion concentrations (i.e. NH₄Cl, bafilomycin, and Cl⁻ changes). The model results are clearly described and bring the entire interplay of different elements in regulating pH, membrane potential, Cl⁻ concentrations, and volume into a clearer picture. In particular, the R87C ASOR mutant was hypothesized to increase the speed of volume decrease, and it does. Moreover, the rescue of MP formation by the CIC-5 mutant is very nice! It is impressive that the authors carried out these directed studies in a cell and showed that the system was working as expected based on their model. In my mind, this is a very nice piece of cell biology that provides a vetted mechanistic model for how endosomal volume is regulated upon endocytosis.

I have no major comments, only a few minor ones.

1. Before reading this manuscript, I did not know what "MP resolution" meant. Please define this early on.

2. Confusing sentence:

43 Shrinkage of intracellular vesicles cannot occur exclusively by budding of small vesicles
44 because they would need more membrane than available in the parent vesicle.

I had to read this sentence 3 times – what is shrinking – the budding vesicles, the original parent, and why is more membrane needed?

3. Are the authors suggesting here that MP formation can still occur in the absence of ASOR?
102 Disruption of Tmem206 or ion replacement did not affect acute, M-CSF-
103 triggered formation of MPs (Suppl. Fig. 5), as expected from an exclusive role of ASOR in
104 luminal salt loss that is only effective after vesicles have lost their connection to the ...
Is this acute MP resolution ever resolved later by CLC KO or not? I got lost.

4151 However, since ASOR KO induced acidification also in the presence of
152 bafilomycin (Fig. 5d), there must be additional electrogenic acidification mechanisms.
Could this be because you created a very strong negative (inside) membrane potential, and proton
leak ended up acidifying the compartment? Maybe you suggested this, but I didn't get it.

Reviewer #2:

Remarks to the Author:

Zeziulia et al examines the influence of proton-gated anion transport on macropinosome shrinkage in macrophages. The study identifies ASOR/TMEM206 as a chloride channel that regulates macropinosome resolution, which involves macropinosome volume reduction as an aspect of macropinosome maturation. Although other proton-gated chloride channels have been shown to regulate endosomal acidification and receptor-mediated endocytosis (RME; see Osei-Owusu et al., Cell Reports, 2021), the identification of ASOR/TMEM206 as functioning in this process is very novel since the endocytic compartment being analyzed is the macropinosome, which is totally unique, independent, and distinct from RME. The study however has three major issues. First, the manuscript emphasizes that it identifies "the" missing Cl⁻ channel that regulates macropinosome resolution. This is not necessarily the case since the study only includes one macropinocytosis context – macrophages. As the authors note, macropinocytosis has been now explicitly and extensively described in cancer cells. The macropinocytosis pathway is sufficiently unique between immune cells and cancer cells, and one cannot make such broad statements of ASOR/TMEM206 function without the appropriate data to back up the claim. Importantly, the differential tissue expression of the various Cl⁻ channels indicates that different channels might be functioning in different tissues. Second, macropinosomes function in antigen sampling and nutrient acquisition. How macropinosome resolution, and the role that ASOR/TMEM206, impact macropinosome function is not addressed at all. It is not clear that the role of ASOR/TMEM206 in macropinosome maturation is required for any function. Third, there are no efforts to establish the in vivo physiological relevance of ASOR/TMEM206 macropinosome regulation.

Reviewer #3:

Remarks to the Author:

The manuscript by Zeziulia, et al., analyzes the roles of membrane channels, pH and Cl⁻ transporters in the process of macropinosome shrinkage in macrophages. Macropinosome shrinkage is necessary for membrane recycling to the plasma membrane and for continuous ingestion of extracellular solutes. It was already established that macropinosome shrinkage requires loss of the major osmotic species engulfed into macropinosomes: Na⁺, which exits macropinosome lumens via TPC channels, and Cl⁻ (Freeman, et al., Science, 2020). Using various methods, the work demonstrates that the egress of Cl⁻ from macropinosomes into cytoplasm is mediated by the chloride channel TMEM206, which had been previously shown to require low pH and membrane depolarization to transport chloride. TMEM206 localized to macropinosomes, and deletion of TMEM206 inhibited macropinosome shrinkage. Cl⁻/H⁺ exchange proteins made little or no contributions to shrinkage. The model supported by the

5experiments and by a mathematical model of ion transport across macropinosome membranes is that TMEM206 in the plasma membrane is internalized into forming macropinosomes, where subsequent membrane depolarization by loss of Na⁺, and acidification of the lumen by the Vacuolar-ATPase or the Cl⁻/H⁺ exchange protein CLC-5 permits Cl⁻ efflux into cytoplasm, with attendant vacuole shrinkage. This is an important conceptual advance which is well supported by the evidence presented here. A recent study by Osei-Owusu, et al. (Cell Reports 2021. 34: 108683) demonstrated roles for endocytosed TMEM206 (referred to as PAC) in the regulation of endosomal pH, with consequences for transferrin trafficking in early endosomes. That work showed that endosome acidification by the V-ATPase and Cl⁻/H⁺ exchange by CLC proteins would increase luminal Cl⁻ concentrations, and the activity of TMEM206 at low pH removed the accumulating Cl⁻; hence, TMEM206 regulated endosomal pH and Cl⁻ concentration. The roles for TMEM206 in endosomal volume regulation were not considered, largely because most of the studies were of endosomes formed by receptor-mediated endocytosis, whose volume is negligible compared to macropinosomes. Based on these two studies, this reviewer considers the role of TMEM206 in endosomal volume regulation to be more significant than its role in pH regulation. Also, the combination of modeling and experiments in the present manuscript results in a more robust model of the functions of TMEM206, CLC-5, V-ATPase and TPC channels in the regulation of pH, Cl⁻, pH and membrane potential in endocytic vesicles, including macropinosomes.

With the exception noted below, the data and methodology support the conclusions and the statistical tests are appropriate. The writing is clear.

Concerns:

1. The morphology should be quantified. The fluorescence images look acceptable, but no measures of colocalization are provided to support the claims. The methods state how many times the experiments were performed (Supplementary methods, line 95), but the manuscript does not provide numbers to support the claims about colocalization or the lack thereof. Presently, the presence of the color yellow in the fluorescence micrographs is the only indication of colocalization of green and red fluorophores. Statistical tests of quantitative measurements should be provided to support the claims.
2. The discussion should compare and contrast this study with that of Osei-Owusu, with particular emphasis on the roles for TMEM206 in the regulation of pH, Cl⁻ and volume.
3. The authors might also consider the role of osmotic pressure differences between macropinosomes and interacting lysosomes in the mechanism of macropinosome shrinkage. Previous studies suggest that rapid macropinosome shrinkage may be mediated by transient fusion between macropinosomes and the tubular lysosomal compartment, whose osmotic pressure may be less than that of the distended macropinosomes (Yoshida, et al., 2015. J. Cell Biol. 211: 1590-172; videos). The mechanisms that dehydrate lysosomes may create a negative hydrostatic pressure that drains macropinosomes after fusion between the organelles.
4. The channel is referred to as ASOR or TMEM206, with no obvious rationale for using one acronym or the other. For clarity, the manuscript should use one term consistently throughout; TMEM206 seems most appropriate, as some names of mutant proteins include TMEM206.

Joel Swanson

AUTHOR CONTRIBUTIONS – must be included after the Acknowledgements, detailing the contributions of each author to the paper (e.g. experimental work, project planning, data analysis etc.). Each author

7should be listed by his/her initials.

Methods should be written concisely, but should contain all elements necessary to allow interpretation and replication of the results. As a guideline, Methods sections typically do not exceed 3,000 words. The Methods should be divided into subsections listing reagents and techniques. When citing previous methods, accurate references should be provided and any alterations should be noted. Information must be provided about: antibody dilutions, company names, catalogue numbers and clone numbers for monoclonal antibodies; sequences of RNAi and cDNA probes/primers or company names and catalogue numbers if reagents are commercial; cell line names, sources and information on cell line identity and authentication. Animal studies and experiments involving human subjects must be reported in detail, identifying the committees approving the protocols. For studies involving human subjects/samples, a statement must be included confirming that informed consent was obtained. Statistical analyses and information on the reproducibility of experimental results should be provided in a section titled "Statistics and Reproducibility".

All Nature Cell Biology manuscripts submitted on or after March 21 2016 must include a Data availability statement as a separate section after Methods but before references, under the heading "Data Availability". . For Springer Nature policies on data availability see <http://www.nature.com/authors/policies/availability.html>; for more information on this particular policy see <http://www.nature.com/authors/policies/data/data-availability-statements-data-citations.pdf>. The Data availability statement should include:

- Accession codes for primary datasets (generated during the study under consideration and

designated as "primary accessions") and secondary datasets (published datasets reanalysed during the study under consideration, designated as "referenced accessions"). For primary accessions data should be made public to coincide with publication of the manuscript. A list of data types for which submission to community-endorsed public repositories is mandated (including sequence, structure, microarray, deep sequencing data) can be found here <http://www.nature.com/authors/policies/availability.html#data>.

- Unique identifiers (accession codes, DOIs or other unique persistent identifier) and hyperlinks for datasets deposited in an approved repository, but for which data deposition is not mandated (see here for details <http://www.nature.com/sdata/data-policies/repositories>).
- At a minimum, please include a statement confirming that all relevant data are available from the authors, and/or are included with the manuscript (e.g. as source data or supplementary information), listing which data are included (e.g. by figure panels and data types) and mentioning any restrictions on availability.
- If a dataset has a Digital Object Identifier (DOI) as its unique identifier, we strongly encourage including this in the Reference list and citing the dataset in the Methods.

We recommend that you upload the step-by-step protocols used in this manuscript to the Protocol Exchange. More details can be found at www.nature.com/protocolexchange/about.

All imaging data should be accompanied by scale bars, which should be defined in the legend. Cropped images of gels/blots are acceptable, but need to be accompanied by size markers, and to retain visible background signal within the linear range (i.e. should not be saturated). The boundaries of panels with low background have to be demarked with black lines. Splicing of panels should only be considered if unavoidable, and must be clearly marked on the figure, and noted in the legend with a statement on whether the samples were obtained and processed simultaneously. Quantitative comparisons between samples on different gels/blots are discouraged; if this is unavoidable, it should only be performed for samples derived from the same experiment with gels/blots were processed in parallel, which needs to be stated in the legend.

Figures should be provided at approximately the size that they are to be printed at (single column is 86 mm, double column is 170 mm) and should not exceed an A4 page (8.5 x 11"). Reduction to the scale that will be used on the page is not necessary, but multi-panel figures should be sized so that

9the whole figure can be reduced by the same amount at the smallest size at which essential details in each panel are visible. In the interest of our colour-blind readers we ask that you avoid using red and green for contrast in figures. Replacing red with magenta and green with turquoise are two possible colour-safe alternatives. Lines with widths of less than 1 point should be avoided. Sans serif typefaces, such as Helvetica (preferred) or Arial should be used. All text that forms part of a figure should be rewritable and removable.

TABLES – main tables should be provided as individual Word files, together with a brief title and

10legend. For supplementary tables see below.

The total number of Supplementary Figures (not including the “unprocessed scans” Supplementary Figure) should not exceed the number of main display items (figures and/or tables (see our Guide to Authors and March 2012 editorial <http://www.nature.com/ncb/authors/submit/index.html#suppinfo>; <http://www.nature.com/ncb/journal/v14/n3/index.html#ed>). No restrictions apply to Supplementary Tables or Videos, but we advise authors to be selective in including supplemental data.

GUIDELINES FOR EXPERIMENTAL AND STATISTICAL REPORTING

REPORTING REQUIREMENTS – We are trying to improve the quality of methods and statistics reporting in our papers. To that end, we are now asking authors to complete a reporting summary that collects information on experimental design and reagents. The Reporting Summary can be found here <https://www.nature.com/documents/nr-reporting-summary.pdf>) If you would like to reference the guidance text as you complete the template, please access these flattened versions

11at <http://www.nature.com/authors/policies/availability.html>.

Author Rebuttal to Initial comments

Detailed response to reviewers

Zeziulia *et al*: Proton-gated anion transport governs endocytic vacuole shrinkage

We thank all three reviewers for their insightful comments, which overall were very positive. We have carefully considered their suggestions, have performed several new experiments, and have changed our manuscript accordingly. In the following, please find detailed responses to their points.

Reviewer #1:

Remarks to the Author:

In this manuscript the authors identify the ASOR/PAC/TMEM206 proton activated Cl⁻ channel as the

12previously unknown Cl⁻ conductance essential for macropinosome (MP) resolution. The role of other key transporters and channels, namely CIC-5 and TPC channels, in volume decrease is also clearly delineated, and a mathematical model is constructed that qualitatively matches many of the results including those that are counter-intuitive. My enthusiasm for this work is slightly decreased because the Qiu group at Hopkins recently reported that ASOR localized to the endosomal membrane and plays a role in Cl⁻ conductance (Osei-Owusu et al. Cell Reports 2021). However, they did not explore the role of volume regulation, and their Cell Reports paper, while quite comprehensive, does not involve any mathematical modeling, which significantly strengthens the current manuscript. For these last two reasons, I still find a lot of merit in this current study. In particular, the authors used several functional mutants to better understand all of these regulatory aspects including over expression, E211A CIC-5 uncoupling mutants, ASOR mutants (K319C transport deficient and R87C mutant that exhibits current at neutral pH), and chemical manipulation of ion concentrations (i.e. NH₄Cl, bafilomycin, and Cl⁻ changes). The model results are clearly described and bring the entire interplay of different elements in regulating pH, membrane potential, Cl⁻ concentrations, and volume into a clearer picture. In particular, the R87C ASOR mutant was hypothesized to increase the speed of volume decrease, and it does. Moreover, the rescue of MP formation by the CIC-5 mutant is very nice! It is impressive that the authors carried out these directed studies in a cell and showed that the system was working as expected based on their model. In my mind, this is a very nice piece of cell biology that provides a vetted mechanistic model for how endosomal volume is regulated upon endocytosis.

We thank the reviewer for appreciating the importance and novelty of our paper.

I have no major comments, only a few minor ones.

1. Before reading this manuscript, I did not know what “MP resolution” meant. Please define this early on.

We have now mentioned this now in the first main text paragraph, line 49-50, where we now put ‘...The shrinkage (‘resolution’) of the large vacuoles generated by macropinocytosis ...’

2. Confusing sentence:

43 Shrinkage of intracellular vesicles cannot occur exclusively by budding of small vesicles

44 because they would need more membrane than available in the parent vesicle.

I had to read this sentence 3 times – what is shrinking – the budding vesicles, the original parent, and why is more membrane needed?

Sorry for the explanation that we shortened too much to conform to word limits of the journal. The parent vesicle is shrinking, but budding of vesicles (with the limited membrane available) cannot carry all of the originally contained volume because of geometry (surface versus volume of spheres). We have now changed the wording to (lines 45-49: 'With the exception of tubular structures, reduction of vesicle size cannot occur solely by the release of smaller vesicles because vesicle volume depends on the third power, and surface area on the square, of the radius. Although small amounts of luminal fluid might be taken up by tubular endolysosomes in 'kiss-and-run' processes¹, decrease of vesicle volume requires transmembrane water flux that is driven by osmotic gradients..')

3. Are the authors suggesting here that MP formation can still occur in the absence of ASOR? 102 Disruption of Tmem206 or ion replacement did not affect acute, M-CSF- 103 triggered formation of MPs (Suppl. Fig. 5), as expected from an exclusive role of ASOR in 104 luminal salt loss that is only effective after vesicles have lost their connection to the ... Is this acute MP resolution ever resolved later by CLC KO or not? I got lost.

Yes, whereas macropinosome resolution (shrinkage) is strongly impaired by ASOR KO, the initial formation of macropinosomes from the plasma membrane is not affected by the KO. And, as stated, this is expected: ASOR affects MP resolution by mediating the efflux of Cl⁻ from MPs (accompanied by Na⁺ efflux through TPCs), leading to osmotic shrinkage. As long as the lumen of the nascent MP is still connected to the extracellular space no osmotic shrinkage can occur. Moreover, voltage- and pH-dependent ASOR will be closed under these conditions as the voltage of the nascent MP will be almost identical to the negative-inside plasma membrane voltage, and the 'luminal' pH will be very close to extracellular pH – both pH and voltage will keep ASOR shut. Only later, ASOR will open due to TPC-mediated Na⁺-efflux and ClC-5 and H⁺-ATPase mediated acidification.

In an attempt to save words, our sentence was probably too short to be easily understood. We therefore expanded it to (lines 115-120: 'Disruption of Tmem206 or ion replacement did not affect acute, M-CSF-triggered formation of MPs (Suppl. Fig. 5). This is expected from ASOR's role in luminal salt loss and ensuing osmotic shrinkage, which is only effective after the vesicle lumen has lost its connection to the extracellular medium, and from the voltage- and pH-dependence of ASOR that shuts it down at the plasma membrane.'

Concerning the last question: As stated in the paper, ClC-5 KO might have a moderate impact on MP resolution at early time points, but not at late ones (as it soon reaches thermodynamic equilibrium), see e.g. Fig. 5f, in contrast to the very strong effect of Tmem206 KO. Please bear in mind that we cannot measure at the earliest time points after M-CSF application for technical reasons, where the effect of ClC-5 is predicted to be largest, as stated e.g. in the legend to Fig. 5.

151 However, since ASOR KO induced acidification also in the presence of
152 bafilomycin (Fig. 5d), there must be additional electrogenic acidification mechanisms.
Could this be because you created a very strong negative (inside) membrane potential, and proton leak
ended up acidifying the compartment? Maybe you suggested this, but I didn't get it.

The reviewer is right, a proton 'leak' is a distinct possibility, as first discussed in line 202 'Theoretically, even an H⁺-conductance may acidify vacuoles by up to 1 pH unit when they are depolarized by TPCs...' Further down, after having introduced our model, we show that this is entirely feasible in mathematical modelling in lines 245-253: ' Even a model vesicle containing only an H⁺-conductance together with TPC and ASOR showed marked shrinkage, albeit with a slightly lower rate that can be attributed to reduced acidification (Suppl. Fig. 16j). A hint for such a conductance comes from the acidification upon luminal Cl⁻ removal (Fig. 5e). It cannot be caused by anion/proton exchangers (Suppl. Fig. 16h), but can be explained by the presence of either a V-type-ATPases or an H⁺-conductance (Suppl. Fig. 16i,j) that respond to the more negative luminal potential with increased H⁺ transport. The H⁺ conductance might be embodied by the Hv1 H⁺-channel^{38,39} which, reminiscent of ASOR and CIC-5, is strongly outwardly rectifying. However, it is unclear whether it is present on MPs.'

Reviewer #2:

Remarks to the Author:

Zeziulia et al examines the influence of proton-gated anion transport on macropinosome shrinkage in macrophages. The study identifies ASOR/TMEM206 as a chloride channel that regulates macropinosome resolution, which involves macropinosome volume reduction as an aspect of macropinosome maturation. Although other proton-gated chloride channels have been shown to regulate endosomal acidification and receptor-mediated endocytosis (RME; see Osei-Owusu et al., Cell Reports, 2021), the identification of ASOR/TMEM206 as functioning in this process is very novel since the endocytic compartment being analyzed is the macropinosome, which is totally unique, independent, and distinct from RME.

We thank the reviewer for appreciating the novelty of our findings.

The study however has three major issues. First, the manuscript emphasizes that it identifies “the” missing Cl⁻ channel that regulates macropinosome resolution. This is not necessarily the case since the study only includes one macropinosome context – macrophages. As the authors note, macropinosome has been now explicitly and extensively described in cancer cells. The

15macropinocytosis pathway is sufficiently unique between immune cells and cancer cells, and one cannot make such broad statements of ASOR/TMEM206 function without the appropriate data to back up the claim. Importantly, the differential tissue expression of the various Cl⁻ channels indicates that different channels might be functioning in different tissues.

We totally agree with the reviewer that it is, even in principle, impossible to rigorously conclude that ASOR/TMEM106 is 'the' missing chloride channel in all different cell types. To accommodate the wish of the reviewer we have, on the one hand, changed the abstract (lines 29-34) to: 'Shrinkage of macrophage macropinosomes depends on Na⁺ efflux through TPC channels and Cl⁻ exit through unknown channels. Relieving osmotic pressure facilitates vesicle budding, positioning osmotic shrinkage upstream of vesicular sorting and trafficking. We now identify the missing Cl⁻ channel as ASOR/TMEM206, a ubiquitously expressed proton-activated Cl⁻ channel involved in acid-induced cell death and stroke', which restricts our statement to macrophages in which most of our experiments were done.

On the other hand, we have performed new experiments to address the important question raised by the reviewer whether ASOR also has a role in MP resolution in cancer cells. We used human HT-1080 fibrosarcoma cells in which macropinocytosis can be stimulated by a short pulse of EGF. Also in these cells, we found that ASOR is crucial for MP resolution. TMEM206 KD with siRNA led to markedly reduced MP shrinkage as evident both from MP size and TMR-dextran fluorescence measurements. This is now shown in Fig. 4d-f and mentioned in lines 107-109 of the main text. We believe that ASOR plays a role in MP shrinkage in almost all cells which display macropinocytosis – not only because of its nearly ubiquitous expression pattern, but also because its localization, its inactivity at the plasma membrane and its activation by acidic pH and inside-positive voltage. These properties fit extremely well to this role.

Second, macropinosomes function in antigen sampling and nutrient acquisition. How macropinosome resolution, and the role that ASOR/TMEM206, impact macropinosome function is not addressed at all. It is not clear that the role of ASOR/TMEM206 in macropinosome maturation is required for any function.

In response to this important comment, we have performed new experiments studying the effect of ASOR on nutrient acquisition of tumor cells (A) and on migration of macrophages (B).

(A) *The supply of amino-acids derived from macropinocytosed extracellular proteins to nutrient-starved cancer cells is receiving increasing attention in the scientific and medical communities. A well-established model system are MIA PaCa-2 pancreatic cancer cells in which mutated KRAS (which is mutated in the majority of pancreatic cancers) leads to increased rates of constitutive macropinocytosis. We generated TMEM206 KO MIA-PaCa-2 cells by CRISPR-Cas9 (with two different sgRNAs to avoid effects of off-target effects) and compared the growth of KO and WT cells under amino-acid starvation with and without addition of 3% BSA. We used three different KO clones and 'WT' clones (which were 'mock-edited' using*

the same sgRNAs, but which displayed WT TMEM206 alleles) to avoid artifacts of clonal selection. In all cases, *TMEM206*^{-/-} cells survived/grew better than WT cells in the presence of BSA (new Figure 6e). This is compatible with (1) identical rate of macropinosome formation (see Suppl. Fig. 5) and (2) reduced recycling of BSA back to the surface when MP resolution is impaired by TMEM206 disruption.

These results suggest that decrease in ASOR expression levels might be correlated with better tumor growth in vivo. Indeed, the TCGA database as presented in the Human Protein Atlas reveals that decreased TMEM206 levels are (weakly) correlated with decreased life expectancy of patients with pancreatic tumors, which we shortly mention in the text.

Figure for the reviewer taken from the Protein Atlas for survival of patients with pancreatic tumors stratified according to TMEM206 (PACC1) expression levels. Patients with lower TMEM206 expression levels in tumors die on average somewhat earlier.

These important new results are now presented and discussed on lines 291-307 of the revised manuscript and are also mentioned shortly in the abstract and the final paragraph of the paper.

(B) A physiological role of surface receptor recycling is also suggested by a newly added series of experiments studying BMDM migration. In a scratch assay, we compared migration of WT and *Tmem260* KO BMDMs in the absence and presence of the C5a chemoattractant. ASOR disruption moderately reduced the enhance migration of BMDMs. These results are compatible with reduced recycling of the C5a receptor back to the plasma membrane. This is now shown in Fig. 6d and described on lines 281-290 in the manuscript.

Both experiments show important effects of ASOR on cellular functions, both of which are related to ASOR's role in MP resolution.

Third, there are no efforts to establish the in vivo physiological relevance of ASOR/TMEM206 macropinosome regulation.

We agree that it will be interesting to investigate the importance of ASOR in macropinosome biology in vivo, which we plan to do in the future using our KO mice (e.g. concerning antigen processing). However, addressing these issues will take more than a year of focused experimental effort, in addition to the marked delay of obtaining German animal experimentation permits and breeding enough animals.

We are confident that the new above-mentioned experiments on tumor growth and cell migration demonstrate important and medically relevant facets of biological roles of ASOR. We believe that more aspects will be discovered in whole animal studies, but these are beyond the scope of our work.

Reviewer #3:

Remarks to the Author:

The manuscript by Zeziulia, et al., analyzes the roles of membrane channels, pH and Cl⁻ transporters in the process of macropinosome shrinkage in macrophages. Macropinosome shrinkage is necessary for membrane recycling to the plasma membrane and for continuous ingestion of extracellular solutes. It was already established that macropinosome shrinkage requires loss of the major osmotic species engulfed into macropinosomes: Na⁺, which exits macropinosome lumens via TPC channels, and Cl⁻ (Freeman, et al., Science, 2020). Using various methods, the work demonstrates that the egress of Cl⁻ from macropinosomes into cytoplasm is mediated by the chloride channel TMEM206, which had been previously shown to require low pH and membrane depolarization to transport chloride. TMEM206 localized to macropinosomes, and deletion of TMEM206 inhibited macropinosome shrinkage. Cl⁻/H⁺ exchange proteins made little or no contributions to shrinkage. The model supported by the experiments and by a mathematical model of ion transport across macropinosome membranes is that TMEM206 in the plasma membrane is internalized into forming macropinosomes, where subsequent membrane depolarization by loss of Na⁺, and acidification of the lumen by the Vacuolar-ATPase or the Cl⁻/H⁺ exchange protein CLC-5 permits Cl⁻ efflux into cytoplasm, with attendant vacuole shrinkage. This is an important conceptual advance which is well supported by the evidence presented here.

A recent study by Osei-Owusu, et al. (Cell Reports 2021. 34: 108683) demonstrated roles for endocytosed TMEM206 (referred to as PAC) in the regulation of endosomal pH, with consequences for transferrin trafficking in early endosomes. That work showed that endosome acidification by the V-ATPase and Cl⁻/H⁺ exchange by CLC proteins would increase luminal Cl⁻ concentrations, and the activity of TMEM206 at low pH removed the accumulating Cl⁻; hence, TMEM206 regulated endosomal pH and Cl⁻ concentration. The roles for TMEM206 in endosomal volume regulation were not considered, largely because most of the studies were of endosomes formed by receptor-mediated endocytosis, whose

18volume is negligible compared to macropinosomes. Based on these two studies, this reviewer considers the role of TMEM206 in endosomal volume regulation to be more significant than its role in pH regulation. Also, the combination of modeling and experiments in the present manuscript results in a more robust model of the functions of TMEM206, CLC-5, V-ATPase and TPC channels in the regulation of pH, Cl⁻, pH and membrane potential in endocytic vesicles, including macropinosomes. With the exception noted below, the data and methodology support the conclusions and the statistical tests are appropriate. The writing is clear.

We thank the reviewer for the very positive comments on our manuscript.

Concerns:

1. The morphology should be quantified. The fluorescence images look acceptable, but no measures of colocalization are provided to support the claims. The methods state how many times the experiments were performed (Supplementary methods, line 95), but the manuscript does not provide numbers to support the claims about colocalization or the lack thereof. Presently, the presence of the color yellow in the fluorescence micrographs is the only indication of colocalization of green and red fluorophores. Statistical tests of quantitative measurements should be provided to support the claims.

We have now performed a quantitative analysis of the co-localization of different proteins, using Pearson coefficients, which sometimes required new IF studies. We show these values, including their scatter and mean values, next to the figure panels where such an evaluation makes sense and is important for the conclusion of the paper. This quantitative evaluation fully confirms our conclusions. However, Pearson coefficients are based on the presence of two IF signals in individual pixels, but this method does not consider the presence of individual proteins on identifiable objects such as macropinosomes. Hence, if two proteins are clustered or otherwise non-homogenously distributed over the MP membrane, as is often the case, Pearson coefficients will systematically underestimate the 'real' co-localization on vesicles, as they report co-localization of proteins (within the optical resolution limit). Pearson coefficients also yield wrong results of labeling of different markers, located on different compartments, is very dense, leading to false positive results. For a thorough discussion of methods to quantify co-localization and their pitfalls see e.g. Dunn et al., Am J Physiol. 300, C723 (2011) doi:10.1152/ajpcell.00462.2010.

We have added this quantitative evaluation to Fig. 2c,d, Fig. 3a,b, Fig. 5 j,k, Suppl. Fig. 2a,c, Suppl. Fig. 7b,c, Suppl. Fig. 8b,c, Suppl. Fig. 11a,b,c, Suppl. Fig. 12c,d,e.

192. The discussion should compare and contrast this study with that of Osei-Owusu, with particular emphasis on the roles for TMEM206 in the regulation of pH, Cl⁻ and volume.

We already had compared our study on macropinosomes to that of Osei-Owusu that is concerned with endosomes, highlighting the difference in initial luminal Cl⁻ concentration between these compartments. Whereas initial ion concentrations in the large macropinosomes correspond to those of the extracellular medium, the much smaller endosomes have much lower chloride concentrations due to the negative surface charges of the lipid bilayer (the concentrations have been measured by A. Verkman, and our unpublished calculations taking literature values for surface charges show that this is feasible). Hence, endosomes take up, rather than release, Cl⁻ during their progression towards lysosomes, a process that involve CLCs and probably ASOR. We now compare our data to those of Osei-Owusu et al. in a slightly changed paragraph in Discussion (lines 313-332):

‘In addition to macropinosomes, ASOR/TMEM206 is found on endosomes. Similar to our results for MPs, endosomes were more acidic in *Tmem206*^{-/-} HEK cells²¹. Transferrin endocytosis was increased in *Tmem206*^{-/-} cells owing to increased transferrin receptor recycling²¹, arguing against the hypothesis that ASOR-mediated endosome shrinkage increases endocytic trafficking. An important difference between macropinosomes and endosomes is their initial luminal ion composition. Endosomal Cl⁻ concentration is initially low (≈ 15 mM) in several cell types and progressively increases over time in parallel to acidification^{49,50}. This increase depends on ClC-5⁴⁹ and ClC-3⁵⁰ for early and late endosomes, respectively. The low initial Cl⁻ concentration was attributed to negative surface charges that can decrease luminal anion concentrations in small endosomes, but not in vastly larger macropinosomes. Unlike macropinosomes, endosomes may accumulate Cl⁻ into their lumen through both CLCs^{49,50} and ASOR in a process driven by proton pumps. Model calculations⁵¹ and analysis of mice expressing mutant CLCs mutated into uncoupled Cl⁻ conductors^{26,51} show that both Cl⁻ channels and 2Cl⁻/H⁺-exchangers support proton pump-driven acidification and luminal Cl⁻-accumulation, with exchangers being more efficient than channels. The described effect²¹ of ASOR KO on endosomal pH and Cl⁻ may reflect the higher efficiency of CLCs versus Cl⁻ channels in active acidification⁵¹ rather than luminal Cl⁻ loss. Once endosomes have acquired sufficiently high luminal [Cl⁻], they might use ASOR and TPCs for shrinkage and vesicle budding as found here with MPs.’

3. The authors might also consider the role of osmotic pressure differences between macropinosomes and interacting lysosomes in the mechanism of macropinosome shrinkage. Previous studies suggest that

20rapid macropinosome shrinkage may be mediated by transient fusion between macropinosomes and the tubular lysosomal compartment, whose osmotic pressure may be less than that of the distended macropinosomes (Yoshida, et al., 2015. J. Cell Biol. 211: 1590-172; videos). The mechanisms that dehydrate lysosomes may create a negative hydrostatic pressure that drains macropinosomes after fusion between the organelles.

Thank you for the interesting suggestion that ‘piranhalysis’ might participate to some degree in the resolution of macropinosomes. During the first 10 min or so, the time frame in the spot light of our study, fluid exchange with lysosomes will not be prominent, but rather contacts with endosomes which will also deliver membrane transport proteins such TPCs, proton pumps, etc during MP maturation. We can certainly not exclude that some of the MP fluid escapes in these ‘kiss and run’ contacts with tubular endolysosomes (which can swell due to their shape), which may work in parallel to the tubulation / budding processes by which MPs shed their membranes (together with fluid, but less than needed because of surface/volume considerations). In any case, our study focuses on the importance of ion transport across the limiting membranes of MPs and strongly suggests that this transport, together with resulting osmotic gradients, is crucial for MP resolution.

We now mention this interesting aspect in the first paragraph (lines 47-49) : ‘Although small amounts of luminal fluid might be taken up by tubular endolysosomes in ‘kiss-and-run’ processes¹, decrease of vesicle volume requires transmembrane water flux that is driven by osmotic gradients’, with ref. 1 referring to the Yoshida et al paper.

4. The channel is referred to as ASOR or TMEM206, with no obvious rationale for using one acronym or the other. For clarity, the manuscript should use one term consistently throughout; TMEM206 seems most appropriate, as some names of mutant proteins include TMEM206.

Joel Swanson

This is a somewhat tricky point as by now four names have been proposed for the same channel. PAORAC and ASOR have been used before its molecular identification, with ASOR being more often used. TMEM206 was the name for the gene and protein before it was clear that it constitutes the ASOR/PAORAC channel, and Zhaozhu Qiu gave it a new, fourth name (PAC for proton activated channel). We prefer to name it both ASOR and TMEM206, often together as ASOR/TMEM206, to enable reference to previous literature. When referring to KO cell lines, we prefer to use the name of the gene (Tmem206), as suggested by the reviewer, whereas we preferentially use the name ASOR for the channel. For indexing purposes, we have mentioned that the channel is also known as PAORAC or PAC.

Other changes

*We have added a new experiment that had not been requested by the reviewers, but which is a valuable addition, as we believe. ASOR contains an N-terminal Y-based endocytosis motif that has been reported to have a role in targeting ASOR from the plasma membrane to endosomes. We asked whether this motif is crucial for targeting ASOR to macropinosomes – in that case, we expect that a mutant lacking this endocytic motif does not rescue macropinosome resolution of *Tmem206*^{-/-} BMDMs. However, rescue appeared as efficient as with WT ASOR, demonstrating that this endocytosis motif has no important role for the formation and function of macropinosomes. We show this new experiment in Fig. 4h and mention this result in lines 114-115.*

Decision Letter, first revision:

Dear Professor Jentsch,

Thank you for submitting your revised manuscript, "Proton-gated anion transport governs endocytic vacuole shrinkage", to Nature Cell Biology. The revision has now been seen by the original Reviewers #2 and #3 (Rev#3 also weighed in on how Rev#1's comments were addressed). As you will see from their comments (attached below), while they continue to find the work of interest, some issues remain. Although we are also very interested in this study, we believe that their concerns should be addressed before we can consider publication in Nature Cell Biology.

As per our standard process, we discussed the remaining comments from Rev#2 within the editorial team at NCB in-depth, including with our Chief Editor.

We find the remaining comments from Rev#2 consistent with the points of major concern raised during the first round of review, and we agree with the reviewer that additional controls and functional analyses - at a minimum in vitro - are needed to bolster the claims related to ASOR/TMEM206's role in macropinosome resolution.

Per journal policy, we strive to limit all papers to one round of major experimental revision, and given that the reviewer's comments are relatively minor, we are open to a final round of revision. Please note that we would only consider a revision if it can definitively and convincingly establish the functional importance of ASOR/TMEM206 macropinosome regulation in macrophages, as delineated by the reviewer in their comments. In other words, we do not encourage resubmission without strong new data to address the reviewer's comments in full. Please let us know if you would like to discuss the reviews further or anticipate any issues addressing the reviews.

As before, prior to resubmission, please pay close attention to our guidelines on statistical and methodological reporting (listed below) as failure to do so may delay the reconsideration of the

22revised manuscript. In particular please provide:

- a Supplementary Figure including unprocessed images of all gels/blots in the form of a multi-page pdf file. Please ensure that blots/gels are labeled and the sections presented in the figures are clearly indicated.
- a Supplementary Table including all numerical source data in Excel format, with data for different figures provided as different sheets within a single Excel file. The file should include source data giving rise to graphical representations and statistical descriptions in the paper and for all instances where the figures present representative experiments of multiple independent repeats, the source data of all repeats should be provided.

We therefore invite you to take these points into account when revising the manuscript. In addition, when preparing the revision please:

- ensure that it conforms to our format instructions and publication policies (see below and www.nature.com/nature/authors/).
- provide a point-by-point rebuttal to the full referee reports verbatim, as provided at the end of this letter.
- provide the completed Editorial Policy Checklist (found here <https://www.nature.com/authors/policies/Policy.pdf>), and Reporting Summary (found here <https://www.nature.com/authors/policies/ReportingSummary.pdf>). This is essential for reconsideration of the manuscript and these documents will be available to editors and referees in the event of peer review. For more information see <http://www.nature.com/authors/policies/availability.html> or contact me.

Nature Cell Biology is committed to improving transparency in authorship. As part of our efforts in this direction, we are now requesting that all authors identified as 'corresponding author' on published papers create and link their Open Researcher and Contributor Identifier (ORCID) with their account on the Manuscript Tracking System (MTS), prior to acceptance. ORCID helps the scientific community achieve unambiguous attribution of all scholarly contributions. You can create and link your ORCID from the home page of the MTS by clicking on 'Modify my Springer Nature account'. For more information please visit www.springernature.com/orcid.

[REDACTED]

23We would like to receive the revision within four - eight weeks. If submitted within this time period, reconsideration of the revised manuscript will not be affected by related studies published elsewhere or accepted for publication in Nature Cell Biology in the meantime. We would be happy to consider a revision even after this timeframe, but in that case we will consider the published literature at the time of resubmission when assessing the file.

We hope that you will find our referees' comments and editorial guidance helpful. Please do not hesitate to contact me if there is anything you would like to discuss. Thank you again for considering NCB for your work.

Best wishes,

Melina

Melina Casadio, PhD
Senior Editor, Nature Cell Biology
ORCID ID: <https://orcid.org/0000-0003-2389-2243>

Reviewers' Comments:

Reviewer #2:

Remarks to the Author:

In the revised manuscript, Zeziulia et al have made text modifications and have added new data to the manuscript. In the response to reviewers, they have also provided additional data that pertain to some of the specific reviewer critiques. While the efforts to improve the manuscript are applauded, there are still some remaining concerns that should be considered:

1. To modify some of the claims in the manuscript, the authors have amended the abstract in an attempt to restrict the statements to macrophages; however, the modified statements are still somewhat misleading. Specifically, the following statement is misleading: "We now identify the missing Cl⁻ channel as ASOR/TMEM206, a ubiquitously expressed proton-activated Cl⁻ channel involved in acid-induced cell death and stroke." Modifying this statement to include "in macrophages" would be more accurate and a better reflection of the data in the manuscript. The new data in the HT-1080 cells support this change. The fact that TMEM206 functions in these cells in macropinosome shrinkage broadens the scope of the work; however, it is quite evident that TMEM206 knockdown is not as effective in the HT-1080 cells as it is in the macrophages (comparing Fig. 4d/e to Fig. 4a/b). Based on these new experiments, it seems that either Cl⁻ channels are only partially required for macropinosome resolution in cancer cells or Cl⁻ channels other than TMEM206 are playing a role in the cancer cells. Additionally, these experiments seemingly still only include a single targeting siRNA.
2. To address the functionality, instead of using a macrophage-based assay, the authors have provided a cancer cell assay where viability is assessed (Fig. 6e). In this new experiment, TMEM206

24KO in MIA PaCa-2 cells results in enhanced viability in the presence of BSA. These data are not explained mechanistically, but the authors speculate that it could relate to recycling of BSA. The authors also claim that this is in line with the initial macropinosome initiation steps being unaffected by TMEM206 in macrophages (Suppl Fig. 5). There are several unresolved issues with these data. TMEM206 having a role in macropinosome shrinkage in MIA PaCa-2 cells is not shown. This should be demonstrated, or the viability assays should be done in the HT-1080 cells. The results of the viability data suggests that TMEM206 has no role in preventing acidification of macropinosomes or fusion with the lysosome, which is a critical step in macropinosome function at least in cancer cells. This is not assessed, which could have been done with a simple DQ-BSA assay. Also, the data suggest that BSA utilization for survival is actually increased, which could be linked to either more efficient BSA degradation or an effect on recycling. Both of these scenarios are very easy to test. It is not clear how the BMDM migration assays relate to macropinosome resolution, as those data could be related to other functions of ASOR.

3. It is fair to argue that establishing the *in vivo* physiological relevance could be time-consuming and costly. If this lack of physiological relevance is not an issue to the editors of NCB for publication, then at the very least it would be critical to demonstrate functional relevance *in vitro* using an *in vitro* assay in macrophages (antigen presentation, antigen processing etc.).

Such assays in macrophages related to macropinosomes and Na⁺/Cl⁻ flux in immune surveillance have previous been established (see Freeman et al., Science, 2020 doi: 10.1126/science.aaw9544)

Reviewer #3:

Remarks to the Author:

The authors have adequately addressed the concerns raised in my first review. Additionally, I think that the other two reviewers' concerns have been thoroughly addressed. As explained in my first review, I think this work provides an important advance in understanding of macropinosome biology.

GUIDELINES FOR SUBMISSION OF NATURE CELL BIOLOGY ARTICLES

READABILITY OF MANUSCRIPTS – Nature Cell Biology is read by cell biologists from diverse backgrounds, many of whom are not native English speakers. Authors should aim to communicate their findings clearly, explaining technical jargon that might be unfamiliar to non-specialists, and avoiding non-standard abbreviations. Titles and abstracts should concisely communicate the main

25findings of the study, and the background, rationale, results and conclusions should be clearly explained in the manuscript in a manner accessible to a broad cell biology audience. Nature Cell Biology uses British spelling.

ARTICLE FORMAT

ABSTRACT – should not exceed 150 words and should be unreferenced. This paragraph is the most visible part of the paper and should briefly outline the background and rationale for the work, and accurately summarize the main results and conclusions. Key genes, proteins and organisms should be specified to ensure discoverability of the paper in online searches.

TEXT – the main text consists of the Introduction, Results, and Discussion sections and must not exceed 3500 words including the abstract. The Introduction should expand on the background relating to the work. The Results should be divided in subsections with subheadings, and should provide a concise and accurate description of the experimental findings. The Discussion should expand on the findings and their implications. All relevant primary literature should be cited, in particular when discussing the background and specific findings.

REFERENCES – are limited to a total of 70 in the main text and Methods combined,. They must be numbered sequentially as they appear in the main text, tables and figure legends and Methods and must follow the precise style of Nature Cell Biology references. References only cited in the Methods

should be numbered consecutively following the last reference cited in the main text. References only associated with Supplementary Information (e.g. in supplementary legends) do not count toward the total reference limit and do not need to be cited in numerical continuity with references in the main text. Only published papers can be cited, and each publication cited should be included in the numbered reference list, which should include the manuscript titles. Footnotes are not permitted.

Methods should be written concisely, but should contain all elements necessary to allow interpretation and replication of the results. As a guideline, Methods sections typically do not exceed 3,000 words. The Methods should be divided into subsections listing reagents and techniques. When citing previous methods, accurate references should be provided and any alterations should be noted. Information must be provided about: antibody dilutions, company names, catalogue numbers and clone numbers for monoclonal antibodies; sequences of RNAi and cDNA probes/primers or company names and catalogue numbers if reagents are commercial; cell line names, sources and information on cell line identity and authentication. Animal studies and experiments involving human subjects must be reported in detail, identifying the committees approving the protocols. For studies involving human subjects/samples, a statement must be included confirming that informed consent was obtained. Statistical analyses and information on the reproducibility of experimental results should be provided in a section titled "Statistics and Reproducibility".

All Nature Cell Biology manuscripts submitted on or after March 21 2016, must include a Data availability statement as a separate section after Methods but before references, under the heading "Data Availability". For Springer Nature policies on data availability see <http://www.nature.com/authors/policies/availability.html>; for more information on this particular policy see <http://www.nature.com/authors/policies/data/data-availability-statements-data-citations.pdf>. The Data availability statement should include:

- Accession codes for primary datasets (generated during the study under consideration and designated as "primary accessions") and secondary datasets (published datasets reanalysed during the study under consideration, designated as "referenced accessions"). For primary accessions data should be made public to coincide with publication of the manuscript. A list of data types for which submission to community-endorsed public repositories is mandated (including sequence, structure, microarray, deep sequencing data) can be found here <http://www.nature.com/authors/policies/availability.html#data>.
- Unique identifiers (accession codes, DOIs or other unique persistent identifier) and hyperlinks for datasets deposited in an approved repository, but for which data deposition is not mandated (see here for details <http://www.nature.com/sdata/data-policies/repositories>).
- At a minimum, please include a statement confirming that all relevant data are available from the authors, and/or are included with the manuscript (e.g. as source data or supplementary information), listing which data are included (e.g. by figure panels and data types) and mentioning any restrictions

27on availability.

- If a dataset has a Digital Object Identifier (DOI) as its unique identifier, we strongly encourage including this in the Reference list and citing the dataset in the Methods.

We recommend that you upload the step-by-step protocols used in this manuscript to the Protocol Exchange. More details can found at www.nature.com/protocolexchange/about.

DISPLAY ITEMS – main display items are limited to 6-8 main figures and/or main tables. For Supplementary Information see below.

FIGURES – Colour figure publication costs \$395 per colour figure. All panels of a multi-panel figure must be logically connected and arranged as they would appear in the final version. Unnecessary figures and figure panels should be avoided (e.g. data presented in small tables could be stated briefly in the text instead).

All imaging data should be accompanied by scale bars, which should be defined in the legend. Cropped images of gels/blots are acceptable, but need to be accompanied by size markers, and to retain visible background signal within the linear range (i.e. should not be saturated). The boundaries of panels with low background have to be demarked with black lines. Splicing of panels should only be considered if unavoidable, and must be clearly marked on the figure, and noted in the legend with a statement on whether the samples were obtained and processed simultaneously. Quantitative comparisons between samples on different gels/blots are discouraged; if this is unavoidable, it has to be performed for samples derived from the same experiment with gels/blots were processed in parallel, which needs to be stated in the legend.

- We accept PowerPoint (.PPT) files if they are fully editable. However, please refrain from adding

28PowerPoint graphical effects to objects, as this results in them outputting poor quality raster art. Text used for PowerPoint figures should be Helvetica (preferred) or Arial.

Regardless of format, all figures must be vector graphic compatible files, not supplied in a flattened raster/bitmap graphics format, but should be fully editable, allowing us to highlight/copy/paste all text and move individual parts of the figures (i.e. arrows, lines, x and y axes, graphs, tick marks, scale bars etc). The only parts of the figure that should be in pixel raster/bitmap format are photographic images or 3D rendered graphics/complex technical illustrations.

Supplementary items should relate to a main text figure, wherever possible, and should be mentioned sequentially in the main manuscript, designated as Supplementary Figure, Table, Video, or Note, and numbered continuously (e.g. Supplementary Figure 1, Supplementary Figure 2, Supplementary Table

291, Supplementary Table 2 etc.).

Unprocessed scans of all key data generated through electrophoretic separation techniques need to be presented in a supplementary figure that should be labeled and numbered as the final supplementary figure, and should be mentioned in every relevant figure legend. This figure does not count towards the total number of figures and is the only figure that can be displayed over multiple pages, but should be provided as a single file, in PDF or TIFF format. Data in this figure can be displayed in a relatively informal style, but size markers and the figures panels corresponding to the presented data must be indicated.

The total number of Supplementary Figures (not including the “unprocessed scans” Supplementary Figure) should not exceed the number of main display items (figures and/or tables (see our Guide to Authors and March 2012 editorial <http://www.nature.com/ncb/authors/submit/index.html#supinfo>; <http://www.nature.com/ncb/journal/v14/n3/index.html#ed>). No restrictions apply to Supplementary Tables or Videos, but we advise authors to be selective in including supplemental data.

GUIDELINES FOR EXPERIMENTAL AND STATISTICAL REPORTING

REPORTING REQUIREMENTS – To improve the quality of methods and statistics reporting in our papers we have recently revised the reporting checklist we introduced in 2013. We are now asking all life sciences authors to complete two items: an Editorial Policy Checklist (found here <https://www.nature.com/authors/policies/Policy.pdf>) that verifies compliance with all required editorial policies and a Reporting Summary (found here <https://www.nature.com/authors/policies/ReportingSummary.pdf>) that collects information on experimental design and reagents. These documents are available to referees to aid the evaluation of the manuscript. Please note that these forms are dynamic ‘smart pdfs’ and must therefore be downloaded and completed in Adobe Reader. We will then flatten them for ease of use by the reviewers. If you would like to reference the guidance text as you complete the template, please access these flattened versions at <http://www.nature.com/authors/policies/availability.html>.

STATISTICS – Wherever statistics have been derived the legend needs to provide the n number (i.e. the sample size used to derive statistics) as a precise value (not a range), and define what this value represents. Error bars need to be defined in the legends (e.g. SD, SEM) together with a measure of centre (e.g. mean, median). Box plots need to be defined in terms of minima, maxima, centre, and percentiles. Ranges are more appropriate than standard errors for small data sets. Wherever statistical significance has been derived, precise p values need to be provided and the statistical test used needs to be stated in the legend. Statistics such as error bars must not be derived from $n < 3$. For sample sizes of $n < 5$ please plot the individual data points rather than providing bar graphs. Deriving

30statistics from technical replicate samples, rather than biological replicates is strongly discouraged. Wherever statistical significance has been derived, precise p values need to be provided and the statistical test stated in the legend.

Author Rebuttal, first revision:

Detailed response to the reviewers of the revised manuscript 'Proton-gated anion transport..' by Zeziulia...Jentsch

Our revised manuscript was sent out to reviewers 2 and 3, but not to reviewer 1, who was very positive in the first round of review. While reviewer 3 is fully satisfied by our revision, and states that in her/his opinion, we have fully satisfied also the requests of both other reviewers, reviewer 2, who stated that our findings are 'very novel' in the first round, likes us to perform more experiments aimed at investigating downstream effects of impaired macropinosome shrinkage. However, besides being beyond the scope of our paper, the suggested new experiments can only demonstrate a role of ASOR, but not of ASOR-dependent macropinosome resolution – as ASOR may have (and likely has) other roles in addition to MP resolution.

Detailed response to the reviewers:

Reviewer #2:

Remarks to the Author:

In the revised manuscript, Zeziulia et al have made text modifications and have added new data to the manuscript. In the response to reviewers, they have also provided additional data that pertain to some of

31the specific reviewer critiques. While the efforts to improve the manuscript are applauded, there are still some remaining concerns that should be considered:

Thank you for appreciating our efforts and the new experiments, which we have included in response to this reviewer in our first revision.

1. To modify some of the claims in the manuscript, the authors have amended the abstract in an attempt to restrict the statements to macrophages; however, the modified statements are still somewhat misleading. Specifically, the following statement is misleading: “We now identify the missing Cl⁻ channel as ASOR/TMEM206, a ubiquitously expressed proton-activated Cl⁻ channel involved in acid-induced cell death and stroke.” Modifying this statement to include “in macrophages” would be more accurate and a better reflection of the data in the manuscript.

We thought that we had sufficiently addressed this concern in the first revision, where we had stated: ‘Shrinkage of macrophage macropinosomes depends on Na⁺ efflux through TPC channels and Cl⁻ exit through unknown channels. Relieving osmotic pressure facilitates vesicle budding, positioning osmotic shrinkage upstream of vesicular sorting and trafficking. We now identify the missing Cl⁻ channel as ASOR/TMEM206’, where ‘the’ refers to macrophages. To satisfy the request of the reviewer, we have now inserted ‘the missing macrophage Cl⁻ channel as ASOR/TMEM206.’ in the abstract, line 33. However, we are not really happy with this change as it seems to wrongly suggest that this channel is only found on macrophage MPs, even though we showed its role also for HT-1080 cancer cells and believe the role of ASOR in MP resolution is much more general.

The new data in the HT-1080 cells support this change. The fact that TMEM206 functions in these cells in macropinosome shrinkage broadens the scope of the work;

Thank you for appreciating this additional experiment.

however, it is quite evident that TMEM206 knockdown is not as effective in the HT-1080 cells as it is in the macrophages (comparing Fig. 4d/e to Fig. 4a/b). Based on these new experiments, it seems that either Cl⁻ channels are only partially required for macropinosome resolution in cancer cells or Cl⁻ channels other than TMEM206 are playing a role in the cancer cells.

Although the new experiment (Fig. 4d,e, done in response to this reviewer) might suggest at first glance that there might be other Cl channels in MP resolution in cancer cells, the mildly reduced effect of ASOR KD in HT-1080 compared to ASOR KO in BMDMs can be easily explained by a combination of two factors:

- (1) *The efficiency of ASOR KD in HT-1080 cells with the siRNA pool is 80% (see Western quantification in Suppl. Fig. 4e), which compares to 100% loss of ASOR in BMDMs derived from constitutive KO mice (Suppl. Fig. 4d).*
- (2) *EGF-stimulated macropinosomes in HT-1080 cells are smaller than MCSF-stimulated MPs of BMDMs, making the optical determination of shrinkage more challenging and possibly introducing some minor systematic error in size determination.*

The point of that experiment is to show that also in cancer cells ASOR plays a major role in MP resolution, as requested by the reviewer. There is no indication that cancer cells may use different Cl channels. Of course, we can never exclude a minor role of another Cl channel in MP resolution. Indeed our work points to a minor role of the CIC-5 Cl/H exchanger (which we show to be present on MPs in our work).

To clarify this point also for the readers, we have now stated in lines 105-107: 'In human HT-1080 cancer cells²³, an 80% reduction of TMEM206 protein levels by siRNA (Suppl. Fig. 4e) sufficed to slow the resolution of MPs that were generated in response to EGF (Fig. 4d-f).'

*A more general concern of the reviewer seems to be that there might be other Cl-channels or transporters that can contribute to MP resolution. Of course, we agree that other anion channels or transporters may play a minor role in parallel to functionally predominant ASOR – in the end, also *Tmem206*^{-/-} MPs shrink (much more slowly), and also TPC1/2 KO MPs shrink finally. One of these transporters is probably CIC-5, and possibly other CLCs later on.*

We have now addressed this issue in the last paragraph of Discussion (lines 325-328):

'In conclusion, acid-activated ASOR/TMEM206 Cl⁻ channels, together with luminal acidification and previously identified TPCs⁵, are crucial for macropinosome shrinkage (Fig. 6a). Macropinosomes express additional, minor Cl⁻ exit pathways such as CIC-5 (Fig. 5j,k, Suppl. Fig. 16d) because they shrink, albeit at a much slower rate, also in the absence of ASOR.'

Additionally, these experiments seemingly still only include a single targeting siRNA.

As stated in Methods, we used a SMART pool siRNA from Dharmacon, containing 4 different siRNAs, which does not per se exclude off-target effects. Indeed when I serve as reviewer, I always request 2 independent siRNAs (or siRNA pools) for experiments on which key conclusions depend (a standard unfortunately not always adhered to in publications). However, the situation is different here: We have used primary BMDMs from three different Tmem206^{-/-} mouse lines generated with three different sgRNAs (a high standard rarely found in the literature) plus functional rescue by transfection. We wanted

33to have rock-solid evidence for a role of ASOR in MP resolution. The added data for HT-1080 cells beautifully confirm these findings. Moreover, the effect of ASOR on MP resolution is mechanistically very direct in contrast to other experiments (such as growth assays) that involve many intermediate processes. The probability that in HT-1080 cells the siRNA pool impairs MP resolution by an off-target effect rather than by the measured 80% decrease in ASOR expression levels is close to zero.

2. To address the functionality, instead of using a macrophage-based assay, the authors have provided a cancer cell assay where viability is assessed (Fig. 6e). In this new experiment, TMEM206 KO in MIA PaCa-2 cells results in enhanced viability in the presence of BSA. These data are not explained mechanistically, but the authors speculate that it could relate to recycling of BSA. The authors also claim that this is in line with the initial macropinosome initiation steps being unaffected by TMEM206 in macrophages (Suppl Fig. 5). There are several unresolved issues with these data. TMEM206 having a role in macropinosome shrinkage in MIA PaCa-2 cells is not shown. This should be demonstrated, or the viability assays should be done in the HT-1080 cells.

These experiments were newly performed in response to reviewer 2, who suggested to study cancer cells, although we consider these experiments to be rather beyond the scope of our work. They have yielded important results. While it might be good to show MP shrinkage and albumin-dependent cancer cell nutrition in the same cell line, we had good reasons for using different cell lines for these specific experiments.

We have used MIA-Paca2 pancreatic cells because they are a 'gold standard' in the field. They have e.g. been used by Commisso et al in their pioneering work (Nature 2013) and by many other groups. Also here, we were very anxious to avoid artifacts due to off-target effects or clonal selection. We used two different sgRNAs to completely disrupt Tmem206 in these cells and used three different clones for both WT and KO in our growth experiments (Fig. 6e). The effect of albumin was seen in every single one of these clonal cell lines.

We were not able to measure MP resolution in these cells, however, for technical reasons: In MIA-Paca2 cells, which only express low levels of EGF-R, macropinocytosis cannot be efficiently stimulated by EGF such as in HT-1080 cells, nor by M-CSF as in BMDMs. The rather small MPs resulting from constitutive macropinocytosis cannot be followed sufficiently well by light microscopy.

We therefore determined MP resolution in HT-1080 cancer cells where EGF stimulation leads to rather large MPs (though smaller than in M-CSF-stimulated BMDMs). We tried hard, but it was nearly impossible to obtain CRISPR-Cas engineered HT-1080 KO clones because these cells are very difficult to

culture at clonal densities. Therefore, we had to resort to siRNA KD in HT-1080 cells (see above). While siRNA KD is fine for assessing MP resolution, it is not suitable for albumin-complementation growth experiments where the difference between WT and KD is less pronounced than with MP resolution. KD (80%) is relatively inefficient compared to CRSPR KO, and more importantly this KD efficiency will not be maintained over the 72 hour growth phase.

We therefore resorted to two different cancer cell lines with the bonus of showing the importance of ASOR not only in one, but in two cancer cell lines.

Since this reasoning is too long to be included in the main text, we have now justified the use of the cell lines in the Supplement, Methods, in the Cell Culture section, page 37-38:

'Among the cancer cell lines, MIA-Paca2 cells were used for growth assays under nutrient restriction (Fig. 6e), for which they are a well-established model system²⁷, whereas HT-1080 were used to examine macropinosome resolution because these cells, but not in MIA-Paca2 cells²⁸, express sufficiently high levels of EGF-receptors for allowing acute stimulation of macropinocytosis.'

The results of the viability data suggests that TMEM206 has no role in preventing acidification of macropinosomes or fusion with the lysosome, which is a critical step in macropinosome function at least in cancer cells. This is not assessed, which could have been done with a simple DQ-BSA assay. Also, the data suggest that BSA utilization for survival is actually increased, which could be linked to either more efficient BSA degradation or an effect on recycling. Both of these scenarios are very easy to test. It is not clear how the BMDM migration assays relate to macropinosome resolution, as those data could be related to other functions of ASOR.

We agree that growth and migration assays, although technically easy to perform, depend on many intracellular signaling cascades and have to be controlled and interpreted extremely carefully. We tried to avoid pitfalls owing to clonal selection and off-target effects as described above. These experiments do show (as requested) physiological consequences of ASOR disruption. We have provided reasonable hypotheses how this fits to an impairment of ASOR-dependent MP shrinkage and vesicle recycling, similar to the work published by Freeman and Grinstein (see below).

While our work clearly shows that ASOR has a major role in MP shrinkage, we certainly cannot exclude that ASOR KO affects complex outcomes (such as cell growth and migration, or antigen presentation) by mechanisms unrelated to macropinosome shrinkage. The reviewer has a very valid point here. This point actually argues against the inclusion of additional functional data that are much more indirect than the clear-cut effect on MP shrinkage or pH, the focus of our work. The reviewer also suggests that ASOR has

no role in preventing acidification (she/he probably meant ASOR_KO rather than ASOR). Indeed, ASOR KO leads to more acidification, as measured experimentally and as robustly predicted by modeling (Fig. 5a, i). This increased acidification might contribute to earlier and more efficient albumin degradation in our growth assay. We thank the reviewer for raising this point.

We have now discussed this point, lines 292-295: 'These results are consistent with unchanged rate of macropinosome formation in TMEM206^{-/-} cells (Suppl. Fig. 5) in conjunction with impaired recycling of internalized albumin and increased MP acidification (Fig. 5a, i). The slow resolution of KO MPs may give them more time to fuse with degradative lysosomes while still containing substantial amounts of albumin and being over-acidified, leading to an increased cellular supply of amino-acids.

These two effects of ASOR KO (reduced MP shrinkage and increased acidification) are intrinsically linked. It would be extremely difficult, if not impossible, to separate these effects. A precise clarification of all the steps leading from ASOR disruption to impaired cell migration and cancer nutrition is clearly beyond the scope of our work and may even not be achievable at all. Nevertheless, we are glad to include these data.

I would also like to state another point here: of course, impairment of MP shrinkage does not abolish macropinocytosis, as shown here (and for TPCs in the Freeman..Grinstein work). Compared to a total disruption of macropinocytosis, effects will be smaller and might be qualitatively different (such as the better growth of cancer cells upon TMEM206 disruption as shown by us). Please consider that our data are based on carefully controlled KO or KD of ASOR, but not on unspecific inhibitors. By contrast, many studies on cancer cell nutrition and immune cells infer a role of macropinocytosis from the inhibitory effect of EIPA, an amiloride derivate. However, amiloride and EIPA inhibit many types of Na/H exchangers, Na channels, and Na/Ca exchangers, thereby influencing cytosolic and organellar concentrations of H⁺ and Ca²⁺. This is summarized in the review by J. Canton (Front. Immunol. 2018, PMID: 30333835) who writes on page 4: 'Therefore, the effect of amiloride derivatives on macropinocytosis is incidental and cannot be used alone as specific inhibitor of macropinocytosis'. In comparison to many cancer cell nutrition studies, our work employing Tmem206 KO and KD is considerably cleaner.

3. It is fair to argue that establishing the in vivo physiological relevance could be time-consuming and costly. If this lack of physiological relevance is not an issue to the editors of NCB for publication, then at the very least it would be critical to demonstrate functional relevance in vitro using an in vitro assay in macrophages (antigen presentation, antigen processing etc.). Such assays in macrophages related to

macropinosomes and Na⁺/Cl⁻ flux in immune surveillance have previously been established (see Freeman et al., Science, 2020 doi: 10.1126/science.aaw9544)

*We are of course aware of the excellent work of Freeman and Grinstein and have tried some of their approaches. For instance, we studied Mac-1 ($\alpha_M\beta_2$ integrin) recycling, but experiments were highly variable in our hands. Importantly, however, in their experiment Freeman et al. did not use TPC KO cells, but rather incubated cells with a PIKfyve inhibitor (YM201636) or kept them in Na⁺-free medium for an hour or more. While YM inhibits TPC activity by changing PIPs, many other cellular processes are influenced by PIPs. The same is true for Na⁺ removal (replaced (often by K⁺) in the culture medium for an hour or more. This contrasts with the replacement just in the lumen of MP in resolution experiments, achieved by changing extracellular ion concentrations only during the few minutes of M-CSF triggered MP formation – this makes MP resolution experiments so beautiful and conclusive. By contrast, replacing extracellular Na⁺ during an hour or more likely changes membrane potential, cytosolic Ca²⁺ through V-dependent Ca-channels or Na/Ca exchangers, pH through NHEs, etc., in addition to Na-flux through TPCs (the preferred interpretation of Freeman et al.). In most functional assays (e.g. transferrin recycling, neutrophil swarming, EGF-R recycling) Freeman et al. used PIKfyve inhibition, long-term general Na⁺ removal, or tetrandrine, an inhibitor of TPC channels, instead of TPC KO cells. Tetrandrine is very unspecific: it is a well-known inhibitor of Ca-channels, **inhibits also Ca-dependent K-channels, some PKC isoforms, and probably other processes. Importantly, even functional outcomes of TPC disruption cannot prove that these are owed to reduced MP shrinkage – they may well result from other roles of TPCs (which are not only Na-channels, the role important for the work of Freeman et al, but also mediate NAADP-triggered Ca-release, and impact many cellular functions).***

Thus, the functional assays that Freeman et al. added to their great manuscript are compatible with being caused by reduced macropinosome shrinkage, but do not prove this notion. Many experiments not even prove that these assays reflect alteration of TPC function as they were obtained with unspecific tools. This critique does not affect the main 'resolution' part of the work of Freeman/Grinstein, which is truly inspirational.

We thus would not like to follow the suggestion of the reviewer and include more 'physiological' data on diverse functions of ASOR, many of which likewise may not be related to its role in MP shrinkage (a possibility also mentioned by rev 2). Our work is concerned with the mechanism of macropinosome shrinkage, identifies several transporters involved (ASOR/TMEM206, CIC-5, H⁺-ATPase) and integrates experimental findings in a quantitative model. Addition of the experiments suggested by reviewer 2 would be beyond the scope of our paper and would have the potential to rather confuse our work.

Reviewer #3:

Remarks to the Author:

The authors have adequately addressed the concerns raised in my first review. Additionally, I think that the other two reviewers' concerns have been thoroughly addressed. As explained in my first review, I think this work provides an important advance in understanding of macropinosome biology.

We thank the reviewer, who is fully satisfied by our revision, for carefully assessing our manuscript and for stating that our work provides an important advance in macropinosome biology.

Decision Letter, second revision:

8th March 2022

Dear Dr. Jentsch,

Thank you for submitting your revised manuscript "Proton-gated anion transport governs endocytic vacuole shrinkage" (NCB-J46189B). It has now been seen by the original Referee #2 and their comments are below. The reviewer finds that the paper has improved in revision, and therefore we'll be happy in principle to publish it in Nature Cell Biology, pending minor revisions to comply with our editorial and formatting guidelines.

****The current version of your manuscript is in a PDF format, please email us a copy of the file in an editable format (Microsoft Word or LaTeX)-- we can not proceed with PDFs at this stage.****

After we receive the Word file, we will be performing detailed checks on your paper and will send you a checklist detailing our editorial and formatting requirements in about a week. Please do not upload the final materials and make any revisions until you receive this additional information from us.

Thank you again for your interest in Nature Cell Biology. Please do not hesitate to contact me if you have any questions.

Sincerely,

Melina

Melina Casadio, PhD
Senior Editor, Nature Cell Biology

38ORCID ID: <https://orcid.org/0000-0003-2389-2243>

Reviewer #2 (Remarks to the Author):

Authors have adequately addressed my concerns. This will be an impactful addition to the literature that establishes mechanistic understanding of how macropinosomes shrink.

22nd March 2022

Dear Dr. Jentsch,

Thank you for your patience as we've prepared the guidelines for final submission of your Nature Cell Biology manuscript, "Proton-gated anion transport governs endocytic vacuole shrinkage" (NCB-146189B). Please carefully follow the step-by-step instructions provided in the attached file, and add a response in each row of the table to indicate the changes that you have made. Please also check and comment on any additional marked-up edits we have proposed within the text. Ensuring that each point is addressed will help to ensure that your revised manuscript can be swiftly handed over to our production team.

We would like to start working on your revised paper, with all of the requested files and forms, as soon as possible (preferably within one week). Please get in contact with us if you anticipate delays.

In recognition of the time and expertise our reviewers provide to Nature Cell Biology's editorial process, we would like to formally acknowledge their contribution to the external peer review of your manuscript entitled "Proton-gated anion transport governs endocytic vacuole shrinkage". For those reviewers who give their assent, we will be publishing their names alongside the published article.

Nature Cell Biology offers a Transparent Peer Review option for new original research manuscripts submitted after December 1st, 2019. As part of this initiative, we encourage our authors to support increased transparency into the peer review process by agreeing to have the reviewer comments, author rebuttal letters, and editorial decision letters published as a Supplementary item. When you submit your final files please clearly state in your cover letter whether or not you would like to participate in this initiative. Please note that failure to state your preference will result in delays in accepting your manuscript for publication.

39Cover suggestions

As you prepare your final files we encourage you to consider whether you have any images or illustrations that may be appropriate for use on the cover of Nature Cell Biology.

Nature Cell Biology has now transitioned to a unified Rights Collection system which will allow our Author Services team to quickly and easily collect the rights and permissions required to publish your work. Approximately 10 days after your paper is formally accepted, you will receive an email in providing you with a link to complete the grant of rights. If your paper is eligible for Open Access, our Author Services team will also be in touch regarding any additional information that may be required to arrange payment for your article.

Please note that *Nature Cell Biology* is a Transformative Journal (TJ). Authors may publish their research with us through the traditional subscription access route or make their paper immediately open access through payment of an article-processing charge (APC). Authors will not be required to make a final decision about access to their article until it has been accepted. Find out more about Transformative Journals

For information regarding our different publishing models please see our Transformative

40Journals page. If you have any questions about costs, Open Access requirements, or our legal forms, please contact ASJournals@springernature.com.

Please use the following link for uploading these materials:
[REDACTED]

Best regards,

Nyx Hills
Staff
Nature Cell Biology

On behalf of

Melina Casadio, PhD
Senior Editor, Nature Cell Biology
ORCID ID: <https://orcid.org/0000-0003-2389-2243>

Reviewer #2:

Remarks to the Author:

Authors have adequately addressed my concerns. This will be an impactful addition to the literature that establishes mechanistic understanding of how macropinosomes shrink.

Author rebuttal, second revision:

Response to the reviewer

We are glad that Reviewer 2, whose comment is copied below, now states that we have adequately addressed her/his concerns.

Reviewer #2 (Remarks to the Author):

Authors have adequately addressed my concerns. This will be an impactful addition to the literature that

establishes mechanistic understanding of how macropinosomes shrink.

Final Decision Letter:

Dear Dr Jentsch,

I am pleased to inform you that your manuscript, "Proton-gated anion transport governs macropinosome shrinkage", has now been accepted for publication in Nature Cell Biology.

Please note that *Nature Cell Biology* is a Transformative Journal (TJ). Authors may publish their

42research with us through the traditional subscription access route or make their paper immediately open access through payment of an article-processing charge (APC). Authors will not be required to make a final decision about access to their article until it has been accepted. Find out more about Transformative Journals

If you have not already done so, we strongly recommend that you upload the step-by-step protocols used in this manuscript to the Protocol Exchange (www.nature.com/protocolexchange), an open online resource established by Nature Protocols that allows researchers to share their detailed experimental know-how. All uploaded protocols are made freely available, assigned DOIs for ease of citation and are fully searchable through nature.com. Protocols and Nature Portfolio journal papers in which they are used can be linked to one another, and this link is clearly and prominently visible in the online versions of both papers. Authors who performed the specific experiments can act as primary authors for the Protocol as they will be best placed to share the methodology details, but the Corresponding Author of the present research paper should be included as one of the authors. By uploading your Protocols to Protocol Exchange, you are enabling researchers to more readily reproduce or adapt the methodology you use, as well as increasing the visibility of your protocols and papers. You can also establish a dedicated page to collect your lab Protocols. Further information can be found at www.nature.com/protocolexchange/about

With kind regards,

43Melina

Melina Casadio, PhD
Senior Editor, Nature Cell Biology
ORCID ID: <https://orcid.org/0000-0003-2389-2243>

** Visit the Springer Nature Editorial and Publishing website at www.springernature.com/editorial-and-publishing-jobs for more information about our career opportunities. If you have any questions please click here.**